# Uni-DPO: A Unified Paradigm for Dynamic Preference Optimization of LLMs

**Shangpin Peng**[*,1]     **Weinong Wang**[*,†,2]     **Zhuotao Tian**[‡,1]     **Senqiao Yang**[3]

**Xing Wu**[4]  **Haotian Xu**[5]  **Chengquan Zhang**[6]  **Takashi Isobe**[5]  **Baotian Hu**[1]  **Min Zhang**[1]

[1]Harbin Institute of Technology, Shenzhen     [2]Xi'an Jiaotong University
[3]The Chinese University of Hong Kong     [4]University of Chinese Academy of Sciences
[5]Tsinghua University     [6]Huazhong University of Science and Technology
Code & Models: ⬤ https://github.com/pspdada/Uni-DPO

## Abstract

Direct Preference Optimization (DPO) has emerged as a cornerstone of reinforcement learning from human feedback (RLHF) due to its simplicity and efficiency. However, existing DPO-based methods typically treat all preference pairs equally, overlooking substantial variations in data quality and learning difficulty, which leads to inefficient data utilization and suboptimal performance. To address this limitation, we propose **Uni-DPO**, a unified dynamic preference optimization framework that jointly considers (a) the inherent quality of preference pairs and (b) the model's evolving performance during training. By adaptively reweighting samples based on both factors, Uni-DPO enables more effective use of preference data and achieves superior performance. Extensive experiments across models and benchmarks demonstrate the effectiveness and generalization of Uni-DPO. On textual tasks, Gemma-2-9B-IT fine-tuned with Uni-DPO surpasses the leading LLM, Claude 3 Opus, by 6.7 points on Arena-Hard. On mathematical and multimodal tasks, Uni-DPO consistently outperforms baseline methods across all benchmarks, providing strong empirical evidence of its effectiveness and robustness.

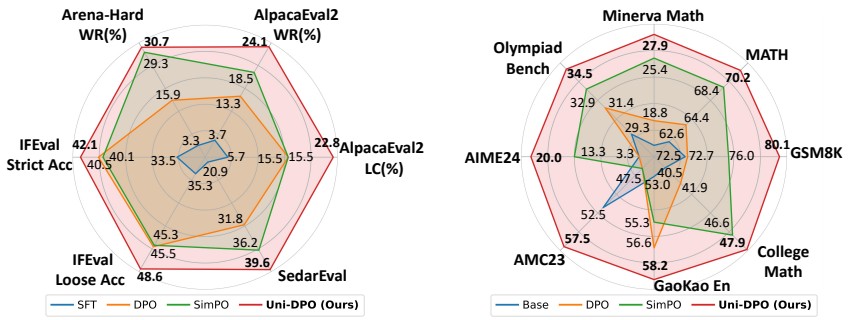

(a) Results on textual understanding tasks      (b) Results on mathematical reasoning tasks

## 1 Introduction

Incorporating human feedback has become essential for developing large language models (LLMs), as it aligns model behavior with human values and intentions, ensuring that generated outputs are useful, reliable, and harmless (Leike et al., 2018; Askell et al., 2021). Reinforcement learning from human feedback (RLHF) is a widely adopted paradigm for fine-tuning LLMs (Wang et al., 2024a). It typically involves training a reward model to capture human preferences and using it to optimize a policy via reinforcement learning algorithms such as Proximal Policy Optimization (PPO) (Schulman et al., 2017). Despite its effectiveness, this multi-stage pipeline is often time-consuming and suffers from the difficulty of training a reliable reward model (Casper et al., 2023).

---

*Equal contribution, †Project lead, ‡Corresponding author (tianzhuotao@hit.edu.cn).

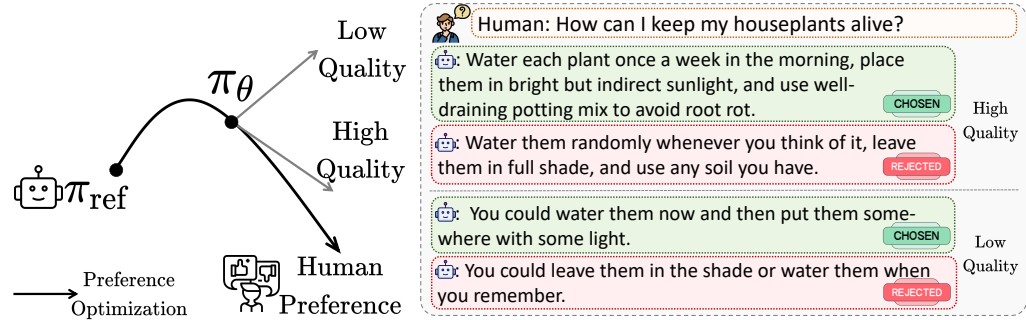

Figure 2: **Demonstrations of data quality and its effect.** During preference optimization, high-quality sample pairs exhibit clear distinctions and effectively reflect human preferences, whereas low-quality pairs have ambiguous differences between positive and negative samples, failing to represent human preferences accurately. Therefore, it is crucial to consider data quality when optimizing LLMs with preference data.

To streamline the training pipeline, Direct Preference Optimization (DPO) (Rafailov et al., 2023) reparameterizes the reward function so that the policy can be learned directly from the preference data without a reward model. Concretely, it uses the log-likelihood ratio between the policy and the reference model on the same input as an implicit reward signal, thereby rendering the training process simpler and efficient, while still maintaining performance (Jiang et al., 2024; Dubois et al., 2023).

**Key observations.** Although DPO and its recent variants (*e.g.,* SimPO (Meng et al., 2024)) have achieved promising performance, they treat all preference pairs equally during training, neglecting the varying quality levels present in the data collected for preference learning (Deng et al., 2025; Amini et al., 2024; Wu et al., 2024a; Wang et al., 2024f; Pattnaik et al., 2024). However, as demonstrated in Fig. 2, the high-quality preference pairs can help the model better align with the human preference. In contrast, the low-quality pairs with ambiguity between positive and negative samples may instead cause difficulty for representation learning. Therefore, the indiscriminate treatment of different data pairs may prevent the model from fully leveraging the data, thereby limiting the final performance.

An intuitive solution to this issue is to assign higher weights to high-quality data pairs during training, where *quality* can be quantified by the score difference between positive and negative samples as assessed by external experts. However, as shown in Fig. 5c, aggressively training on well-fitted high-quality pairs may lead to overfitting, ultimately degrading model performance (Wu et al., 2024b; Azar et al., 2024; Chen et al., 2024a). This naturally leads to a key question:

*How can we dynamically adjust the focus across training pairs by jointly considering both intrinsic data quality and the model's learning dynamics during training?*

**Our solution.** To address this challenge, we propose **Uni-DPO**, a dual-perspective optimization paradigm that jointly addresses two critical aspects of dynamic preference learning: (1) the inherent quality of the preference data and (2) the evolving learning dynamics of the model. The overall objective of the proposed Uni-DPO and the comparison with vanilla DPO is illustrated in Fig. 3.

Specifically, first, we incorporate a *quality-aware weight* $w_{\text{qual}}$ that adaptively prioritizes high-quality samples and down-weights low-quality ones. Then, we introduce a *performance-based weight* $w_{\text{perf}}$ that shifts the learning focus towards the sample pairs that are not well-fitted, thereby mitigating overfitting. Additionally, since DPO may decrease the probabilities of generating preferred samples (Rafailov et al., 2024; Pal et al., 2024), we introduce a calibrated negative log-likelihood loss $\mathcal{L}_{\text{c-NLL}}$ that targets difficult yet high-quality positive samples to further improve model's performance.

To this end, the overall preference learning objective is obtained by combining the above components with the vanilla DPO loss, thereby explicitly modeling both the *off-policy* aspect of preference optimization, which accounts for intrinsic data quality, and the *on-policy* aspect that captures model learning dynamics during training. This dual-perspective paradigm facilitates adaptive corrections during preference learning, resulting in improved final performance. By holistically addressing both facets of preference optimization within a *unified* framework, we introduce our method as Uni-DPO.

To summarize, our contributions are as follows:

- In this study, we reveal the fact that DPO and its recent variants treat all pairs equally during preference learning, potentially limiting the models' learning capacity and impeding performance.
- To tackle this issue, we introduce **Uni-DPO**. This method facilitates unified dynamic preference optimization of LLMs by adjusting the learning focus to informative training samples based on the intrinsic data quality and the model's learning dynamics.
- Uni-DPO is straightforward yet effective. The comprehensive experimental results on textual understanding tasks, math reasoning tasks, and multimodal tasks across multiple models and benchmarks demonstrate the superiority and generalization capabilities of our method.

## 2 BACKGROUND AND MOTIVATION

In this section, we briefly introduce the basic concepts and works relevant to this study in Sec. 2.1, establishing the necessary background. Following this, in Sec. 2.2, we outline our key insights and elucidate the motivations behind our proposed designs.

### 2.1 PRELIMINARIES

**DPO.** Direct Preference Optimization (DPO) (Rafailov et al., 2023) has emerged as a cornerstone technique within RLHF due to its simplicity and efficiency. DPO dispenses with an explicit reward model and instead learns a policy directly from preference data. By turning the traditional multi-stage RLHF pipeline into a single preference learning step, its training objective can be expressed as:

$$\mathcal{L}_{\text{DPO}} = -\mathbb{E}_{(x,y_w,y_l)\sim D}\left[\log\sigma\left(\beta\log\frac{\pi_\theta(y_w\mid x)}{\pi_{\text{ref}}(y_w\mid x)} - \beta\log\frac{\pi_\theta(y_l\mid x)}{\pi_{\text{ref}}(y_l\mid x)}\right)\right]. \tag{1}$$

Here, $D$ represents the preference learning dataset, $\sigma$ denotes the sigmoid function, $\pi_\theta$ indicates the policy model under training, and $\pi_{\text{ref}}$ represents the unchanged reference model. $y_w$ stands for the positive sample, and $y_l$ represents the negative sample, both based on input $x$. $\beta$ is a parameter controlling the deviation from the reference policy $\pi_{\text{ref}}$.

**The variants of DPO.** Recent works have provided theoretical insights and improvements to DPO. Some studies show that DPO can reduce the probabilities of preferred samples $y_w$ (Feng et al., 2024), causing LLMs to struggle with generating human-preferred outputs. To counter this, Pal et al. (2024) incorporates an auxiliary negative log-likelihood loss to better balance learning. Others find that DPO induces a length bias, where the policy model favors longer sequences (Meng et al., 2024; Park et al., 2024; Lu et al., 2024), which can be alleviated via length normalization (LN). Among variants, SimPO (Meng et al., 2024) stands out for removing the reference model, achieving better training efficiency and performance. More related works are discussed in Sec. F.

**Implicit reward margin.** During optimization with preference pairs, the *implicit reward signal* derived from DPO objective (see Eq. (1)) proves valuable, serving both as an effective metric for assessing alignment quality and as a useful tool for downstream analysis (Lu et al., 2024; Pan et al., 2025). Accordingly, the closed-form expression of DPO's implicit reward $r(x,y)$ is formulated as:

$$r(x,y) = \beta\log\frac{\pi_\theta(y\mid x)}{\pi_{\text{ref}}(y\mid x)} + \beta\log Z(x), \tag{2}$$

where $Z(x)$ denotes the partition function, independent of $\pi_\theta$. The implicit reward defined in Eq. (2) measures the difference in log-likelihood between the policy and the reference policy on the same sample. Based on this, we further define the *implicit reward margin* $\Delta_r$ as the difference between the implicit rewards of the preferred ($y_w$) and inferior ($y_l$) samples, which can be expressed as:

$$\Delta_r = r(x,y_w) - r(x,y_l). \tag{3}$$

A recent study (Houliston et al., 2024) shows that the implicit reward margin $\Delta_r$ of a pair $(y_w, y_l)$ reflects both the policy model's performance and its learning difficulty for that instance. In this study, the reward margin $\Delta_r$ will be utilized to dynamically adapt the focus towards various samples.

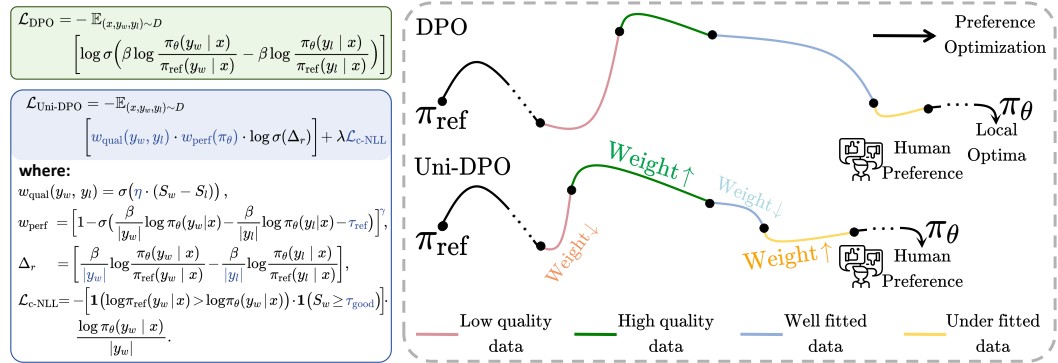

Figure 3: **Comparison of DPO and Uni-DPO objectives.** The Uni-DPO objective introduces a dual-perspective weighting mechanism, including a quality-aware weight $w_{\text{qual}}$, a performance-based weight $w_{\text{perf}}$, and a calibrated negative log-likelihood term $\mathcal{L}_{\text{c-NLL}}$ that emphasizes challenging and high-quality positive samples. **Left:** schematic illustration of the two objectives. **Right:** compared with DPO, Uni-DPO dynamically reweights data pairs during training, guiding the optimization trajectory toward regions that better reflect human preference.

## 2.2 KEY OBSERVATIONS

Although DPO simplifies RLHF into a single preference learning step, it implicitly assumes that every preference pair contributes equally to optimization. However, human-annotated or model-labeled preference data can vary widely in quality (Wu et al., 2024b; Amini et al., 2024; Wu et al., 2024a). Treating all sample pairs uniformly during optimization gives rise to the critical issues of under-utilization of high-quality pairs that could more effectively facilitate the policy learning compared to low-quality counterparts (Wu et al., 2024b; Penedo et al., 2023). A straightforward solution to this issue is allocating distinct training weights to various data pairs based on their quality.

However, we observe a potential mismatch between the inherent quality of a preference pair, measured by the score margin $(S_w - S_l)$ assigned by external experts, and the model's dynamic performance on that pair, captured by its reward margin $\Delta_r$ (see Eq. (3)). As shown in Fig. 4, high-quality pairs can already be well-fitted by the model. While such data aids alignment, overly emphasizing them with large training weights could amplify overfitting risks (Wu et al., 2024b), as shown in Fig. 5c, and thus may undermine the generalization performance. More details about this study are in Sec. A.2.

Consequently, this motivates us also to consider the model's learning dynamics when adjusting the training focus based on data quality, *i.e.,* specifically, by prioritizing underperforming pairs while downweighting well-fitted ones to mitigate overfitting. To this end, the contributions of data quality and model performance are balanced for weight assignment during the optimization process.

## 3 METHOD

### 3.1 OVERVIEW

To address the issues discussed in Sec. 2, in this section, we propose an adaptive weighting mechanism that dynamically adjusts the importance of each pair based on two key factors: (1) Data quality $(w_{\text{qual}})$, which reflects the inherent value of the preference pair $(y_w, y_l)$; (2) Model performance $(w_{\text{perf}})$, which measures how well the current policy $\pi_\theta$ aligns with the pair. To address all four cases mentioned in Fig. 4a, these two factors are combined multiplicatively $(w_{\text{qual}} \cdot w_{\text{perf}})$ in our method.

Additionally, recognizing the importance of difficult yet high-quality positive responses $(y_w)$, we incorporate a calibrated negative log-likelihood (c-NLL) loss $\mathcal{L}_{\text{c-NLL}}$ that selectively amplifies the policy's confidence in such examples. To this end, the overall training objective is formulated as:

$$\mathcal{L}_{\text{Uni-DPO}} = -\mathbb{E}_{(x,y_w,y_l)\sim D}\left[w_{\text{qual}}(y_w, y_l) \cdot w_{\text{perf}}(\pi_\theta) \cdot \log \sigma(\Delta_r)\right] + \lambda \mathcal{L}_{\text{c-NLL}}, \quad (4)$$

where $\Delta_r$ denotes the implicit reward margin defined in Eq. (3). The comparison between Uni-DPO and the vanilla DPO is presented in Fig. 3.

In the following sections, the detailed formulations of dual-perspective adaptive weights $w_{\text{qual}}$ and $w_{\text{perf}}$ are presented in Secs. 3.2 and 3.3, and the design of the c-NLL loss is elaborated in Sec. 3.4.

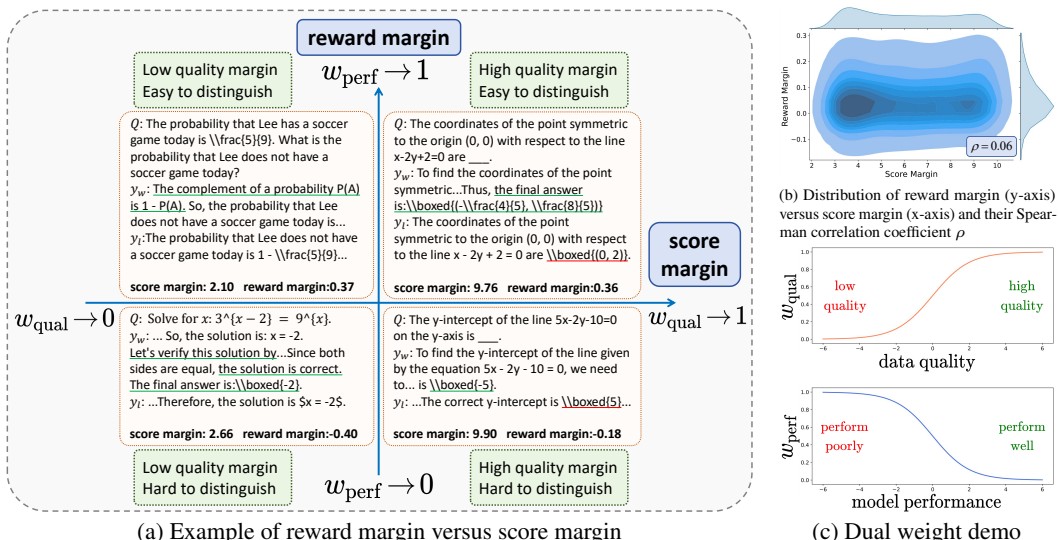

Figure 4: **Analysis of reward margin versus score margin.** (a) Illustrative reward margin versus score margin examples across training data. The four quadrants reflect the combinations of high/low data quality and easy/hard learning difficulty. (b) Distribution of reward margin (y-axis) versus score margin (x-axis), along with their Spearman correlation coefficient $\rho$, indicating data quality is not necessarily aligned with learning difficulty, suggesting the need to account for both factors during training. (c) Demonstration of dual-perspective weighting mechanism: the quality weight $w_{\mathrm{qual}}$ (top) rises with data quality, while the performance weight $w_{\mathrm{perf}}$ (bottom) decreases as model performance improves, ensuring updates target both high-quality yet under-fitted samples.

## 3.2 QUALITY WEIGHTING FACTOR

In this section, we propose a measure to quantify the quality of each preference pair $(y_w, y_l)$ based on the distinction between the preferred $(y_w)$ and inferior $(y_l)$ samples (Wang et al., 2024a). Specifically, we introduce a quality-based weighting factor $w_{\mathrm{qual}}$ for the preference pair $(y_w, y_l)$ that is defined as:

$$w_{\mathrm{qual}}(y_w,\, y_l) = \sigma\big(\eta \cdot (S_w - S_l)\big), \tag{5}$$

where $S_w$ and $S_l$ are the scalar quality scores for the preferred and inferior responses, derived from prior knowledge that reflects human preferences, either through human annotations, strong proprietary models (such as GPT-4 (Achiam et al., 2023)), or domain-specific reward models. Score sources in this work are detailed in Sec. 4, where we observe that better data quality yields better results. $\sigma$ denotes the sigmoid function to ensure $w_{\mathrm{qual}} \in [0, 1]$, and the scalar factor $\eta$ is used to adjust the score margin $(S_w - S_l)$ to an appropriate scale. In this work, $\eta$ is consistent across models and datasets.

A large score margin $(S_w - S_l)$ implies confident preference, making $w_{\mathrm{qual}}$ close to 1 and amplifying the training signal of this preference pair. A small margin indicates ambiguity, leading $w_{\mathrm{qual}}$ to minimize, filtering noisy or uncertain pairs, and enhancing the robustness of training.

## 3.3 PERFORMANCE WEIGHTING FACTOR

Although prioritizing high-quality samples may be beneficial for policy learning, as shown in Fig. 5c, overemphasizing well-fitted examples may cause overfitting (Wu et al., 2024b; Azar et al., 2024; Chen et al., 2024a). Therefore, the model's current performance is critical for dynamically rebalancing learning focus across training samples.

**Revisit the focal loss.** To better adapt to the model's learning dynamics during training, focal loss (FL) (Lin et al., 2017) has been introduced to address the imbalance learning issue in object detection tasks (Zou et al., 2023). It adds a modulating factor $(1 - p_{\mathrm{t}})^\gamma$ to the standard cross-entropy loss $\mathcal{L}_{\mathrm{CE}}(p_{\mathrm{t}}) = -\log(p_{\mathrm{t}})$, where $p_{\mathrm{t}}$ is the predicted probability for the target. This factor down-weights easy, well-fitted samples (with high $p_{\mathrm{t}}$) while preserving gradients for harder ones, thus rebalancing the training focus. The focal loss is defined as:

$$\mathcal{L}_{\mathrm{FL}}(p_{\mathrm{t}}) = -(1 - p_{\mathrm{t}})^\gamma \log(p_{\mathrm{t}}), \tag{6}$$

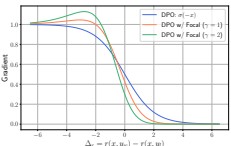
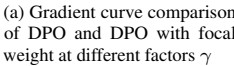
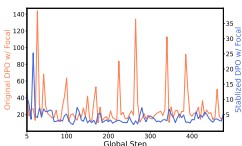
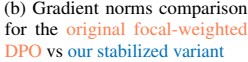
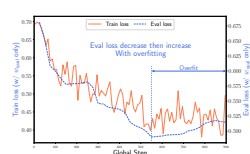
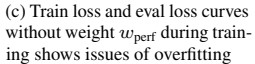
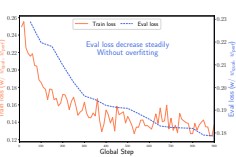

| (a) Gradient curve comparison of DPO and DPO with focal weight at different factors $\gamma$ | (b) Gradient norms comparison for the original focal-weighted DPO vs our stabilized variant | (c) Train loss and eval loss curves without weight $w_{\text{perf}}$ during training shows issues of overfitting | (d) Train loss and eval loss curves with weight $w_{\text{perf}}$ during training effectively mitigates overfitting |
|---|---|---|---|

Figure 5: **Illustration of model performance weighting.** (a) Focal-weighted DPO gradients are larger for hard examples and decrease as the model improves, encouraging the focus on harder examples. (b) Direct focal weighting (Eq. (7)) can lead to unstable training, whereas our stabilized form (Eq. (10)) applies a uniform constraint on each sample, resulting in more stable training. (c)-(d) Incorporating $w_{\text{perf}}$ can reduce overfitting.

where $\gamma \geq 0$ is a tunable parameter. When $\gamma = 0$, the focal loss becomes equivalent to the standard cross-entropy loss. As $\gamma$ increases, the modulating factor's influence strengthens, causing the loss function to increasingly discount well-classified examples and prioritize hard, informative ones.

**The direct integration with DPO.** Given the concept of focal loss, we first directly incorporate it into our framework as an adaptive weighting factor $w_{\text{focal}}$ based on the model's performance:

$$
\begin{aligned}
w_{\text{focal}} &= \left[1 - \sigma\left(\beta \log \frac{\pi_\theta(y_w|x)}{\pi_{\text{ref}}(y_w|x)} - \beta \log \frac{\pi_\theta(y_l|x)}{\pi_{\text{ref}}(y_l|x)}\right)\right]^\gamma \\
&= \left[1 - \sigma\left(\beta \log \pi_\theta(y_w|x) - \beta \log \pi_\theta(y_l|x) - (\beta \log \pi_{\text{ref}}(y_w|x) - \beta \log \pi_{\text{ref}}(y_l|x))\right)\right]^\gamma,
\end{aligned}
\tag{7}
$$

Here, $\gamma$ controls the sharpness of weight decay. This formulation ensures that when the implicit reward margin $\Delta_r$ (see Eq. (3)) is significantly large, the weight approaches zero, effectively reducing emphasis on already well-learned samples. The resulting focal-weighted DPO objective becomes:

$$
\mathcal{L}_{\text{DPO w/ FL}} = -\mathbb{E}_{(x,y_w,y_l)\sim D}\left[w_{\text{focal}}(\pi_\theta) \cdot \log \sigma(\Delta_r)\right],
\tag{8}
$$

and the corresponding gradient is given by (the derivation process is presented in Sec. B.2):

$$
\nabla_\theta \mathcal{L}_{\text{DPO w/ FL}} = -\mathbb{E}_{(x,y_w,y_l)\sim D}\left[(1 - \sigma(\Delta_r))^\gamma(1 - \sigma(\Delta_r) - \gamma\sigma(\Delta_r)\ln\sigma(\Delta_r))\right] \cdot \nabla_\theta(\Delta_r).
\tag{9}
$$

As illustrated in Fig. 5a, the gradient of the focal-weighted DPO objective peaks when the model performs poorly and diminishes as performance improves. This adaptive weighting prioritizes challenging examples by reducing the influence of well-fitted samples during training.

**Issues of the direct integration.** However, as shown in Fig. 5b, directly applying the focal weighting factor $w_{\text{focal}}$ into preference learning leads to unstable training dynamics. We attribute this to the focal factor imposing a rigid, per-sample constraint. Concretely, for every preference pair $(y_w, y_l)$, $w_{\text{focal}}$ requires the policy's log-probability gap $(\log\pi_\theta(y_w|x) - \log\pi_\theta(y_l|x))$ to surpass that of the reference model. While helpful for disambiguating hard examples, this constraint also forces the policy to further enlarge the margins on already well-handled cases, *i.e.,* the reference model's log-probability gap between $y_w$ and $y_l$ is already large, risking overfitting to noise or spurious samples. This suggests that extending the focal weighting mechanism to preference learning requires a relaxed formulation that reduces emphasis on well-fitted samples.

**Calibrated weighting factor.** To mitigate the above issue, we first apply length normalization (LN) following prior work (Meng et al., 2024; Park et al., 2024; Lu et al., 2024), ensuring the learning objective reflects preference differences rather than sequence length. Then, we *relax* the per-sample constraint by introducing a fixed, uniform *performance threshold* $\tau_{\text{ref}}$ across all training pairs. Since this threshold is fixed, the margin required from the policy model no longer depends on how well the reference model performs on a given pair. This decoupling reduces unnecessary optimization pressure on easy samples and lowers the risk of overfitting to already well-fitted cases. The resulting performance factor $w_{\text{perf}}$ is formulated as:

$$w_{\text{LN focal}} = \Big[ 1 - \sigma \Big( \frac{\beta}{|y_w|} \log \pi_\theta(y_w|x) - \frac{\beta}{|y_l|} \log \pi_\theta(y_l|x) - \underbrace{(\frac{\beta}{|y_w|} \log \pi_{\text{ref}}(y_w|x) - \frac{\beta}{|y_l|} \log \pi_{\text{ref}}(y_l|x))}_{\tau_{\text{ref}}} \Big) \Big]^\gamma$$

$$\xrightarrow{\text{unify margin}} w_{\text{perf}} = \Big[ 1 - \sigma \Big( \frac{\beta}{|y_w|} \log \pi_\theta(y_w|x) - \frac{\beta}{|y_l|} \log \pi_\theta(y_l|x) - \tau_{\text{ref}} \Big) \Big]^\gamma .$$
(10)

Here, $\tau_{\text{ref}}$ is a hyperparameter controlling the required performance margin. Empirically, increasing $\gamma$ strengthens the down-weighting of well-learned samples during training, which makes the optimal $\tau_{\text{ref}}$ rise, suggesting stronger margin constraints are needed for optimal performance (see Sec. 4.4). More details about the rationale for relaxing the original focal weight are discussed in Sec. E.4.

### 3.4 CALIBRATED NEGATIVE LOG-LIKELIHOOD LOSS

While the DPO objective enforces the log-likelihood of the preferred response ($y_w$) exceeds that of the inferior one ($y_l$), it does not explicitly promote a higher absolute probability for $y_w$ (Rafailov et al., 2024; Pal et al., 2024). Empirically, this "relative" optimization mechanism can suppress the probabilities of preferred responses ($y_w$), thereby hindering the ability of LLMs trained with DPO to generate outputs that align with human preferences. To remedy this, we introduce a calibrated negative log-likelihood (c-NLL) loss targeted exclusively at difficult yet high-quality positive samples:

$$\mathcal{L}_{\text{c-NLL}} = - \Big[ \underbrace{\mathbf{1}\big(\log \pi_{\text{ref}}(y_w \mid x) > \log \pi_\theta(y_w \mid x)\big)}_{\substack{\text{only active when the reference model's} \\ \text{likelihood exceeds the policy's}}} \quad \underbrace{\mathbf{1}\big(S_w \geq \tau_{\text{good}}\big)}_{\substack{\text{only active for} \\ \text{high-quality positive samples}}} \Big] \cdot \frac{\log \pi_\theta(y_w \mid x)}{|y_w|} .$$
(11)

The first indicator $\mathbf{1}(\cdot)$ ensures $\mathcal{L}_{\text{c-NLL}}$ is only applied when the reference model's log-likelihood exceeds that of the policy, and the second $\mathbf{1}(\cdot)$ restricts the penalty only to high-quality positive samples. This calibration strengthens the policy's confidence in challenging, high-quality responses without disturbing well-fitted or low-quality ones. Further discussions about $\mathcal{L}_{\text{c-NLL}}$ are in Sec. E.3.

## 4 EXPERIMENTS

In this section, we present a comprehensive suite of experiments to demonstrate the superior performance of Uni-DPO across multiple dimensions. In Sec. 4.1, we evaluate Uni-DPO on open-ended instruction-following tasks to verify its effectiveness in textual understanding; in Sec. 4.2, we investigate its ability to enhance mathematical reasoning; and in Sec. 4.3, we conduct ablation studies to assess the contribution of each component within Uni-DPO. In the appendix, we further present evaluations on multimodal tasks in Sec. D.2, along with additional ablation analyses in Sec. D.3.

### 4.1 TEXTUAL UNDERSTANDING EVALUATION

**Experimental settings.** We apply Uni-DPO to Llama3-8B (AI@Meta, 2024b) under two setups: *Base* and *Instruct*, and to Gemma-2-9B-IT (Team et al., 2024b). All training configurations are aligned with those used in SimPO to ensure a fair comparison.[1] Additionally, we extend our evaluation to Qwen2.5 (Yang et al., 2024a), a more recent and stronger model. Following the *Base* setup in SimPO, we perform off-policy preference training on the UltraFeedback (Cui et al., 2023) dataset. We primarily evaluate our method on four widely adopted open-ended instruction-following benchmarks: AlpacaEval 2.0 (Li et al., 2023b)[2], Arena-Hard v0.1 (Li et al., 2024a), IFEval (Zhou et al., 2023), and SedarEval (Fan et al., 2025). These benchmarks evaluate the models' conversational abilities across a diverse set of queries. We compare Uni-DPO against DPO and its improved variant, SimPO. Supervised fine-tuning (SFT) is also included as a baseline. All methods share identical training corpora to guarantee a rigorously fair evaluation. Details of evaluation settings are in Sec. D.1.

---

[1] We also compare with SimPO v0.2 setting, which utilizes preference labels from a stronger reward model ArmoRM (Wang et al., 2024c) for *Instruct* setup. For *Base* setup, we adopt GPT-4o (Hurst et al., 2024) to obtain higher-quality preference data. The details of the data generation process are in Sec. C.2.

[2] As noted by the original authors, the AlpacaEval results for SimPO were obtained using an older version of the evaluation toolkit, which is no longer compatible with the current setup. We re-evaluated SimPO using the publicly available checkpoint under the updated evaluation framework for a fair comparison.

Table 1: **Main evaluation result of textual understanding.** WR denotes the Win Rate, LC denotes the Length-Controlled win rate, and Acc. denotes the Accuracy. The best results are highlighted in **bold**. The results show that Uni-DPO consistently outperforms the SimPO and DPO methods across models and benchmarks.

| Model | Method | AlpacaEval2 | | Arena-Hard | IFEval | | SedarEval |
|---|---|---|---|---|---|---|---|
| | | LC(%) | WR(%) | WR(%) | Strict Acc.(%) | Loose Acc.(%) | Overall(%) |
| Llama3-8B Base | SFT | 5.7 | 3.7 | 3.3 | 33.5 | 35.3 | 20.85 |
| | DPO | 15.5 | 13.3 | 15.9 | **40.5** | 45.5 | 31.80 |
| | SimPO | 19.4 | 18.1 | 23.4 | **40.5** | 45.7 | 32.43 |
| | Uni-DPO | **23.8** | **20.5** | **23.9** | 38.1 | **47.9** | **38.49** |
| | SimPO $_{v0.2}$ | 15.5 | 18.5 | 29.3 | 40.1 | 45.3 | 36.23 |
| | Uni-DPO $_{v0.2}$ | **22.8** | **24.1** | **30.7** | **42.1** | **48.6** | **39.62** |
| Llama3-8B Instruct | SFT | 28.5 | 28.4 | 25.7 | 59.4 | 62.2 | 40.93 |
| | DPO | 43.7 | 41.7 | 32.6 | - | - | - |
| | SimPO | 44.7 | 40.4 | 33.8 | **61.2** | 65.8 | 44.81 |
| | Uni-DPO | **47.2** | **44.2** | **37.3** | **61.2** | **67.8** | **46.32** |
| | SimPO $_{v0.2}$ | 41.2 | 37.1 | 36.5 | 60.8 | 68.6 | 42.37 |
| | Uni-DPO $_{v0.2}$ | **46.8** | **42.6** | **40.6** | **66.9** | **73.2** | **46.50** |
| Gemma-2-9B-IT | SFT | 52.5 | 40.5 | 40.8 | - | - | - |
| | SimPO | 53.2 | 48.0 | 59.1 | 60.4 | 67.7 | **57.7** |
| | Uni-DPO | **54.7** | **52.7** | **67.1** | **71.2** | **72.8** | 57.5 |
| Qwen2.5-7B | SimPO | 20.9 | 18.4 | 39.5 | **43.6** | **48.2** | 48.2 |
| | Uni-DPO | **25.8** | **24.7** | **43.5** | 43.3 | 48.1 | **49.9** |

Table 2: **Main evaluation result of mathematical reasoning.** All benchmarks are evaluated with zero-shot chain-of-thought prompting and greedy decoding. The best results are highlighted in **bold**. The Uni-DPO method delivers significant improvements across all eight benchmarks for both 1.5B and 7B model scales.

| Model | Method | GSM8K | MATH | Minerva Math | Olympiad Bench | AIME 2024 | AMC 2023 | GaoKao 2023 En | College Math | Average |
|---|---|---|---|---|---|---|---|---|---|---|
| Qwen2.5-Math 1.5B | Baseline | 72.5 | 62.6 | 16.2 | 29.3 | 3.3 | 52.5 | 53.0 | 40.5 | 41.20 |
| | DPO | 72.7 | 64.4 | 18.8 | 31.4 | 3.3 | 47.5 | 56.6 | 41.9 | 42.08 |
| | SimPO | 76.0 | 68.4 | 25.4 | 32.9 | 13.3 | 47.5 | 55.3 | 46.6 | 45.68 |
| | Uni-DPO | **80.1** | **70.2** | **27.9** | **34.5** | **20.0** | **57.5** | **58.2** | **47.9** | **49.54** |
| Qwen2.5-Math 7B | Baseline | 64.3 | 65.8 | 10.7 | 24.3 | 23.3 | 47.5 | 44.7 | 32.3 | 39.11 |
| | DPO | 83.2 | 75.8 | 20.2 | 37.9 | **26.7** | 57.5 | 61.6 | 49.5 | 51.55 |
| | SimPO | 85.7 | 76.4 | 32.0 | 39.3 | **26.7** | 57.5 | 62.1 | 50.1 | 53.73 |
| | Uni-DPO | **88.9** | **78.2** | **34.6** | **41.5** | **26.7** | **67.5** | **65.7** | **51.3** | **56.80** |

**Experimental results.** As in Tab. 1, Uni-DPO consistently and significantly outperforms existing preference learning techniques across all settings. While both DPO and SimPO improve upon the SFT baseline, Uni-DPO achieves the highest overall scores, underscoring its robustness and effectiveness. Notably, Uni-DPO surpasses DPO and SimPO on both AlpacaEval2 and the Arena-Hard datasets across all models. Specifically, on the Arena-Hard dataset, Gemma-2-9B-IT finetuned with Uni-DPO attains a score of 67.1, outperforming the leading models Claude 3 Opus (2024-02-29) (Anthropic and others, 2024) (60.4) and Llama-3.1-70B-Instruct (AI@Meta, 2024a) (55.7)[3]. Details are in Sec. D.2.

**Scalability analysis.** As indicated by the v0.2 entry in Tab. 1, our method's performance steadily increases as the quality of training data improves, validating the effectiveness of our dynamic weighting mechanism in leveraging high-quality samples. Besides, as illustrated in Figs. 6d and 6e, Uni-DPO consistently outperforms SimPO across a spectrum of model sizes, revealing its scalability.

## 4.2 MATHEMATICAL REASONING EVALUATION

**Experimental settings.** To validate the effectiveness of Uni-DPO in mathematical reasoning, we employ the Qwen2.5 Math base model (Yang et al., 2024b) with 1.5B and 7B parameters. For training data, we select 20K math problems from the NuminaMath (LI et al., 2024) dataset. We use the same training data across all methods to compare fairly. The performance is assessed on eight widely used math reasoning benchmarks. All evaluations employ greedy decoding and zero-shot chain-of-thought prompting for consistency. More details are provided in Sec. D.1.

---

[3]Results are from the official Chatbot Arena Leaderboard.

Table 3: **Main ablation results.** Removing either the quality weight $w_{\text{qual}}$, the performance weight $w_{\text{perf}}$, length normalization (LN), or the calibrated NLL loss $\mathcal{L}_{\text{c-NLL}}$ consistently degrades performance compared to Uni-DPO, confirming that each component of Uni-DPO contributes meaningfully to its overall effectiveness.

| Model | Method | AlpacaEval2 | | Arena-Hard | IFEval | | SedarEval |
|---|---|---|---|---|---|---|---|
| | | LC(%) | WR(%) | WR(%) | Strict Acc.(%) | Loose Acc.(%) | Overall(%) |
| Llama3-8B Base | SFT | 5.7 | 3.7 | 3.3 | 33.5 | 35.3 | 20.85 |
| | Ours | **23.8** | **20.5** | **23.9** | 38.1 | **47.9** | 38.49 |
| | w/o $w_{\text{qual}}$ | 19.7 | 15.9 | 22.8 | 39.4 | 44.5 | 37.43 |
| | w/o $w_{\text{perf}}$ | 22.1 | 18.5 | 21.4 | 37.5 | 44.5 | **40.46** |
| | w/o LN | 5.2 | 3.8 | 2.7 | 36.8 | 39.6 | 28.18 |
| | w/o $\mathcal{L}_{\text{c-NLL}}$ | 22.4 | 19.4 | 23.3 | **40.7** | 47.5 | 37.73 |
| Llama3-8B Instruct | SFT | 28.5 | 28.4 | 25.7 | 59.4 | 62.2 | 40.93 |
| | Ours | **47.2** | **44.2** | 37.3 | 61.2 | **67.8** | 46.32 |
| | w/o $w_{\text{qual}}$ | 44.7 | 41.0 | 36.1 | **61.9** | 66.7 | 44.94 |
| | w/o $w_{\text{perf}}$ | 44.2 | 40.5 | 37.0 | 57.5 | 66.0 | 43.76 |
| | w/o LN | 47.0 | 43.7 | **38.4** | 59.5 | 66.5 | **46.60** |
| | w/o $\mathcal{L}_{\text{c-NLL}}$ | 46.2 | 43.2 | 37.6 | 58.6 | 65.4 | 44.48 |

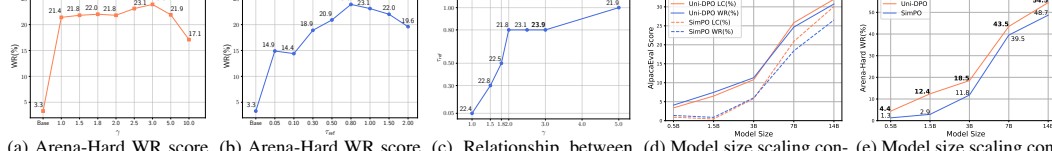

(a) Arena-Hard WR score when $\tau_{\text{ref}} = 0.8$, varying $\gamma$  (b) Arena-Hard WR score when $\gamma = 3.0$, varying $\tau_{\text{ref}}$  (c) Relationship between $w_{\text{perf}}$ parameter $\gamma$ and $\tau_{\text{ref}}$  (d) Model size scaling consistency in AlpacaEval  (e) Model size scaling consistency in Arena-Hard

Figure 6: **Parameter sensitivity and scaling consistency.** (a)–(c) Sensitivity of the performance weight $w_{\text{perf}}$ (Eq. (10)) hyperparameters $\gamma$ and $\tau_{\text{ref}}$: (a),(b) show practical ranges of $[1.0, 5.0]$ for $\gamma$ and $[0.5, 2.0]$ for $\tau_{\text{ref}}$; (c) plots the optimal $\tau_{\text{ref}}$ and corresponding Arena-Hard WR scores across different $\gamma$ values, revealing that higher $\gamma$ indicates sharper gradient decrease on easy sample, requires stronger margin constraints $\tau_{\text{ref}}$. (d),(e) Uni-DPO consistently outperforms SimPO on Qwen2.5 base model across different sizes, revealing its scalability.

**Experimental results.** As shown in Tab. 2, our proposed Uni-DPO method yields significant improvements over both SimPO and DPO across model scales of 1.5B and 7B parameters. Specifically, Uni-DPO-trained models consistently outperform the baselines on all eight mathematical reasoning benchmarks, providing strong empirical evidence for the effectiveness and robustness of our approach. Moreover, the improvement margin of Uni-DPO grows with model scale. Uni-DPO achieves an average gain of 8.3 points on the 1.5B model, which further increases to 17.7 points on the 7B model. This observation indicates that our approach scales effectively with model capacity, exhibiting stronger advantages as the model size increases.

## 4.3 ABLATION STUDY

In this section, we conduct an ablation study to evaluate the individual contributions of each component in our Uni-DPO framework. As shown in Tab. 3, models without $w_{\text{qual}}$ or $w_{\text{perf}}$ exhibit significant performance drops, indicating the critical roles of these weighting factors in the optimization process. The base model without length normalization shows a notable decline, suggesting that the base model is more unstable and sensitive to data length. Finally, models without $\mathcal{L}_{\text{c-NLL}}$ perform poorly across all benchmarks, highlighting the essential role of c-NLL loss during training. Further ablations on the components of $\mathcal{L}_{\text{c-NLL}}$ and the selection of the expert model are discussed in Sec. D.3.

## 4.4 PARAMETER SENSITIVITY

In our textual experiments, most hyperparameters are fixed across all models and benchmarks, $w_{\text{qual}} = 0.7$, c-NLL coefficient $\lambda = 0.001$, and $\tau_{\text{good}} = 3.2$, yielding stable and strong performance, which demonstrates the robustness of Uni-DPO. Only the performance-weight $w_{\text{perf}}$ parameters vary: sensitivity analysis in Figs. 6a to 6c shows the practical ranges of these parameters are $\gamma \in [1.0, 5.0]$ and $\tau_{\text{ref}} \in [0.5, 2.0]$, leading us to use $\gamma = 3.0$ throughout all experiments. Detailed training settings and environments are provided in Sec. C.1.

# 5 CONCLUDING REMARKS

**Further discussions.** Within the *Unified Paradigm* proposed by Shao et al. (2024b), all methods, such as SFT, DPO, PPO, and GRPO, can be conceptualized as RL methods. Therefore, we re-examine the underlying principles of Uni-DPO from the RL perspective. In algorithms such as PPO and GRPO, the gradient coefficient is derived from the advantage function, which is computed based on a reward model and (where applicable) a value model, enabling them to dynamically reinforce or penalize responses with varying intensity. By contrast, DPO relies solely on its intrinsic reward margin (Eq. (3)) as the weighting signal, without incorporating a fine-grained external reward signal. As a result, it cannot dynamically weight data samples according to their quality. Additionally, during training, DPO is limited in its ability to effectively distinguish between samples of varying difficulty. Uni-DPO closes this gap by incorporating both a data quality weight $w_{\text{qual}}$ and a performance weight $w_{\text{perf}}$, thus explicitly enabling the model to adaptively prioritize samples based on both their intrinsic quality and training difficulty in a manner analogous to advantage-based reweighting. This dual-perspective weighting mechanism allows Uni-DPO to achieve more fine-grained credit assignment during policy optimization, addressing the limitations of DPO. More discussions are provided in Sec. E.

**Summary.** In this work, we introduce Uni-DPO, a novel unified optimization framework for dynamic preference learning of LLMs. Our approach incorporates a dual-perspective weighting mechanism that dynamically adjusts each sample's contribution during training based on the intrinsic data quality and the model's learning dynamics, enabling the model to focus on high-quality yet challenging examples and thereby improving training efficiency and overall performance.

**Limitation and future work.** While our method demonstrates significant performance improvements compared to existing methods, the quality of the training data remains a critical factor. Future work could explore more sophisticated methods for estimating data quality for preference alignment.

**Reproducibility statement.** To facilitate a clearer understanding of our contributions and to enable broader use and adoption of our work, we provide extensive materials to support reproducibility. In the main text, we describe the key components of our method in Sec. 3 and present the main experimental results in Sec. 4. In the supplementary materials, we provide further detailed information, including Motivation Details (Sec. A), Method Details (Sec. B), Training Details (Sec. C), and Evaluation Details (Sec. D), which we hope will help reproduce our results. Furthermore, we release our code, data, training scripts, and model checkpoints to facilitate future research and applications.

**Acknowledgement.** This work was supported by the Guangdong Basic and Applied Basic Research Foundation (2025A1515011546), and by the Shenzhen Science and Technology Program (JCYJ20240813105901003, ZDCY20250901113000001).

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

# Uni-DPO: A Unified Paradigm for
# Dynamic Preference Optimization of LLMs
## Supplementary Material

### OVERVIEW

This material provides supplementary details to the main paper, including the following sections:

- (A) **Motivation Details**
  - (A.1) Analysis of Reward vs. Score Margins
  - (A.2) Overfitting Behavior of Quality Weight
  - (A.3) Stability of Performance Weighting
- (B) **Method Details**
  - (B.1) DPO with Length Normalization
  - (B.2) Performance Weighting Factor
  - (B.3) Overall Optimization Objective
- (C) **Training Details**
  - (C.1) Training Setup
  - (C.2) Training Data
  - (C.3) Training Pseudocode
  - (C.4) Training Complexity
  - (C.5) Training Dynamics
- (D) **Evaluation Details**
  - (D.1) Evaluation Setup
  - (D.2) Evaluation Results
  - (D.3) Further Ablation Study
  - (D.4) Downstream Task Evaluation
- (E) **Discussions**
  - (E.1) Revisiting Uni-DPO through the RL Paradigm
  - (E.2) Impact of Length Normalization on Llama3-8B-Base
  - (E.3) Further Discussions on c-NLL Loss
  - (E.4) Rationale for Relaxing the Original Focal Weight
- (F) **Related Work**
- (G) **Case Studies**
- (H) **Broader Impact**
- (I) **LLM Usage Statement**

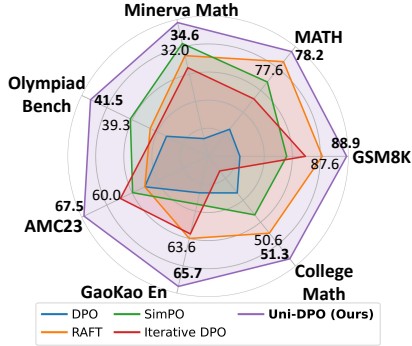

(a) Results on mathematical reasoning tasks (7B)

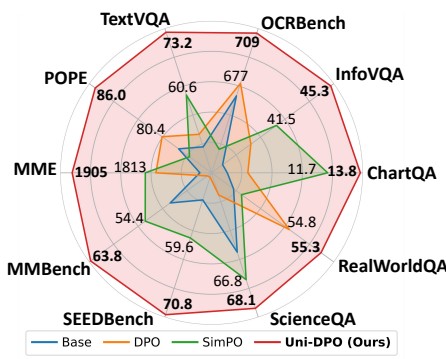

(b) Results on multimodal tasks

## A  MOTIVATION DETAILS

In this section, we deepen the discussion supporting the key observations from the main paper Sec. 2.2. We focus on three critical phenomena: (1) the empirical relationship between reward margins and expert-assigned score margins of training pairs, as illustrated in the main paper Fig. 4, (2) the overfitting dynamics that emerge during preference learning if only consider the data quality, shown in the main paper Fig. 5c, and (3) the stability of performance weighting factors, as shown in the main paper Fig. 5b. These analyses help to shed light on the challenges inherent in preference learning and suggest the need for a more flexible approach, motivating our design of the Uni-DPO framework.

### A.1  ANALYSIS OF REWARD VS. SCORE MARGINS

To investigate how the data quality of preference samples relates to the model's learning dynamics during training, we fine-tune Qwen2.5-Math base model on a 50/50 split of the preference data, using half of the dataset for training and reserving the remainder for validation, to emulate exposure to unseen pairs during training. As illustrated in main paper Fig. 4, the results reveal that a pair's intrinsic quality (measured by its expert-assigned score margin $(S_w - S_l)$) does not necessarily correlate to the model's learning difficulty, as quantified by its reward margin $\Delta_r$ (defined in the main paper Eq. (3)). For instance, a pair with high quality (*i.e.,*a large score margin) may be already well-fitted by the model. This observation prompts us to consider whether relying solely on intrinsic data quality in preference training is sufficient and to examine the potential negative consequences further.

In addition, we conducted a parallel analysis for textual understanding. We fine-tune Qwen2.5-7B on the *train_prefs* split of the UltraFeedback (Cui et al., 2023) dataset and then evaluate it on the held-out *test_prefs* split. The resulting Spearman correlation between expert score margin $(S_w - S_l)$ and model reward margin $\Delta_r$ was only $\rho = 0.08$ (using the same metrics as in the main paper Fig. 4), indicating almost no correlation between intrinsic data quality and learning difficulty. This result further reinforces the need for weighting mechanisms that go beyond only considering data quality.

### A.2  OVERFITTING BEHAVIOR OF QUALITY WEIGHT

**DPO with $w_{\textbf{qual}}$ only.**   We perform an investigation of overfitting in preference learning. Specifically, during fine-tuning of Qwen2.5-7B (Yang et al., 2024a) on the *train_prefs* split of the UltraFeedback (Cui et al., 2023) dataset, we extend the standard DPO objective (the main paper Eq. (1)) by incorporating a data quality weighting factor $w_{\text{qual}}$ (the main paper Eq. (5)). We tracked both the training loss and the evaluation loss measured on the held-out *test_prefs* split. The main paper Fig. 5c shows that as training progresses, the training loss steadily decreases, whereas the evaluation loss, after initially falling, reaches a minimum (marked by the dashed vertical line) and then begins to rise. This divergence signals that the model starts to overfit to the training data beyond that point. This suggests that effective training strategies must also adapt to the model's learning dynamics in addition to the data quality.

**DPO with $w_{\textbf{qual}}$ and $w_{\textbf{perf}}$.**   After we also incorporate the performance weighting factor $w_{\text{perf}}$ (the main paper Eq. (10)) into the DPO objective, we observe that both the training and evaluation losses exhibit a consistent downward trend throughout the training process, as shown in the main paper Fig. 5d. This suggests that our proposed $w_{\text{perf}}$ effectively mitigates overfitting, allowing the model to better leverage the data pairs during training and better align with human preferences.

### A.3  STABILITY OF PERFORMANCE WEIGHTING

Finally, we compare the gradient norms of the original focal-weighted DPO (the main paper Eq. (7)) and our stabilized variant (the main paper Eq. (10)) during fine-tuning Llama3-8B-Base (AI@Meta, 2024b) on the UltraFeedback (Cui et al., 2023) dataset. The results show that the original focal-weighted DPO exhibits frequent, large spikes in gradient norm throughout training, whereas our stabilized variant maintains a much smoother profile. This finding suggests that naively applying focal loss (Lin et al., 2017) weights in preference learning can induce unstable updates, while our stabilized form, which utilizes length normalization (LN) and a relaxed, unified constraint $\tau_{\text{ref}}$, yields consistently more stable training dynamics.

## B    METHOD DETAILS

In this section, we provide a detailed exposition of the key components of our proposed Uni-DPO framework, including its formulations and gradient derivations. We begin with a review of the fundamental principles of DPO, then introduce the length normalization (LN) in Sec. B.1, next present the performance weighting strategies in Sec. B.2, and finally conclude with the derivation of the overall objective function in Sec. B.3.

### B.1    DPO WITH LENGTH NORMALIZATION

The DPO objective (Rafailov et al., 2023) augmented with length normalization (LN) demonstrates significantly more stable training dynamics in our experiments. In the following, we briefly introduce DPO with length normalization to prepare for the subsequent sections. The gradient of the DPO loss (the main paper Eq. (1)) with respect to the policy model $\pi_\theta$ parameters $\theta$ can be written as:

$$\nabla_\theta \mathcal{L}_{\text{DPO}} =$$
$$- \beta \mathbb{E}_{(x,y_w,y_l)\sim D}\Big[ \underbrace{\sigma(r(x,y_l) - r(x,y_w))}_{\text{higher when reward estimate is wrong}} \Big( \underbrace{\nabla_\theta \log \pi_\theta(y_w|x)}_{\text{increase likelihood of } y_w} - \underbrace{\nabla_\theta \log \pi_\theta(y_l|x)}_{\text{decrease likelihood of } y_l} \Big)\Big], \quad (12)$$

where the implicit reward function (excluding the partition function $Z(x)$ for simplicity) is:

$$r(x,y) = \beta \log \frac{\pi_\theta(y \mid x)}{\pi_{\text{ref}}(y \mid x)}. \quad (13)$$

The gradient of DPO with length normalization (DPO w/ LN) is:

$$\nabla_\theta \mathcal{L}_{\text{DPO w/ LN}} =$$
$$- \beta \mathbb{E}_{(x,y_w,y_l)\sim D}\Big[\sigma\big(r_{\text{LN}}(x,y_l) - r_{\text{LN}}(x,y_w)\big)\Big(\nabla_\theta \frac{\log \pi_\theta(y_w|x)}{|y_w|} - \nabla_\theta \frac{\log \pi_\theta(y_l|x)}{|y_l|}\Big)\Big], \quad (14)$$

where the reward function with length normalization ($r_{\text{LN}}$) is defined as follows:

$$r_{\text{LN}}(x,y) = \frac{\beta}{|y|} \log \frac{\pi_\theta(y \mid x)}{\pi_{\text{ref}}(y \mid x)}. \quad (15)$$

Recent studies (Meng et al., 2024; Park et al., 2024; Lu et al., 2024) have observed that DPO training allows the policy to exploit a length bias in the data, assigning disproportionately high probabilities to longer sequences. As a result, DPO-trained models tend to produce unduly long responses, even though the quality of a response should remain independent of its token length (Dubois et al., 2024; Singhal et al., 2023; Durmus et al., 2022; Kabir et al., 2024). Thus, length normalization (LN) has been introduced in the framework of DPO by recent studies (Meng et al., 2024; Lu et al., 2024).

### B.2    PERFORMANCE WEIGHTING FACTOR

Building on our previous discussion of length normalization, in this section, we further discuss the performance-weighting factor. We first derive the gradient for the original DPO objective augmented with vanilla focal loss (Lin et al., 2017) weighting factor $w_{\text{focal}}$ (the main paper Eq. (8)), and then transition to the gradient derivation for our more stable variant $w_{\text{perf}}$ (the main paper Eq. (10)). This analysis prepares the groundwork for deriving the gradient of our overall Uni-DPO optimization objective in Sec. B.3.

**The direct integration with DPO.**    We first derive the gradient of the focal-weighted DPO loss $\mathcal{L}_{\text{DPO w/ FL}}$, arriving at the expression in the main paper Eq. (9). Recall that $\mathcal{L}_{\text{DPO w/ FL}}$ is defined as:

$$\mathcal{L}_{\text{DPO w/ FL}} = -\mathbb{E}_{(x,y_w,y_l)\sim D}\big[w_{\text{focal}}(\pi_\theta)\cdot\log\sigma(\Delta_r)\big], \quad (16)$$

where the implicit reward margin $\Delta_r$ and the weighting factor $w_{\text{focal}}$ are defined as:

$$\Delta_r = r(x,y_w) - r(x,y_l) = \beta \log \frac{\pi_\theta(y_w \mid x)}{\pi_{\text{ref}}(y_w \mid x)} - \beta \log \frac{\pi_\theta(y_l \mid x)}{\pi_{\text{ref}}(y_l \mid x)},$$

$$w_{\text{focal}} = \left[1 - \sigma\left(\beta \log \frac{\pi_\theta(y_w|x)}{\pi_{\text{ref}}(y_w|x)} - \beta \log \frac{\pi_\theta(y_l|x)}{\pi_{\text{ref}}(y_l|x)}\right)\right]^\gamma = [1 - \sigma(\Delta_r)]^\gamma. \quad (17)$$

The gradient of this loss concerning the policy model $\pi_\theta$ parameters $\theta$ proceeds as follows:

$$
\begin{aligned}
\nabla_\theta \mathcal{L}_{\text{DPO w/ FL}} &= -\nabla_\theta \mathbb{E}\big[(1 - \sigma(\Delta_r))^\gamma \cdot \ln \sigma(\Delta_r)\big] \\
&= -\mathbb{E}\Big[(1 - \sigma(\Delta_r))^\gamma \cdot \nabla_\theta \ln \sigma(\Delta_r) + \nabla_\theta (1 - \sigma(\Delta_r))^\gamma \cdot \ln \sigma(\Delta_r)\Big] \\
&= -\mathbb{E}\Big[(1 - \sigma(\Delta_r))^\gamma \cdot \frac{\nabla_\theta \sigma(\Delta_r)}{\sigma(\Delta_r)} + \gamma(1 - \sigma(\Delta_r))^{\gamma-1} \cdot \nabla_\theta\big(-\sigma(\Delta_r)\big) \cdot \ln \sigma(\Delta_r)\Big] \\
&= -\mathbb{E}\Big[(1 - \sigma(\Delta_r))^\gamma (1 - \sigma(\Delta_r)) - \gamma \ln \sigma(\Delta_r)(1 - \sigma(\Delta_r))^{(\gamma-1)+1} \sigma(\Delta_r)\Big] \nabla_\theta(\Delta_r) \\
&= -\mathbb{E}\Big[(1 - \sigma(\Delta_r))^\gamma \cdot \big(1 - \sigma(\Delta_r) - \gamma \sigma(\Delta_r) \ln \sigma(\Delta_r)\big)\Big] \nabla_\theta(\Delta_r)\,,
\end{aligned}
$$

where $\nabla_\theta(\Delta_r) = \beta\Big[\nabla_\theta \log \pi_\theta(y_w|x) - \nabla_\theta \log \pi_\theta(y_l|x)\Big]\,.$

(18)

As shown in the main paper Fig. 5a, the gradient of the focal-weighted DPO objective peaks when the model performs poorly and diminishes as performance improves. This dynamic naturally prioritizes harder examples during training by downweighting the contribution of already well-fitted samples.

**Calibrated weighting factor.** Since directly incorporating focal-loss weights into preference learning can destabilize training (as described in the main paper Sec. 3.3), we incorporate length normalization (LN) together with a unified performance margin $\tau_{\text{ref}}$, thus arriving at our more stable performance weight $w_{\text{perf}}$ as defined in Eq. (10) of the main paper. In the sequel, we derive the gradient of this stabilized form. The stabilized DPO loss with performance weighting is defined as:

$$
\mathcal{L}_{\text{DPO w/ perf}} = -\mathbb{E}_{(x,y_w,y_l)\sim D}\big[w_{\text{perf}}(\pi_\theta) \cdot \log \sigma(\Delta_r)\big]\,,
\tag{19}
$$

where the implicit reward margin $\Delta_r$ and the performance weight $w_{\text{perf}}$ are defined as:

$$
\begin{aligned}
\Delta_r &= r_{\text{LN}}(x, y_w) - r_{\text{LN}}(x, y_l) = \frac{\beta}{|y_w|} \log \frac{\pi_\theta(y_w \mid x)}{\pi_{\text{ref}}(y_w \mid x)} - \frac{\beta}{|y_l|} \log \frac{\pi_\theta(y_l \mid x)}{\pi_{\text{ref}}(y_l \mid x)}\,, \\
w_{\text{perf}} &= \left[1 - \sigma\left(\frac{\beta}{|y_w|} \log \pi_\theta(y_w|x) - \frac{\beta}{|y_l|} \log \pi_\theta(y_l|x) - \tau_{\text{ref}}\right)\right]^\gamma.
\end{aligned}
\tag{20}
$$

First, we introduce the *adjusted reward margins* $\Delta_{\text{adj}}$ for simplicity, which is defined as:

$$
\Delta_{\text{adj}} = \frac{\beta}{|y_w|} \log \pi_\theta(y_w \mid x) - \frac{\beta}{|y_l|} \log \pi_\theta(y_l \mid x) - \tau_{\text{ref}}\,.
\tag{21}
$$

Thus, we have:

$$
\nabla_\theta \Delta_{\text{adj}} = \nabla_\theta \Delta_r = \frac{\beta}{|y_w|} \nabla_\theta \log \pi_\theta(y_w \mid x) - \frac{\beta}{|y_l|} \nabla_\theta \log \pi_\theta(y_l \mid x).
\tag{22}
$$

For brevity, denote the performance weight:

$$
w_{\text{perf}} = \big(1 - \sigma(\Delta_{\text{adj}})\big)^\gamma.
\tag{23}
$$

We also require the following intermediate gradients:

$$
\begin{aligned}
\nabla_\theta \log \sigma(\Delta_r) &= \frac{1}{\sigma(\Delta_r)} \sigma(\Delta_r)\big(1 - \sigma(\Delta_r)\big) \nabla_\theta \Delta_r = \big(1 - \sigma(\Delta_r)\big) \nabla_\theta \Delta_r\,, \\
\nabla_\theta \sigma(\Delta_{\text{adj}}) &= \sigma(\Delta_{\text{adj}})\big(1 - \sigma(\Delta_{\text{adj}})\big) \nabla_\theta \Delta_{\text{adj}}.
\end{aligned}
\tag{24}
$$

It follows that:

$$
\begin{aligned}
\nabla_\theta w_{\text{perf}} &= \nabla_\theta \Big[\big(1 - \sigma(\Delta_{\text{adj}})\big)^\gamma\Big] \\
&= -\gamma \big(1 - \sigma(\Delta_{\text{adj}})\big)^{\gamma-1} \sigma(\Delta_{\text{adj}}) \big(1 - \sigma(\Delta_{\text{adj}})\big) \nabla_\theta \Delta_{\text{adj}} \\
&= -\gamma \big(1 - \sigma(\Delta_{\text{adj}})\big)^\gamma \sigma(\Delta_{\text{adj}}) \nabla_\theta \Delta_{\text{adj}}.
\end{aligned}
\tag{25}
$$

Consequently, the overall gradient of the loss can be expressed as:

$\nabla_\theta \mathcal{L}_{\text{DPO w/ perf}}$

$$
\begin{aligned}
&= -\mathbb{E}\Big[\nabla_\theta\big(w_{\text{perf}}\log\sigma(\Delta_r)\big)\Big]\\
&= -\mathbb{E}\Big[w_{\text{perf}}(\nabla_\theta\log\sigma(\Delta_r)) + (\nabla_\theta w_{\text{perf}})\log\sigma(\Delta_r)\Big]\\
&= -\mathbb{E}\Big[(1-\sigma(\Delta_{\text{adj}}))^\gamma(1-\sigma(\Delta_r))\nabla_\theta\Delta_r - \gamma(1-\sigma(\Delta_{\text{adj}}))^\gamma\sigma(\Delta_{\text{adj}})\log\sigma(\Delta_r)\nabla_\theta\Delta_{\text{adj}}\Big]\\
&= -\mathbb{E}\Big[(1-\sigma(\Delta_{\text{adj}}))^\gamma\big((1-\sigma(\Delta_r))\nabla_\theta\Delta_r - \gamma\sigma(\Delta_{\text{adj}})\log\sigma(\Delta_r)\nabla_\theta\Delta_{\text{adj}}\big)\Big]\\
&= -\mathbb{E}\Big[(1-\sigma(\Delta_{\text{adj}}))^\gamma\big((1-\sigma(\Delta_r)) - \gamma\sigma(\Delta_{\text{adj}})\log\sigma(\Delta_r)\big)\cdot\nabla_\theta\Delta_r\Big].
\end{aligned}
\tag{26}
$$

Both the gradient of the original focal-weighted DPO $\mathcal{L}_{\text{DPO w/ FL}}$ (Eq. (16)) and its stabilized performance-weighted counterpart $\mathcal{L}_{\text{DPO w/ perf}}$ (Eq. (19)) share a common structural backbone: they modulate the base DPO gradient $\nabla_\theta\Delta_r$ by a multiplicative "focusing" term of the form $(1-\sigma(\cdot))^\gamma\big[(1-\sigma(\Delta_r)) - \gamma\sigma(\cdot)\ln\sigma(\Delta_r)\big]$.

This factor down-weights examples for which the model is already confident (*i.e.,*$\sigma(\Delta_r)$ is high) and up-weights those where it underperforms (*i.e.,*$\sigma(\Delta_r)$ is low). In both the original focal-weighted DPO loss $\mathcal{L}_{\text{DPO w/ FL}}$ and our stabilized performance-weighted variant $\mathcal{L}_{\text{DPO w/ perf}}$, the exponent $\gamma$ determines how sharply the model concentrates on harder samples. Both losses share the same $(1-\sigma(\cdot))^\gamma$ focal multiplier, preserving a targeted emphasis on difficult cases.

With this formula established, we now incorporate it into the overall objective function of Uni-DPO.

### B.3  OVERALL OPTIMIZATION OBJECTIVE

Recall that the overall objective of Uni-DPO is defined as:

$$
\mathcal{L}_{\text{Uni-DPO}} = -\mathbb{E}_{(x,y_w,y_l)\sim D}\Big[w_{\text{qual}}(y_w,y_l)\cdot w_{\text{perf}}(\pi_\theta)\cdot\log\sigma(\Delta_r)\Big] + \lambda\mathcal{L}_{\text{c-NLL}}.
\tag{27}
$$

Based on the derivation in Eq. (26), the gradient of Uni-DPO can be expressed as:

$\nabla_\theta\mathcal{L}_{\text{Uni-DPO}}$

$$
\begin{aligned}
&= -\mathbb{E}_{(x,y_w,y_l)\sim D}\Big[w_{\text{qual}}(1-\sigma(\Delta_{\text{adj}}))^\gamma\big((1-\sigma(\Delta_r)) - \gamma\sigma(\Delta_{\text{adj}})\log\sigma(\Delta_r)\big)\nabla_\theta\Delta_r\\
&\quad + \lambda\cdot\mathbf{1}\big(\log\pi_{\text{ref}}(y_w\mid x) > \log\pi_\theta(y_w\mid x)\big)\mathbf{1}\big(S_w\geq\tau_{\text{good}}\big)\cdot\nabla_\theta\frac{\log\pi_\theta(y_w\mid x)}{|y_w|}\Big]\\
&= -\mathbb{E}\Big\{\Big[\beta w_{\text{qual}}(1-\sigma(\Delta_{\text{adj}}))^\gamma\big((1-\sigma(\Delta_r))-\gamma\sigma(\Delta_{\text{adj}})\log\sigma(\Delta_r)\big)+\lambda\cdot I\Big]\nabla_\theta\frac{\log\pi_\theta(y_w\mid x)}{|y_w|}\\
&\quad + \Big[\beta w_{\text{qual}}(1-\sigma(\Delta_{\text{adj}}))^\gamma\big((1-\sigma(\Delta_r))-\gamma\sigma(\Delta_{\text{adj}})\log\sigma(\Delta_r)\big)\Big]\nabla_\theta\frac{\log\pi_\theta(y_l\mid x)}{|y_l|}\Big\}.
\end{aligned}
\tag{28}
$$

where $I$ denotes the indicator function $I = \mathbf{1}\big(\log\pi_{\text{ref}}(y_w\mid x) > \log\pi_\theta(y_w\mid x)\big)\mathbf{1}\big(S_w\geq\tau_{\text{good}}\big)$, $w_{\text{qual}}$ is the quality weighting factor, and $S_w$ is the score of the preferred completion $y_w$. The $\Delta_r$ is the reward margin in Eq. (20) and $\Delta_{\text{adj}}$ is the adjusted reward margin in Eq. (21).

Upon deriving the gradient of the overall objective $\mathcal{L}_{\text{Uni-DPO}}$, we proceed with the discussion from the main paper (in main paper Sec. 5) by analyzing Uni-DPO's gradient coefficient $GC_\mathcal{A}$ within the *Unified Paradigm* proposed by Shao et al. (2024b) and comparing it to that of DPO in Sec. E.1.

## C  TRAINING DETAILS

In this section, we present comprehensive details related to training, including the training setup (Sec. C.1), the training datasets (Sec. C.2), the training pseudocode (Sec. C.3), the training complexity (Sec. C.4), and the training dynamics (Sec. C.5). By making these details available, we aim to enhance reproducibility and further validate the effectiveness of Uni-DPO.

Table C.1: **Training setup for textual understanding evaluation.** We maintain largely consistent hyperparameter configurations across all model variants and datasets to demonstrate the robustness and scalability of Uni-DPO. In this table, the placeholder $x$ in Qwen2.5-$x$B denotes the model size in billions of parameters, with $x \in \{0.5, 1.5, 3, 7, 14\}$. We set the threshold $\tau_{\text{good}} = 3.2$ since this value corresponds to the median score of the chosen response $y_w$ in the training dataset, thereby reflecting the central tendency of the score distribution.

| Model
Setting | Llama3-8B-Base | Llama3-8B-Base v0.2 | Llama3-8B-Instruct | Llama3-8B-Instruct v0.2 | Gemma-2-9B-IT | Qwen2.5-$x$B |
|---|---|---|---|---|---|---|
| Learning rate | 1.0e-6 | | | | 8.0e-7 | 2.0e-6 |
| DPO beta $\beta$ | 2.0 | | 10.0 | | | 3.5 |
| $w_{\text{perf}}$ param $\tau_{\text{ref}}$ | 0.8 | | 1.0 | | 0.8 | |
| $w_{\text{qual}}$ param $\eta$
$w_{\text{perf}}$ param $\gamma$
c-NLL loss lambda $\lambda$
Data threshold $\tau_{\text{good}}$ | 0.7
3
0.001
3.2 | | | | | |
| Mix precision training
Learning rate scheduler
Global batch size
Warmup ratio
Optimizer
Epoch
Seed | Bfloat16
Cosine
128
0.1 (10% steps)
AdamW (Loshchilov & Hutter, 2017)
1
42 | | | | | |

Table C.2: **Prompt template for math reasoning ability training and evaluation.** All evaluations employ greedy decoding and widely used zero-shot chain-of-thought (CoT) (Wei et al., 2022) prompting for consistency.

```
<|im_start|>system
Please reason step by step, and put your final answer within \boxed{ }.<|im_end|>
<|im_start|>user
{Question} Let's think step by step and output the final answer within \boxed{ }<|im_end|>
<|im_start|>assistant
```

## C.1 TRAINING SETUP

**General training hyperparameters.** All general training settings, including the batch size, number of epochs, learning rate schedule, optimizer choice, and so on, are identical to those used in SimPO (Meng et al., 2024), ensuring a fair and controlled comparison.

**Method-specific training hyperparameters.** For the training hyperparameters of DPO (Rafailov et al., 2023) and SimPO (Meng et al., 2024), we follow the officially recommended best practices and perform a grid search to select configurations that achieve the best performance. For Uni-DPO, we observed that the choice of hyperparameters can vary between models and affect the benchmark outcomes. To address this, we run preliminary experiments to identify appropriate values for the performance weight $w_{\text{perf}}$ hyperparameters $\gamma$ and $\tau_{\text{ref}}$, as illustrated in the main paper Figs. 6a and 6b, the practical range of $\gamma$ is $[1.0, 5.0]$ and that of $\tau_{\text{ref}}$ is $[0.5, 2.0]$. Once these ranges were established, we **maintained largely consistent hyperparameter configurations across all models, datasets, and benchmarks** to demonstrate the robustness and scalability of Uni-DPO. Detailed training configurations for textual understanding tasks are given in Tab. C.1, and for mathematical reasoning tasks and multimodal tasks are provided in Tab. C.3.

**Computation environment.** Our experiments were mainly conducted based on the code from the SimPO repository to ensure a fair comparison. All computations are performed on 8xH800 GPUs.

## C.2 TRAINING DATA

**Textual training data.** All training configurations, including the training data and training pipeline, are kept strictly aligned with those used in SimPO (Meng et al., 2024) to ensure a fair comparison. For the *v0.1* version of the data, the quality score of each preference pair is directly provided by the publicly available UltraFeedback dataset. Unlike DPO and SimPO, which can only exploit paired preference signals, Uni-DPO leverages these supervision signals in a more fine-grained manner and performs adaptive correction throughout the preference learning process.

For the *v0.2* version of the training data, which utilizes preference labels from a more powerful reward model, we obtain the quality scores of data using two models, GPT-4o (Hurst et al., 2024) and ArmoRM (Wang et al., 2024c). For GPT-4o, we adopt the widely used 'single-v1' judge prompt from

FastChat (Zheng et al., 2023b) to call the model, which first asks the model to explain its judgment in terms of the helpfulness, relevance, accuracy, depth, creativity, and level of detail of the response, and then produces an overall quality score. The range of the quality score is from 1 to 10. For ArmoRM, the reward model evaluates responses along dimensions such as helpfulness, correctness, coherence, complexity, and verbosity, and we use the official scoring script released with ArmoRM to obtain the final scores.[4] Except for the way quality scores are obtained, the experimental settings for v0.2 are identical to those of v0.1 in all other aspects.

**Math training data.** For mathematical reasoning ability training data, following recent studies (Mahdavi et al., 2025; Face, 2025; Xiang et al., 2025; Zhang et al., 2025a), we randomly sampled 20K math problems from the widely used NuminaMath (LI et al., 2024) dataset. We generate preference training data in an *on-policy* manner: using the base model $\pi_\theta$ under training, we sample $N=8$ candidate solutions $y \sim \pi_\theta(Y|x)$ for each problem prompt $x$ using the zero-shot chain of thought (CoT) prompting template shown in Tab. C.2 with sampling temperature of 0.8 and top-p of 0.8. We then apply a rule-based classifier to label each sample as *correct* or *incorrect* using the provided ground truth answer.[5] Furthermore, we score each response $y$ using an open-source domain-specific reward model, Qwen2.5-Math-PRM-7B (Zhang et al., 2025c). [6][7] To maintain the training data quality, we exclude problems $x$ for which all sampled responses $y \sim \pi_\theta(Y|x)$ are labeled as *incorrect*, as these cases prove excessively challenging for the model to learn. From the remaining pool, we select only those positive samples whose solutions are correct and whose reward scores $S_w$ exceed those of corresponding negative samples (*i.e.*, $S_w > S_l$), thereby ensuring a dual level of data quality control. This process yields a final set of 20K high-quality math reasoning training examples, which we use uniformly across DPO, SimPO, and Uni-DPO training objectives to ensure a fair comparison.

**Multimodal training data.** For multimodal training, we employ the recently publicly available MM-RLHF (Zhang et al., 2025b) dataset, which is a high-quality human-annotated preference dataset with reliable ranking rationales. We use the *Image* subset, which includes three question types: *Long* (long-text), *Short* (short-text), and *MCQ* (multiple-choice), totaling approximately 50K preference learning pairs. We then utilize the critique-based MLLM reward model, MM-RLHF-Reward-7B-llava-ov-qwen (Zhang et al., 2025b), to score the sampled responses using the human-provided critiques, to ensure both quality and efficiency.[8] We apply this same set of around 50K multimodal examples across DPO, SimPO, and Uni-DPO objectives to maintain a fair comparison.

### C.3 TRAINING PSEUDOCODE

The pseudocode for computing the training loss in Uni-DPO is presented in Algorithm 1. In brief, the procedure (1) performs forward passes under both the policy $\pi_\theta$ and reference models $\pi_{\text{ref}}$ to obtain log-probability ratios; (2) computes dual adaptive weights; (3) applies these weights to the DPO loss; (4) introduces the calibrated NLL loss for high-quality, hard positive samples; and (5) returns the batch-mean of all loss components.

### C.4 TRAINING COMPLEXITY

In this section, we provide a comparison of the training complexity between DPO and our Uni-DPO.

Both DPO and Uni-DPO share the same computational bottleneck: each training sample requires four LLM forward passes (two from the policy model and two from the reference model), typically in the range of trillions of FLOPs per sample.

Any extra operations in Uni-DPO are negligible. The only additional costs come from computing the dual weights and the c-NLL term, which are simple element-wise arithmetic (differences, sigmoid,

---

[4]Since ArmoRM is a regression model, its output scores are not constrained to a strict range. Therefore, after scoring all entries in the dataset, we apply Z-score normalization to standardize the scores, and then use the CDF of the standard normal distribution to map them into the range 0–10, to match the v0.1 setting.

[5]Our evaluation code for math answer correction is adapted from simpleRL-reason (Zeng et al., 2025).

[6]We use the *solution reformatting* method suggested by Zheng et al. (2024) to split the multi-step responses into individual steps for PRM to process.

[7]The prompt we use to call the math PRM is suggested by the official implementations.

[8]The prompt we use to call the multimodal reward model is suggested by the official implementations.

Table C.3: **Training setup for math reasoning and multimodal tasks.** We maintained largely consistent hyperparameter configurations across all models to demonstrate the robustness and scalability of Uni-DPO.

| Model Setting | Qwen2.5-Math-1.5B | Qwen2.5-Math-7B | Qwen2-VL-2B |
|---|---|---|---|
| Learning rate | 1.0e-6 | | 6.0e-6 |
| c-NLL loss lambda $\lambda$ | 0.2 | | 0.001 |
| Data threshold $\tau_{good}$ | 8.0 | | 2.5 |
| DPO beta $\beta$ | | 2.0 | |
| $w_{perf}$ param $\tau_{ref}$ | | 2.0 | |
| $w_{qual}$ param $\eta$ | | 0.7 | |
| $w_{perf}$ param $\gamma$ | | 3.0 | |
| Mix precision training | | Bfloat16 | |
| Learning rate scheduler | | Cosine | |
| Global batch size | | 128 | |
| DPO label smoothing | | 0.0 | |
| Warmup ratio | | 0.0 | |
| Seed | | 42 | |

---

**Algorithm 1** Uni-DPO loss calculation

---

**Input:** Training samples $(\boldsymbol{x}, \boldsymbol{y}_w, \boldsymbol{y}_l) \sim D$
**Hyperparameters:** eta($\eta$), beta($\beta$), gamma($\gamma$), lambda($\lambda$), tao_ref($\tau_{ref}$), tao_good($\tau_{good}$)
**Output:** Uni-DPO loss

```
1:  import torch
2:  import torch.nn.functional as F
3:
4:  def get_Uni_DPO_loss(self, (x, y_w, y_l)) -> torch.Tensor:
5:      # policy model forward
6:      policy_chosen_logps = model.dpo_forward((x, y_w)) # Note that the 'logps' are length-normalized
7:      policy_rejected_logps = model.dpo_forward((x, y_l))
8:      policy_logratios = policy_chosen_logps - policy_rejected_logps
9:
10:     # reference model forward
11:     with torch.no_grad():
12:         ref_chosen_logps = ref_model.dpo_forward((x, y_w))
13:         ref_rejected_logps = ref_model.dpo_forward((x, y_l))
14:     ref_logratios = ref_chosen_logps - ref_rejected_logps
15:
16:     logits = policy_logratios - ref_logratios
17:
18:     # calculate dual adaptive weights
19:     w_quality = compute_weight_quality(score_margin, self.eta)
20:     w_performance = (1 - F.sigmoid(self.beta * policy_logratios - self.tao_ref)) ** self.gamma
21:
22:     # calculate loss
23:     losses = - w_quality * w_performance * F.logsigmoid(self.beta * logits)
24:
25:     # calculate c-NLL loss
26:     positive_coeffi = self.lambda * sign(reference_chosen_logps - policy_chosen_logps)
27:     positive_coeffi = torch.where(score_chosen >= tau_good, positive_coeffi, 0)
28:     losses -= positive_coeffi * policy_chosen_logps
29:
30:     return loss.mean()
```

---

exponentiation, masking) on vector outputs after the LLM forward pass. These operations contribute only a constant-factor overhead $C$ of around a few hundred FLOPs. Formally, we have:

$$F_{\text{Uni-DPO}} \approx F_{\text{DPO}} + C, \quad \text{with } C \ll F_{\text{DPO}}. \tag{29}$$

In practice, this means that training complexity is effectively identical between DPO and Uni-DPO. Our experiments also verify that the total FLOPs difference between DPO (3.870951e+17) and Uni-DPO (3.870986e+17) is negligible.

In terms of memory, Uni-DPO requires storing additional intermediate tensors such as the weight ($w_{qual}$ and $w_{perf}$) and binary masks. However, these also contribute only a small constant overhead relative to the total activation memory required by LLM inference, which dominates overall GPU memory consumption.

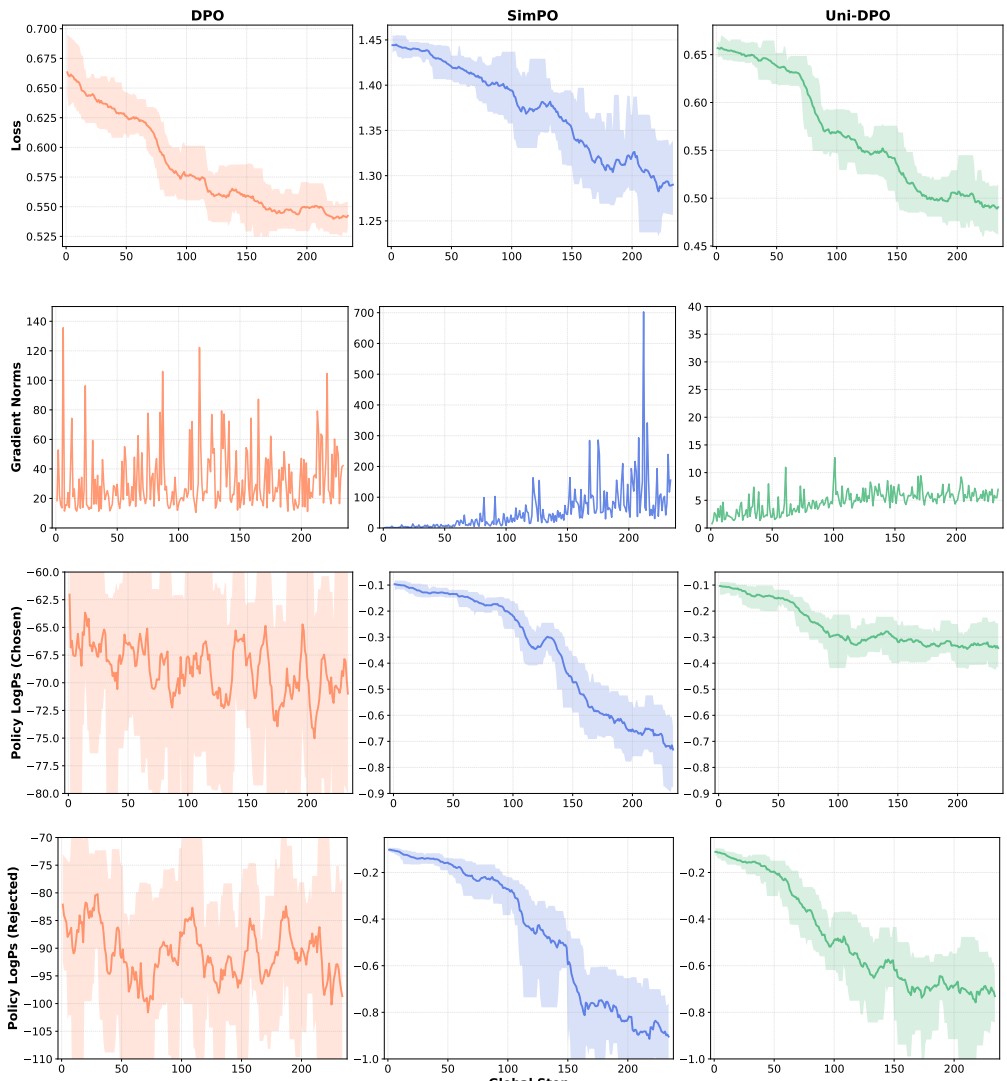

Figure C.1: **Training dynamics on mathematical reasoning tasks.** From top to bottom: training loss, gradient norm, policy log probabilities for chosen responses, and those for rejected responses across training steps.

## C.5 TRAINING DYNAMICS

To more intuitively illustrate the behavior of our Uni-DPO compared with DPO and SimPO during training, we provide training curve comparisons in this section, as shown in Fig. C.1. The training configuration for Uni-DPO is detailed in Tab. C.3. For DPO and SimPO, we perform a grid search over the hyperparameters and report the best-performing configuration. [9] The analysis is as follows.

**Uni-DPO provides more stable training dynamics.** From the first two columns of the figure, we observe that Uni-DPO exhibits more stable training behavior, with smoother and more consistent loss curves and fewer spikes in gradient norms. This suggests that Uni-DPO is less prone to gradient explosion or vanishing issues, enabling more effective learning and convergence. This observation is consistent with the results in Fig. 5b. We attribute this stability to our improved design of the performance-weighting factor $w_{\mathrm{perf}}$ in the optimization objective.

**Uni-DPO better preserves the generation probability of positive samples.** As shown in the third column of the figure, Uni-DPO can better preserve the generation probability of positive samples compared with SimPO. This benefit comes from the c-NLL loss $\mathcal{L}_{\mathrm{c\text{-}NLL}}$ introduced in Sec. 3.4 (main

---

[9]The corresponding evolution of test performance with respect to training steps is shown in Fig. D.1.

paper), which explicitly encourages the model to increase the likelihood of high-quality and hard positive samples, thereby helping it better capture the distribution of good responses.

**Uni-DPO better separates positive and negative samples.** From the last two columns of the figure, we see that Uni-DPO enables the model to better distinguish between positive and negative samples compared with DPO. This indicates that Uni-DPO leverages preference information more effectively during training, leading to stronger discriminative capability. This also validates the effectiveness of the performance-weighting factor $w_{\text{perf}}$ proposed in Sec. 3.3 (main paper).

## D EVALUATION DETAILS

We present detailed descriptions of the evaluation setup (Sec. D.1), the evaluation results (Sec. D.2), additional ablation studies (Sec. D.3), and downstream task evaluation (Sec. D.4). By providing these, we aim to further validate the effectiveness and scalability of Uni-DPO and to enhance reproducibility.

### D.1 EVALUATION SETUP

We present the evaluation setup, including the evaluation benchmarks and decoding strategy.

**Evaluation benchmarks.** We provide the details about the evaluation benchmarks used in this work. The evaluation benchmarks for textual understanding ability are summarized in Tab. D.1.

The descriptions of the benchmarks of *textual understanding evaluation* are as follows:

- **AlpacaEval 2.0** (Li et al., 2023b) comprises 805 questions sourced from five diverse datasets. We report both the raw win rate (WR) and the length-controlled win rate (LC) (Dubois et al., 2024), where LC mitigates verbosity bias using a regression-based causal inference approach to adjust for and control potential response length bias, thereby enhancing the accuracy and robustness of automated evaluation metrics. The prompt template is shown in Tab. D.2.

- **Arena-Hard v0.1** (Li et al., 2024a) builds on MT-Bench (Zheng et al., 2023b) with 500 technically challenging problem-solving queries. We measure performance via WR against the designated baseline model GPT-4-0314. The prompt template is shown in Tab. D.3.

- **IFEval** (Zhou et al., 2023) contains approximately 500 prompts based on 25 "verifiable instructions" (*e.g.,* "write more than 400 words," "mention the keyword 'AI' at least three times"). We report both strict accuracy (Strict Acc.), which demands exact compliance down to formatting details, and loose accuracy (Loose Acc.), which allows minor variations such as differences in case or markup.

- **SedarEval** (Fan et al., 2025; Xu et al., 2025a) is a novel evaluation paradigm that introduces a self-adaptive rubric for scoring 1,000 questions spanning domains such as long-tail knowledge, mathematics, coding, and logical reasoning. Unlike traditional scoring methods, SedarEval's approach provides a more accurate reflection of the problem-solving process. Using an evaluator LM trained on the same dataset, SedarEval achieves higher concordance with human judgments than even GPT-4 (Achiam et al., 2023), offering a more faithful assessment of problem-solving quality. The template is from the official repository.

For *math reasoning ability*, we evaluate model performance on standard mathematical reasoning benchmarks, including GSM8K (Cobbe et al., 2021), MATH 500 (Hendrycks et al., 2021), Minerva Math (Lewkowycz et al., 2022), GaoKao 2023 En, CollegeMath (Tang et al., 2024), and Olympiad Bench (He et al., 2024a), as well as on competition-level benchmarks AIME2024 and AMC2023.

For *multimodal* evaluation, we assess model performance across a diverse set of benchmarks:

- Chart and document understanding: ChartQA (Masry et al., 2022), InfoVQA (Mathew et al., 2022)
- Optical character recognition: OCRBench (Liu et al., 2024g), TextVQA (Singh et al., 2019)
- Hallucination detection: POPE (Li et al., 2023c)
- General-knowledge reasoning: MME (Fu et al., 2023), MMBench (Liu et al., 2024f), SEED-Bench (Li et al., 2023a), ScienceQA (Lu et al., 2022)
- High-resolution and real-world utility: RealWorldQA (xAI, 2024)

Table D.1: **Evaluation benchmarks for textual understanding.** The baseline model refers to the model being compared against. A unified judge model GPT-4o 2024-05-13 is employed across benchmarks to ensure fairness.

| | # Exs. | Baseline Model | Judge Model | Scoring Type | Metric |
|---|---|---|---|---|---|
| **AlpacaEval 2.0** (Li et al., 2023b) | 805 | GPT-4 Turbo | | Pairwise comparison | LC & raw win rate |
| **Arena-Hard v0.1** (Li et al., 2024a) | 500 | GPT-4-0314 | GPT-4o | Pairwise comparison | Win rate |
| **SedarEval** (Fan et al., 2025) | 1,000 | - | 2024-05-13 | Single-answer grading | Rating of 0∼5 |
| **IFEval** (Zhou et al., 2023) | 541 | - | - | Rule-based checking | Strict/Loose Acc. |

Table D.2: **Prompts for AlpacaEval 2.0 (Li et al., 2023b).** This template lets the assistant compare two model responses for a given instruction and return a human-preference rank. This template is from official repository.

```
<|im_start|>system
You are a helpful assistant that ranks models by the quality of their answers.
<|im_end|>
<|im_start|>user
I want you to create a leaderboard of different large-language models. To do so, I will give you the
instructions (prompts) given to the models and the responses of two models. Please rank the models based
on which responses would be preferred by humans. All inputs and outputs should be Python dictionaries.
Here is the prompt:
{
    "instruction": """"instruction"""",
}
Here are the outputs of the models:
[
    {
        "model": "model_1",
        "answer": """"{output_1}""""
    },
    {
        "model": "model_2",
        "answer": """"{output_2}""""
    }
]
Now, please rank the models by the quality of their answers, so that the model with rank 1 has the best
output. Then return a list of the model names and ranks, i.e., produce the following output:
[
    {'model': <model-name>, 'rank': <model-rank>},
    {'model': <model-name>, 'rank': <model-rank>}
]

Your response must be a valid Python dictionary and should contain nothing else because we will directly
execute it in Python. Please provide the ranking that the majority of humans would give.
<|im_end|>
```

**Decoding strategies.** For both textual understanding and mathematical reasoning tasks, we adopt *greedy decoding*, selecting the highest-probability token at each step. All math evaluations use the same prompt template as training (see Tab. C.2), and we report single-run pass@1 results for consistency. For multimodal tasks, we strictly follow the recommendation of the widely used (Chen et al., 2024c) evaluation framework VLMEvalKit (Duan et al., 2024) and perform all the evaluations.

## D.2 EVALUATION RESULTS

In this section, we provide additional details on the evaluation results to supplement the main paper.

**Dynamics of mathematical reasoning performance over training.** To gain a deeper understanding of the training dynamics of Uni-DPO on mathematical reasoning tasks, we periodically evaluate the model on multiple benchmarks throughout training. The overall training dynamics are shown in Fig. C.1, while Fig. D.1 reports the test scores of Uni-DPO, DPO, and SimPO on eight mathematical benchmarks at different training steps. From the figure, Uni-DPO exhibits two clear advantages over DPO and SimPO. First, under exactly the same training data, Uni-DPO typically achieves higher average scores and also reaches higher peak performance. Second, Uni-DPO maintains its performance more stably in the later stages of training, whereas the scores of DPO and SimPO tend to fluctuate or degrade. These observations further validate the effectiveness and robustness of Uni-DPO for mathematical reasoning tasks.

Table D.3: **Prompts for Arena-Hard benchmark.** This template instructs the system to generate its own answer, then compare and judge two assistants' responses against that answer, outputting one of five verdict labels to indicate which assistant is preferred. This template is from the official repository.

> **system_prompt:**
> "Please act as an impartial judge and evaluate the quality of the responses provided by two AI assistants to the user prompt displayed below. You will be given assistant A's answer and assistant B's answer. Your job is to evaluate which assistant's answer is better.\n\n
> Begin your evaluation by generating your own answer to the prompt. You must provide your answers before judging any answers.\n\n
> When evaluating the assistants' answers, compare both assistants' answers with your answer. You must identify and correct any mistakes or inaccurate information.\n\n
> Then consider if the assistant's answers are helpful, relevant, and concise. Helpful means the answer correctly responds to the prompt or follows the instructions. Note when user prompt has any ambiguity or more than one interpretation, it is more helpful and appropriate to ask for clarifications or more information from the user than providing an answer based on assumptions. Relevant means all parts of the response closely connect or are appropriate to what is being asked. Concise means the response is clear and not verbose or excessive.\n\n
> Then consider the creativity and novelty of the assistant's answers when needed. Finally, identify any missing important information in the assistants' answers that would be beneficial to include when responding to the user prompt.\n\n
> After providing your explanation, you must output only one of the following choices as your final verdict with a label:\n\n
> 1. Assistant A is significantly better: [[A≫B]]\n
> 2. Assistant A is slightly better: [[A>B]]\n
> 3. Tie, relatively the same: [[A=B]]\n
> 4. Assistant B is slightly better: [[B>A]]\n
> 5. Assistant B is significantly better: [[B≫A]]\n\n
> Example output: "My final verdict is tie: [[A=B]]"."
>
> **prompt_template:**
> "<|User Prompt|>\nquestion_1\n\n
> <|The Start of Assistant A's Answer|>\n{answer_1}\n<|The End of Assistant A's Answer|>\n\n
> <|The Start of Assistant B's Answer|>\n{answer_2}\n<|The End of Assistant B's Answer|>"

Table D.4: **Main result of multimodal tasks.** We evaluate the performance of Qwen2-VL-2B (Wang et al., 2024d) base model on various multimodal benchmarks. The best results are highlighted in **bold** and the second best in underline. The results demonstrate that Uni-DPO consistently outperforms the baseline, DPO, and SimPO across all benchmarks, achieving significant improvements in overall performance.

| Model | Method | ChartQA$_{test}$ Overall(%) | InfoVQA$_{val}$ Overall(%) | OCRBench Final Score | TextVQA Overall(%) | POPE Overall(%) |
|---|---|---|---|---|---|---|
| Qwen2-VL-2B | Baseline | 5.04 | 37.77 | 669 | 50.21 | 79.55 |
| | DPO | 6.40 | 39.73 | 677 | 52.73 | 80.84 |
| | SimPO | 11.68 | 41.51 | 635 | 60.63 | 78.71 |
| | Uni-DPO | **13.84** | **45.28** | **709** | **73.23** | **86.03** |

| Model | Method | MME Perception + Reasoning | MMBench$_{dev\_en}$ Overall(%) | SEEDBench$_{image}$ Overall(%) | ScienceQA$_{val}$ Overall(%) | RealWordQA Overall(%) |
|---|---|---|---|---|---|---|
| Qwen2-VL-2B | Baseline | 1744.7 | 50.09 | 53.98 | 65.57 | 53.86 |
| | DPO | 1799.9 | 43.73 | 50.65 | 63.00 | 54.77 |
| | SimPO | 1813.2 | 54.38 | 59.57 | 66.81 | 53.99 |
| | Uni-DPO | **1905.1** | **63.75** | **70.83** | **68.10** | **55.29** |

**Results on multimodal tasks.** To further demonstrate the effectiveness of Uni-DPO, we extend our evaluation to the multimodal domain. As in Tab. D.4, Uni-DPO outperforms the baseline model and preference optimization methods DPO (Rafailov et al., 2023) and SimPO (Meng et al., 2024) by a significant margin in all benchmarks, showing the effectiveness of Uni-DPO in multimodal tasks.[10]

---

[10] Due to computational resource limitations, training on multimodal tasks was restricted to a 2B-parameter model. Nevertheless, the encouraging results suggest that scaling to larger models is likely to yield further performance improvements, which we defer to future work.

Table D.5: **Mathematical reasoning evaluation and comparison with additional methods.** We compare *SFT* methods, *alignment* methods, and *RL-based* methods on the Qwen2.5-Math-7B model. Results with † are from Zhang et al. (2025a), and those with ∗ are from Cui et al. (2025). The remaining are from our evaluations. Our Uni-DPO consistently and significantly outperforms the baseline model, SFT-style methods, and other alignment approaches. In particular, Iterative DPO applies *seven* DPO iterations and thus uses roughly *seven* times more training data than Uni-DPO, whereas Uni-DPO uses only one iteration. Uni-DPO surpasses Iterative DPO on almost all benchmarks as well as on the average score, demonstrating its effectiveness. Moreover, compared with RL-based methods (Eurus2-PRIME, OpenReasoner, SimpleRL, Dr. GRPO, and PPO), Uni-DPO achieves noticeably better performance while maintaining a much simpler and more efficient training procedure.

| Model | Method type | Method | Result source | GSM8K | MATH | Minerva Math | Olympiad Bench | AIME 2024 | AMC 2023 | GaoKao 2023 En | College Math | Average |
|---|---|---|---|---|---|---|---|---|---|---|---|---|
| | | Baseline | | 64.3 | 65.8 | 10.7 | 24.3 | 23.3 | 47.5 | 44.7 | 32.3 | 39.11 |
| | | - | † | - | 65.4 | 9.9 | 23.4 | 23.3 | 47.5 | - | - | 38.85 |
| | multirow6*Supervised Fine-tuning | SFT | | **91.3** | 73.2 | 30.5 | 35.6 | 20.0 | **62.5** | 63.6 | **51.6** | **53.54** |
| | | | † | - | 73.2 | 30.5 | 35.6 | 20.0 | **62.5** | - | - | **53.54** |
| | | Eurus2-SFT | | 89.5 | 64.0 | 30.1 | 28.7 | 3.3 | 42.5 | 52.2 | 43.2 | 44.19 |
| | | | * | - | 65.1 | **32.7** | 29.8 | 3.3 | 30.1 | - | - | 43.24 |
| | | RAFT | | 87.6 | **77.6** | 32.7 | 38.2 | 20.0 | 57.5 | 63.6 | 50.6 | 53.48 |
| Qwen2.5-Math-7B Base | | | † | - | **77.6** | 30.5 | **38.7** | 20.0 | 55.0 | - | - | 52.95 |
| | Preference Alignment | DPO | | 83.2 | 75.8 | 20.2 | 37.9 | 26.7 | 57.5 | 61.6 | 49.5 | 51.55 |
| | | Iterative DPO | | 86.7 | 75.4 | 29.0 | 37.9 | 30.0 | 60.0 | 63.4 | 48.9 | 53.91 |
| | | SimPO | | 85.7 | 76.4 | 32.0 | 39.3 | 26.7 | 57.5 | 62.1 | 50.1 | 53.73 |
| | | **Uni-DPO** | | **88.9** | **78.2** | **34.6** | **41.5** | 26.7 | **67.5** | **65.7** | **51.3** | **56.80** |
| | Reinforcement Learning | Eurus2-PRIME | | 87.7 | 75.4 | 34.6 | 37.3 | 23.3 | 57.5 | 61.3 | 49.7 | 53.35 |
| | | OpenReasoner-Zero-7B | | **93.0** | **79.2** | 31.6 | **44.0** | 13.3 | 54.2 | **68.8** | 48.7 | 54.10 |
| | | SimpleRL-Zero-7B | | 89.2 | 76.4 | 27.6 | 40.3 | 26.7 | 57.5 | 65.5 | 50.3 | 54.19 |
| | | Dr. GRPO (Oat-Zero-7B) | | 89.5 | 79.0 | 33.1 | 42.4 | 33.3 | 60.0 | 64.9 | 50.4 | 56.58 |
| | | PPO-MATH7500 | † | - | 77.2 | 33.8 | 40.7 | 33.8 | 67.5 | - | - | - |
| | | PPO | | 89.1 | 79.0 | 33.8 | 40.7 | **36.7** | 62.5 | 62.1 | **50.8** | **56.84** |
| Qwen2.5-Math-7B-Instruct | - | | * | - | 79.8 | 34.6 | 40.7 | 13.3 | 50.6 | - | - | |
| Llama-3.1-70B-Instruct | - | | * | - | 64.6 | 35.3 | 31.9 | 16.7 | 30.1 | - | - | - |
| GPT-4o | - | | * | - | 76.4 | 36.8 | 43.3 | 9.3 | 45.8 | - | - | - |

Table D.6: **Result of textual understanding in Qwen2.5 model.** WR denotes the Win Rate, LC denotes the Length-Controlled win rate, and Acc. denotes the Accuracy. The best results are highlighted in **bold**. The results show that our Uni-DPO method consistently outperforms SimPO across various model sizes and benchmarks.

| Model | Model Size | Method | AlpacaEval2 | | Arena-Hard | IFEval | | SedarEval |
|---|---|---|---|---|---|---|---|---|
| | | | LC(%) | WR(%) | WR(%) | Strict Acc.(%) | Loose Acc.(%) | Overall(%) |
| Qwen2.5 | 0.5B | SimPO | 0.92 | 1.37 | 1.3 | 12.0 | 14.8 | - |
| | | Uni-DPO | **3.37** | **4.10** | **4.4** | **16.6** | **18.9** | - |
| | 1.5B | SimPO | 0.44 | 0.87 | 2.9 | 16.1 | 17.7 | - |
| | | Uni-DPO | **6.49** | **7.46** | **12.4** | **23.1** | **26.2** | - |
| | 3B | SimPO | 5.81 | 6.02 | 11.8 | **32.5** | **35.3** | - |
| | | Uni-DPO | **10.81** | **11.32** | **18.5** | 32.3 | 34.8 | - |
| | 7B | SimPO | 20.9 | 18.4 | 39.5 | **43.6** | **48.2** | 48.2 |
| | | Uni-DPO | **25.8** | **24.7** | **43.5** | 43.3 | 48.1 | **49.9** |
| | 14B | SimPO | 30.07 | 26.52 | 48.7 | 46.0 | **53.6** | - |
| | | Uni-DPO | **31.99** | **30.75** | **54.5** | **47.1** | 52.5 | - |

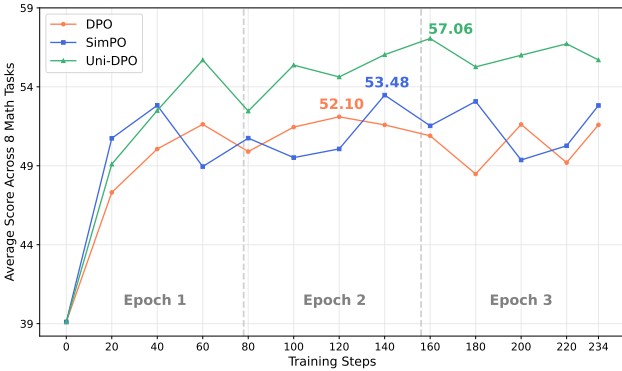

Figure D.1: **Dynamics of performance on mathematical reasoning benchmarks.** We periodically evaluate Uni-DPO, DPO, and SimPO on eight mathematical benchmarks throughout training.

**Comparison with more advanced methods.** We also compare Uni-DPO with more advanced methods, as summarized in Tab. D.5. We first consider the Iterative DPO method proposed by Zhang et al. (2025a) on mathematical reasoning tasks, which adopts the same training setups and training datasets as Uni-DPO. Although Iterative DPO performs seven DPO iterations, which use about seven times the training data as Uni-DPO, it is still outperformed by Uni-DPO on almost all benchmarks as well as in terms of the final average score, highlighting the overall effectiveness of Uni-DPO.

Table D.7: **Calibrated negative log-likelihood loss $\mathcal{L}_{\text{c-NLL}}$ ablation results.** The full c-NLL loss delivers substantial performance gains, and each of the two indicator functions makes a meaningful contribution to the overall improvement, which demonstrates the effectiveness of our proposed loss function.

| Model | Method | AlpacaEval2 | | Arena-Hard | IFEval | | SedarEval |
|---|---|---|---|---|---|---|---|
| | | LC(%) | WR(%) | WR(%) | Strict Acc.(%) | Loose Acc.(%) | Overall(%) |
| Llama3-8B-Base | SFT | 5.7 | 3.7 | 3.3 | 33.5 | 35.3 | 20.85 |
| | Uni-DPO | 23.8 | 20.5 | 23.9 | 38.1 | **47.9** | **38.49** |
| | w/o I | 22.8 | 20.0 | **25.7** | 37.2 | 45.5 | 38.13 |
| | w/o II | **24.7** | **20.9** | 20.2 | 36.8 | **47.9** | 36.82 |
| | w/o I·II | 24.3 | **20.9** | 21.9 | 38.8 | 47.1 | 36.82 |
| | w/o $\mathcal{L}_{\text{c-NLL}}$ | 22.4 | 19.4 | 23.3 | **40.7** | 47.5 | 37.73 |
| Llama3-8B-Instruct | SFT | 28.5 | 28.4 | 25.7 | - | - | - |
| | Uni-DPO | **47.2** | **44.2** | 37.3 | 61.2 | **67.8** | **46.32** |
| | w/o I | 46.1 | 42.7 | **38.6** | 60.3 | **67.8** | 45.85 |
| | w/o II | 46.0 | 42.9 | 37.0 | **61.6** | 67.5 | 44.50 |
| | w/o I·II | 46.4 | 43.3 | 38.0 | 60.4 | 67.5 | 44.80 |
| | w/o $\mathcal{L}_{\text{c-NLL}}$ | 46.2 | 43.2 | 37.6 | 58.6 | 65.4 | 44.48 |

In addition, we compare Uni-DPO with RL-based approaches such as Eurus2-PRIME (Cui et al., 2025), OpenReasoner (Hu et al., 2025), SimpleRL (Zeng et al., 2025), Dr. GRPO (Oat-Zero-7B) (Liu et al., 2025), and PPO (Schulman et al., 2017). Uni-DPO outperforms Eurus2-PRIME, OpenReasoner, and SimpleRL on nearly all benchmarks and achieves an overall score that is on par with the model trained with PPO, while enjoying a much simpler and more efficient training procedure.[11]

**Model size scalability.** As shown in the main paper Figs. 6d and 6e, Uni-DPO consistently outperforms SimPO (Meng et al., 2024) across a range of model sizes, demonstrating its effectiveness at scale. Tab. D.6 provides a detailed breakdown of these results, comparing SimPO and Uni-DPO performance metrics for each model variant.[12]

**Detailed statistical results.** In addition to the primary metrics, Tab. D.8 reports further statistics, including standard errors (Std Error), confidence intervals (CI), and average response lengths (Avg. #tokens), to illustrate the distribution and the reliability of the evaluation results.

**Radar plot details.** To better visualize Uni-DPO's performance across multiple domains, we generate four radar charts comparing our method against strong baselines to highlight key differences:

- Textual understanding (8B, the main paper Fig. 1a): Results for Llama3-8B-Base (AI@Meta, 2024b). SimPO and Uni-DPO results correspond to the *v0.2* setup; see the main paper Tab. 1 for exact numbers.
- Mathematical reasoning (1.5B, the main paper Fig. 1b): Performance of Qwen2.5-Math-1.5B (Yang et al., 2024b) base model after finetuning. Detailed scores are in the main paper Tab. 2.
- Mathematical reasoning (7B, the appendix Fig. .7a): Results for Qwen2.5-Math-7B (Yang et al., 2024b) base model; see Tab. D.5 for the underlying data.
- Multimodal tasks (2B, the appendix Fig. .7b): Evaluation of Qwen2-VL-2B (Wang et al., 2024d) base model on our multimodal benchmark suite. Exact metrics appear in Tab. D.4.

## D.3 FURTHER ABLATION STUDY

In addition to the ablations presented in the main paper Sec. 4.3, which evaluate the individual contributions of each component in the Uni-DPO framework, we further conducted studies focusing on the role and strength of the calibrated negative log-likelihood loss ($\mathcal{L}_{\text{c-NLL}}$). This loss specifically targets challenging yet high-quality positive samples. We also examined the effectiveness of different selections of the expert model, which demonstrates that Uni-DPO's performance gains stem from its ability to dynamically allocate focus across training examples.

---

[11]Details of the Iterative DPO method and its implementation can be found in the official blog. The SFT, RAFT, DPO, Iterative DPO, PPO-MATH7500, and PPO models reported in Tab. D.5 are all from the Iterative DPO work, while the Eurus2-SFT and Eurus2-PRIME models are taken from Cui et al. (2025).

[12]Due to the high costs associated with the extensive GPT-4o (Hurst et al., 2024) API calls required by the SedarEval benchmark (Fan et al., 2025), we report only partial results on this benchmark for reference.

**Ablation of $\mathcal{L}_{\text{c-NLL}}$ components.** As defined in Eq. (11), c-NLL loss includes two indicator functions. To assess the contribution of each component, we conduct ablation experiments by removing each individually. The results, presented in Tab. D.7, provide further insights into the effectiveness of different parts of the loss design.

- Without the first indicator $\mathbf{1}\left(\log \pi_{\text{ref}}(y_w \mid x) > \log \pi_\theta(y_w \mid x)\right)$ (w/o I): This corresponds to not restricting samples where $\pi_\theta$ performs worse than $\pi_{\text{ref}}$. The performance drops slightly, indicating that this term is somewhat important. However, this term may not be optimally configured, potentially requiring a margin parameter $\tau_{\text{in}}$ to achieve the best results. In other words, this indicator term may ideally be formulated as: $\mathbf{1}\left[\frac{\log \pi_{\text{ref}}(y_w|x)}{\log \pi_\theta(y_w|x)} > \tau_{\text{in}}\right]$. We leave this for future exploration.

- Without the second indicator $\mathbf{1}\left(S_w \geq \tau_{\text{good}}\right)$ (w/o II): This means we apply NLL loss compensation to positive samples of all quality. The performance drops significantly, showing the importance of controlling the quality of positive samples when performing NLL loss, *i.e.,* compensation should only be applied to high-quality ones.

- Without both indicators (w/o I·II): In this variant, the NLL loss is applied uniformly to all positive samples $y_w$ without any filtering. Performance degrades substantially, confirming the necessity of selectively augmenting the hard yet high-quality positive examples. Nevertheless, this configuration still slightly outperforms the complete removal of the NLL loss ("w/o $\mathcal{L}_{\text{c-NLL}}$" entry in the table), indicating that the NLL component contributes positively to overall model performance.

**Ablation of $\mathcal{L}_{\text{c-NLL}}$ strength $\lambda$.** We further evaluate the effect of the strength $\lambda$ on the $\mathcal{L}_{\text{c-NLL}}$ term across different domains. While $\mathcal{L}_{\text{c-NLL}}$ is necessary for boosting model capabilities (as discussed above), its optimal strength may vary by task. We sweep $\lambda$ over a wide range of values:

$$\{0.0001, 0.0005, 0.001, 0.005, 0.01, 0.05, 0.1, 0.2, 0.3, 0.5, 1\},$$

and report the final settings in Tabs. C.1 and C.3. For mathematical reasoning tasks, a larger weight (better at $\lambda = 0.2$) is required to achieve peak performance. In contrast, for text understanding and multimodal tasks, a much smaller weight (*e.g.,* $\lambda = 0.001$) suffices. These observations are consistent with the settings of prior work (Pang et al., 2024). We hope these findings will inspire further research into the optimal strength of $\mathcal{L}_{\text{c-NLL}}$.

**Ablation of expert model.** In our main experiments, we strictly follow related works (such as SimPO) in using the same publicly available dataset, UltraFeedback, with preference samples and quality scores given by the dataset as our training signal. Compared with SimPO, Uni-DPO exploits these resources at a finer granularity, enabling adaptive corrections during preference learning. As shown in Tabs. 1 and 2 (main paper) and Tab. D.4 (Appendix), this design consistently improves performance across modalities and benchmarks, demonstrating its effectiveness.

To further verify that the improvement does not stem from the use of a strong expert model, but rather from Uni-DPO's fine-grained utilization of training data, we conduct ablation studies using open-source models to provide the quality scores. Specifically, as shown in Tab. D.9, we replace GPT-4o with Qwen2.5-72B and ArmoRM-7.5B as weaker expert models for data annotation. Except for replacing the expert model, all other settings (*e.g.,* the scoring prompts) are kept unchanged.

On the left, we apply the Uni-DPO framework using data quality scores derived from a weaker and more accessible open-source model, Qwen2.5-72B. The results indicate that even with a weaker expert model, Uni-DPO still achieves competitive performance. On the right, we use the 7.5B ArmoRM model, which is weaker and more accessible than GPT-4o, for scoring. The results also outperformed SimPO, demonstrating the stability and scalability of Uni-DPO. These results highlight that the effectiveness of Uni-DPO does not depend on the availability of a particularly strong expert model, but instead arises from its ability to dynamically assign focus to each training example.

## D.4 DOWNSTREAM TASK EVALUATION

To investigate how preference optimization methods impact downstream task performance, we evaluate models on the benchmarks included in the Huggingface Open LLM Leaderboard (Beeching et al., 2023). The SFT, DPO, and SimPO checkpoints used for comparison are provided by SimPO (Meng et al., 2024), which is consistent with those in the main paper. Specifically, we report results on MMLU (Hendrycks et al., 2020), ARC (Clark et al., 2018), HellaSwag (Zellers et al., 2019), Truth-

Table D.8: **Detailed statistics for textual understanding experiments.** This table presents standard errors, 95% confidence intervals (CI), and average response token lengths for each model variant across the benchmarks.

| Model | Method | AlpacaEval2 | | | | Arena-Hard | | |
|---|---|---|---|---|---|---|---|---|
| | | LC(%) | WR(%) | Std Error | Avg. #tokens | WR(%) | 95% CI | Avg. #tokens |
| Llama3-8B Base | SFT | 5.7 | 3.7 | 0.66 | 1000 | 3.3 | $(-0.6, 0.8)$ | 389 |
| | DPO | 15.5 | 13.3 | 1.20 | 1708 | 15.9 | $(-1.6, 1.6)$ | 542 |
| | SimPO | 19.4 | 18.1 | 1.36 | 1883 | 23.4 | $(-1.9, 1.7)$ | 704 |
| | Uni-DPO | **23.8** | **20.5** | 1.43 | 1757 | **23.9** | $(-1.7, 1.6)$ | 553 |
| | SimPO$_{v0.2}$ | 15.5 | 18.5 | 1.37 | 2349 | 29.3 | $(-2.2, 2.4)$ | 751 |
| | Uni-DPO$_{v0.2}$ | **22.8** | **24.1** | 1.51 | 2139 | **30.7** | $(-2.1, 1.9)$ | 668 |
| Llama3-8B Instruct | SFT | 28.5 | 28.4 | 1.59 | 1955 | 25.7 | $(-2.1, 1.8)$ | 585 |
| | DPO | 43.7 | 41.7 | 1.74 | 1890 | 32.6 | $(-2.3, 2.2)$ | 528 |
| | SimPO | 44.7 | 40.4 | 1.73 | 1779 | 33.8 | $(-1.8, 2.1)$ | 504 |
| | Uni-DPO | **47.2** | **44.2** | 1.75 | 1862 | **37.3** | $(-2.2, 2.1)$ | 524 |
| | SimPO$_{v0.2}$ | 41.2 | 37.1 | 1.70 | 1787 | 36.5 | $(-2.0, 1.9)$ | 530 |
| | Uni-DPO$_{v0.2}$ | **46.8** | **42.6** | 1.74 | 1811 | **40.6** | $(-2.5, 2.1)$ | 518 |
| Gemma-2-9B-IT | SimPO | 53.2 | 48.0 | 1.76 | 1759 | 59.1 | $(-2.2, 1.9)$ | 693 |
| | Uni-DPO | **54.7** | **52.7** | 1.76 | 1968 | **67.1** | $(-2.4, 2.2)$ | 751 |
| Qwen2.5-0.5B | SimPO | 0.92 | 1.37 | 0.41 | 4585 | 1.3 | $(-0.4, 0.4)$ | 1089 |
| | Uni-DPO | **3.37** | **4.10** | 0.70 | 2763 | **4.4** | $(-0.9, 0.8)$ | 678 |
| Qwen2.5-1.5B | SimPO | 0.44 | 0.87 | 0.33 | 6626 | 2.9 | $(-0.5, 0.8)$ | 2199 |
| | Uni-DPO | **6.49** | **7.46** | 0.93 | 3591 | **12.4** | $(-1.6, 1.3)$ | 938 |
| Qwen2.5-3B | SimPO | 5.81 | 6.02 | 0.84 | 2556 | 11.8 | $(-1.3, 1.3)$ | 1816 |
| | Uni-DPO | **10.81** | **11.32** | 1.12 | 2551 | **18.5** | $(-1.7, 1.7)$ | 1050 |
| Qwen2.5-7B | SimPO | 20.9 | 18.4 | 1.37 | 1809 | 39.5 | $(-1.6, 2.9)$ | 585 |
| | Uni-DPO | **25.8** | **24.7** | 1.52 | 2005 | **43.5** | $(-2.4, 2.2)$ | 567 |
| Qwen2.5-14B | SimPO | 30.07 | 26.52 | 1.56 | 1629 | 48.7 | $(-2.7, 1.9)$ | 471 |
| | Uni-DPO | **31.99** | **30.75** | 1.63 | 2126 | **54.5** | $(-2.8, 2.1)$ | 551 |

Table D.9: Ablation study on the expert model used for scoring. Uni-DPO maintains competitive or superior performance even when using weaker and more accessible open source models such as Qwen2.5-72B and ArmoRM-7.5B, for scoring the preference training data.

| Method | Llama3-8B-Base | | | Method | Llama3-8B-Instruct | | |
|---|---|---|---|---|---|---|---|
| | Arena-Hard | IFEval | | | Arena-Hard | IFEval | |
| | WR% | Strict Acc.(%) | Loose Acc.(%) | | WR% | Strict Acc.(%) | Loose Acc.(%) |
| SFT | 3.3 | 33.5 | 35.3 | SFT | 25.7 | - | - |
| DPO (GPT-4) | 15.9 | **40.5** | 45.5 | DPO | 32.6 | - | - |
| SimPO (GPT-4) | 23.4 | **40.5** | 45.7 | SimPO$_{v0.2}$ | 36.5 | 60.8 | 68.6 |
| Uni-DPO (GPT-4) | 23.9 | 38.1 | **47.9** | Uni-DPO$_{v0.2}$ (GPT-4o) | 40.6 | **66.9** | 73.2 |
| Uni-DPO (Qwen) | **24.2** | 39.2 | 46.0 | Uni-DPO$_{v0.2}$ (ArmoRM) | 37.5 | 66.2 | **73.8** |

fulQA (Lin et al., 2022), Winogrande (Levesque et al., 2012) and CommonsenseQA (Talmor et al., 2019). We strictly follow the standard evaluation protocols, and show the results in Tab. D.10.

**Substantial gains in knowledge-intensive understanding.** Compared with the SFT checkpoint, DPO, SimPO, and Uni-DPO all yield improvements on MMLU. For the Llama3-8B family, Uni-DPO achieves gains that are comparable to those of DPO and SimPO. In contrast, for Gemma-2-9B-IT and Qwen2.5-7B, Uni-DPO delivers noticeably larger improvements, suggesting that our method can be particularly beneficial for knowledge-intensive tasks.

**Reading comprehension and commonsense reasoning generally improve.** On ARC, Winogrande, and CommonsenseQA, preference optimization methods consistently improve performance over the SFT checkpoint. We hypothesize that preference training helps the model better exploit context and strengthens its reading comprehension and commonsense reasoning abilities. An exception is HellaSwag, where most preference optimization methods lead to a moderate performance drop. Nevertheless, Uni-DPO exhibits the smallest degradation among DPO, SimPO, and Uni-DPO, thus best preserving the original capability on this benchmark.

**Truthfulness consistently benefits from preference optimization.** Interestingly, we observe that preference optimization methods consistently improve TruthfulQA performance relative to the SFT checkpoint, with gains exceeding 16% in some settings. In particular, Uni-DPO achieves substantial improvements over the SFT baseline and outperforms both DPO and SimPO on TruthfulQA. We conjecture that the preference data contain a non-trivial amount of supervision that implicitly empha-

Table D.10: **Downstream task evaluation.** We strictly follow the standard evaluation protocols provided by Language Model Evaluation Harness(Gao et al., 2024) for each task. The best and second-best results are highlighted in **bold** and underline, respectively.

| Model | Method | MMLU | ARC | HellaSwag | TruthfulQA | Winogrande | CommonsenseQA | Average |
|-------|--------|------|-----|-----------|------------|------------|---------------|---------|
| Llama3-8B Base | SFT | 60.18 | 77.44 | 75.62 | 46.66 | 70.72 | 66.83 | 66.24 |
| | DPO | 61.83 | 74.79 | 71.98 | 59.23 | 60.85 | 67.24 | 65.99 |
| | SimPO | 61.95 | 80.81 | 72.64 | 63.48 | 65.82 | 70.35 | 69.18 |
| | Uni-DPO | 60.41 | 81.65 | 78.75 | 67.00 | 71.67 | 57.74 | 69.54 |
| | Uni-DPO (Qwen) | 60.44 | 82.45 | 79.08 | 64.57 | 72.69 | 62.33 | **70.26** |
| Llama3-8B Instruct | Baseline | 58.08 | 77.10 | 64.80 | 52.81 | 64.17 | 45.70 | 60.44 |
| | DPO | 62.95 | 68.86 | 48.17 | 57.72 | 57.30 | 73.22 | 61.37 |
| | DPO$_{v0.2}$ | 64.63 | 68.35 | 40.70 | 57.44 | 57.38 | 74.77 | 60.55 |
| | SimPO | 61.91 | 74.16 | 42.13 | 62.24 | 57.38 | 75.84 | 62.28 |
| | SimPO$_{v0.2}$ | 63.77 | 79.59 | 52.19 | 65.33 | 61.33 | 76.17 | 66.39 |
| | Uni-DPO | 63.19 | 76.35 | 47.95 | 61.84 | 58.96 | 75.18 | 63.91 |
| | Uni-DPO$_{v0.2}$ (GPT-4o) | 64.21 | 75.08 | 53.48 | 64.12 | 59.35 | 73.87 | 65.02 |
| | Uni-DPO$_{v0.2}$ (ArmoRM) | 63.86 | 79.84 | 60.60 | 64.51 | 65.19 | 74.77 | **68.13** |
| Gemma-2-9B-IT | Baseline | 33.46 | 79.12 | 67.11 | 61.41 | 70.40 | 37.59 | 58.18 |
| | DPO | 36.84 | 61.15 | 27.38 | 57.17 | 52.72 | 31.61 | 44.48 |
| | SimPO | 61.80 | 67.51 | 28.88 | 59.83 | 56.43 | 42.92 | 52.89 |
| | Uni-DPO | 65.82 | 79.04 | 61.59 | 62.03 | 69.22 | 59.46 | **66.19** |
| Qwen2.5-7B | Baseline | 53.16 | 65.87 | 75.02 | 51.65 | 69.22 | 78.05 | 65.49 |
| | Uni-DPO | 70.50 | 83.50 | 81.49 | 67.55 | 73.72 | 79.20 | **75.99** |

sizes factuality and honesty, which encourages the model to produce more truthful responses. The construction process of open-source preference datasets such as UltraFeedback likely incorporates considerations of response truthfulness, further contributing to the observed gains on TruthfulQA.

Overall, Uni-DPO not only outperforms DPO and SimPO on textual understanding benchmarks but also delivers great improvements on a broad range of downstream tasks, with especially pronounced gains in knowledge-intensive understanding and truthfulness. We believe that a more systematic study of how different preference optimization strategies affect downstream performance would be valuable and call for rigorous, large-scale analyses in future work.

# E    DISCUSSIONS

## E.1    REVISITING UNI-DPO THROUGH THE RL PARADIGM

**Unified RL paradigm.**    While the main paper (Sec. 5) outlines the RL perspective underlying Uni-DPO, here we provide a more detailed examination of its foundational principles. The *Unified Paradigm* provided by Shao et al. (2024b) shows that different training methods, such as SFT, DPO (Rafailov et al., 2023), PPO (Schulman et al., 2017), and GRPO (Shao et al., 2024b), can be conceptualized as either direct or simplified RL methods. The paradigm can be expressed as:

$$\nabla_\theta \mathcal{L}_\mathcal{A} = \mathbb{E}[\underbrace{(x,y) \sim \mathcal{D}}_{\text{Data Source}}] \left( \frac{1}{|y|} \sum_{t=1}^{|y|} \underbrace{GC_\mathcal{A}(x,y,t,\pi_{\text{rf}})}_{\text{Gradient Coefficient}} \nabla_\theta \log \pi_\theta(y_t|q,y_{<t}) \right). \tag{30}$$

There exist three key components: a) Data source $\mathcal{D}$, which specifies the set of training examples used to learn preferences; b) Reward function $\pi_{\text{rf}}$, which produces the scalar reward signals that guide model updates; c) Algorithm $\mathcal{A}$, which consumes both the training data and the reward outputs to compute a gradient coefficient $GC_\mathcal{A}$, thereby modulating the magnitude of the reinforcement or penalty applied to each example.

**Gradient coefficient of DPO.**    We begin by presenting the gradient coefficient of DPO as derived by Shao et al. (2024b), which serves as the foundation for our subsequent analysis. For notational convenience, we denote the positive sample $y_w$ in each preference pair by $y^+$, and the negative sample $y_l$ by $y^-$. The objective of DPO is:

$$\mathcal{L}_{\text{DPO}}(\theta) = \mathbb{E}[x \sim P_{\text{sft}}(Q), (y^+, y^-) \sim \pi_{\text{sft}}(Y|x)]$$
$$\log \sigma \left( \beta \frac{1}{|y^+|} \sum_{t=1}^{|y^+|} \log \frac{\pi_\theta(y_t^+|x,y_{<t}^+)}{\pi_{\text{ref}}(y_t^+|x,y_{<t}^+)} - \beta \frac{1}{|y^-|} \sum_{t=1}^{|y^-|} \log \frac{\pi_\theta(y_t^-|x,y_{<t}^-)}{\pi_{\text{ref}}(y_t^-|x,y_{<t}^-)} \right). \tag{31}$$

The gradient of $\mathcal{L}_{\text{DPO}}(\theta)$ is:

$$\nabla_\theta \mathcal{L}_{\text{DPO}}(\theta) = \mathbb{E}[x \sim P_{\text{sft}}(Q), (y^+, y^-) \sim \pi_{\text{sft}}(Y|x)]$$
$$\left( \frac{1}{|y^+|} \sum_{t=1}^{|y^+|} GC_{\text{DPO}}(x,y,t) \nabla_\theta \log \pi_\theta(y_t^+|x, y_{<t}^+) - \frac{1}{|y^-|} \sum_{t=1}^{|y^-|} GC_{\text{DPO}}(x,y,t) \nabla_\theta \log \pi_\theta(y_t^-|x, y_{<t}^-) \right). \tag{32}$$

Data Source: questions in the SFT dataset with outputs sampled from the SFT model.

Reward Function: human preference in the general domain.

Gradient Coefficient:

$$GC_{\text{DPO}}(x,y,t) = \sigma \left( \beta \log \frac{\pi_\theta(y_t^-|x, y_{<t}^-)}{\pi_{\text{ref}}(y_t^-|x, y_{<t}^-)} - \beta \log \frac{\pi_\theta(y_t^+|x, y_{<t}^+)}{\pi_{\text{ref}}(y_t^+|x, y_{<t}^+)} \right). \tag{33}$$

Building on this formulation, we can compare the gradient behavior of DPO and Uni-DPO.

**Gradient coefficient of Uni-DPO.** The objective of Uni-DPO is:

$$\mathcal{L}_{\text{Uni-DPO}}(\theta) = \mathbb{E}[x \sim P_{\text{sft}}(Q), (y^+, y^-) \sim \pi_{\text{sft}}(Y|x)]$$
$$\left\{ w_{\text{qual}} \left[ 1 - \sigma \left( \frac{\beta}{|y^+|} \sum_{t=1}^{|y^+|} \log \pi_\theta(y_t^+|x, y_{<t}^+) - \frac{\beta}{|y^-|} \sum_{t=1}^{|y^-|} \log \pi_\theta(y_t^-|x, y_{<t}^-) - \tau_{\text{ref}} \right) \right]^\gamma \right. \tag{34}$$
$$\left. \cdot \log \sigma \left( \frac{\beta}{|y^+|} \sum_{t=1}^{|y^+|} \log \frac{\pi_\theta(y_t^+|x, y_{<t}^+)}{\pi_{\text{ref}}(y_t^+|x, y_{<t}^+)} - \frac{\beta}{|y^-|} \sum_{t=1}^{|y^-|} \log \frac{\pi_\theta(y_t^-|x, y_{<t}^-)}{\pi_{\text{ref}}(y_t^-|x, y_{<t}^-)} \right) + \frac{\lambda \cdot I}{|y^+|} \sum_{t=1}^{|y^+|} \log \pi_\theta(y_t^+|x, y_{<t}^+) \right\}.$$

The gradient of $\mathcal{L}_{\text{Uni-DPO}}(\theta)$ is:

$$\nabla_\theta \mathcal{L}_{\text{Uni-DPO}}(\theta) = \mathbb{E}[x \sim P_{\text{sft}}(Q), (y^+, y^-) \sim \pi_{\text{sft}}(Y|x)]$$
$$\left( \frac{1}{|y^+|} \sum_{t=1}^{|y^+|} GC_{\text{Uni-DPO}}^{y^+}(x,y,t,\pi_{\text{rf}}) \nabla_\theta \log \pi_\theta(y_t^+|x, y_{<t}^+) - \frac{1}{|y^-|} \sum_{t=1}^{|y^-|} GC_{\text{Uni-DPO}}^{y^-}(x,y,t,\pi_{\text{rf}}) \nabla_\theta \log \pi_\theta(y_t^-|x, y_{<t}^-) \right). \tag{35}$$

Data Source: questions are drawn from the SFT dataset, with outputs sampled from the SFT model.

Reward Function: integrating human preference data from the general domain with predictions from a reward model.

Gradient Coefficient:

$$GC_{\text{Uni-DPO}}^{y^+}(x,y,t,\pi_{\text{rf}}) = \beta \cdot w_{\text{qual}} \cdot (1 - \sigma(\Delta_{\text{adj}}))^\gamma \cdot \left( (1 - \sigma(\Delta_r)) - \gamma \sigma(\Delta_{\text{adj}}) \log \sigma(\Delta_r) \right) + \lambda \cdot I,$$
$$GC_{\text{Uni-DPO}}^{y^-}(x,y,t,\pi_{\text{rf}}) = \beta \cdot w_{\text{qual}} \cdot (1 - \sigma(\Delta_{\text{adj}}))^\gamma \cdot \left( (1 - \sigma(\Delta_r)) - \gamma \sigma(\Delta_{\text{adj}}) \log \sigma(\Delta_r) \right). \tag{36}$$

where $I$ is the indicator function $I = \mathbf{1}\big(\log \pi_{\text{ref}}(y_w \mid x) > \log \pi_\theta(y_w \mid x)\big) \mathbf{1}\big(S_w \geq \tau_{\text{good}}\big)$, and $S_w$ is the score of the preferred completion $y_w$. $\Delta_r$ is the reward margin in Eq. (20) and $\Delta_{\text{adj}}$ is the adjusted reward margin in Eq. (21).

**Discussions on the gradient coefficient.** Here we compare the gradient coefficients in Eqs. (33) and (36) and reveal several key advantages of our Uni-DPO over the standard DPO.

- **Difficulty-agnostic weighting in DPO.** The gradient coefficient $GC_\mathcal{A}$ in DPO depends solely on the implicit reward margin $\Delta_r$. As a result, it cannot fully account for the varying difficulty of the on-policy samples. Even when the reference model $\pi_{\text{ref}}$ exhibits a sufficiently large log-probability gap, DPO will continue to apply strong updates if $\Delta_r$ is not large enough, potentially risking overfitting. In contrast, Uni-DPO addresses this limitation by introducing a performance-weighting term $w_{\text{perf}}$ into the objective and the focal term $(1 - \sigma(\cdot))^\gamma$ into the gradient coefficient $GC$. This modification automatically downweights well-fitted examples and shifts the emphasis to more challenging cases.
- **Lack of quality-adaptive scaling in DPO.** The Gradient Coefficient $GC_\mathcal{A}$ in DPO is largely determined by the policy margin between positive and negative samples. The model's ability to distinguish preferences is inherently limited by its policy margin between positive and negative samples. Consequently, DPO struggles to effectively differentiate between high-quality (large

score margin in this work) and low-quality (small score margin) preference pairs, contradicting the principle that high-quality examples should exert a greater influence on learning, while low-quality pairs should induce more conservative policy updates. Uni-DPO resolves this by introducing an external quality judge $w_{\text{qual}}$ via a scaling function over expert-assigned margins $(S_w - S_l)$, which adaptively adjusts each pair's weight. This mechanism enables the model to modulate the learning signal based on sample quality, analogous to the use of proxy reward signals in PPO and GRPO, thus ensuring that higher quality examples are more likely to receive greater emphasis during policy optimization.

- **Uniform sample-level gradient magnitude.** In DPO, positive and negative samples in a pair share the same update strength $(i.e., GC_{\text{DPO}}^{y^+} = GC_{\text{DPO}}^{y^-})$, which overlooks the fact that different examples may need different levels of adjustment. PPO and GRPO solve this by assigning each sample its coefficient $GC_{\mathcal{A}}$ through advantage estimation. In Uni-DPO, we take a similar approach by adding a calibrated negative log-likelihood loss $\mathcal{L}_{\text{c-NLL}}$ applied selectively to high-quality, difficult positive samples. Thus, the incorporation of the c-NLL loss naturally enables the model to extract more information from high-quality positive samples and thereby enhance overall learning performance. Although this mechanism does not achieve the sample-level granularity of GRPO, it allows positive and negative samples to be assigned different weights based on their differing qualities, enabling the model to effectively leverage information from high-quality positive samples and thereby enhance overall learning performance.

### E.2    Impact of Length Normalization on Llama3-8B-Base

As demonstrated in Sec. 4.3, we observe that, without length normalization, Llama3-8B-Base exhibits minimal post-training gains on AlpacaEval2 and Arena-Hard compared to its instruct counterpart. In contrast, applying length normalization substantially reduces reward oscillations during training, resulting in significant performance improvements that surpass those achieved by DPO and SimPO. Specifically, we attribute this effect to two factors:

- **Weak initial preference alignment in the base model.** We observe that during training, Llama3-8B-Base assigns lower and more unstable probabilities to preferred samples $(y_w)$ compared to the instruct-tuned models. This suggests that it may not be well-calibrated to human preferences—an unsurprising result for a purely pre-trained model that was not trained with human preference data. As a consequence, the reward signal it receives is both weaker and noisier and may make it harder for the model to learn reliable preference signals during training.
- **Off-policy data training.** We strictly follow the experiment setups of SimPO, where the base model learns from off-policy samples (publicly available dataset UltraFeedback), whereas instruct-tuned models learn on-policy. We hypothesize that off-policy training makes learning more difficult—especially for a base model with weak initial preference alignment. Thus, without length normalization (LN), the model may struggle to learn true preference signals during training, leading to higher reward variance and weaker overall performance.

SimPO shows (in Sec. 4.2) that, without length normalization, models exploit length bias—producing ever-longer, but not necessarily higher-quality, outputs—whereas with length normalization, this bias is suppressed and models focus on true preference signals. This finding is consistent with our observations.

### E.3    Further Discussions on c-NLL Loss

In this section, we provide further insights into the design and effectiveness of the calibrated negative log-likelihood loss $(\mathcal{L}_{\text{c-NLL}})$ introduced in Sec. 3.4. We show that incorporating $\mathcal{L}_{\text{c-NLL}}$ does not lead to overfitting and also compare its behavior with the standard SFT loss as used in SimPO.

**Combining c-NLL won't cause overfit.**    The c-NLL loss is a corrective measure for the preference learning algorithm DPO's inherent tendency to shrink generation probabilities. It will not raise concerns about overfitting. Empirically, we strictly follow recent work, such as SimPO, to train the model exclusively on UltraFeedback (2023) and evaluate it on a broad suite of benchmarks, including SedarEval (2025) and Arena-Hard (2024), both released after the training data. Uni-DPO consistently outperforms previous methods on these unseen benchmarks, demonstrating robust generalization. Besides, we have evaluated Uni-DPO on both mathematical reasoning Tab. 2 and

multimodal tasks Tab. D.4, which demonstrate that Uni-DPO extends effectively to a wide range of domains and modalities. Furthermore, our ablation study in Tab. 3 shows that adding the c-NLL term improves performance across almost all benchmarks, confirming that c-NLL addresses a limitation of DPO without causing overfitting.

**Comparison of c-NLL and SFT loss introduced in SimPO.** SimPO reports in their Table 14 (Appendix G) that adding a generic SFT loss degrades performance on benchmark AlpacaEval2, whereas in our framework, incorporating the well-designed calibrated NLL (c-NLL) term improves results across modalities and benchmarks. We attribute this difference to two key factors:

- **Addressing DPO's inherent limitation.** We observed that, after standard DPO training, models tend to decrease the absolute generation probabilities of both positive ($y_w$) and negative samples ($y_l$), which can hinder their ability to learn and internalize human preferences. The proposed c-NLL can effectively address this issue by directly encouraging the model to generate human-preferred outputs. Our ablation study Tab. 3 clearly shows that adding c-NLL boosts model performance, confirming that it remedies the deficiency of vanilla DPO.
- **Being selective of high-quality, hard positive samples.** Unlike a uniform SFT loss used in SimPO, our well-designed c-NLL is applied only to those positive examples that are both high-quality and difficult for the model during training. As detailed in the ablation study (Appendix Table D.7), this selective strategy enables the model to extract more information from the most informative preference samples, thereby enhancing overall learning performance without over-emphasizing easier or lower-quality training data.

We note that the impact of SFT loss in preference optimization may vary depending on the training setup and evaluation task. Our results demonstrate that the proposed calibrated NLL (c-NLL) objective can better complement DPO.

### E.4 RATIONALE FOR RELAXING THE ORIGINAL FOCAL WEIGHT

We introduce focal weighting to adaptively emphasize under-fitted training samples based on the policy model's learning dynamics. However, the original focal factor imposes a rigid, per-sample constraint that can destabilize preference learning. We provide a more detailed explanation below.

**Limitations of the original focal weight.** As defined in *the first line* of Eq. (7), the focal weight dynamically increases when the policy model $\pi_\theta$ underperforms on a poorly-performed sample (*i.e.,* when its generation probability for the positive sample $y_w$ is lower than the reference model's, or its generation probability for the negative sample $y_l$ is higher than the reference model's), encouraging the model to focus on such under-fitted examples.

However, when rewritten as *the second line* of Eq. (7), it becomes clear that this formulation may over-amplify the weight for samples on which the reference model already exhibits extreme log-probability differences. For such samples, regardless of how the policy model performs relative to the reference model (*i.e.,* whether these examples are poorly-performed for the model or not), this weighting will further widen the policy model's probability gap between positive and negative samples, potentially leading to overfitting on those cases, which may cause unstable training dynamics.

**Relaxed formulation with performance weighting.** To address this issue, we replace the reference-dependent margin with a fixed threshold, yielding $w_{\text{perf}}$ in Eq. (10). This modification regulates the margin subtracted from the policy's log-probability disparity, preventing extreme reference scores from dominating the update. As shown in Fig. 5b, this relaxation improves training stability.

**Preserving hard–easy balancing.** Although the margin is fixed via $\tau_{\text{ref}}$, the weight $w_{\text{perf}}$ still adapts to the policy model's per-sample log-probability margin between $y_w$ and $y_l$. Moreover, the multiplier $(1 - \sigma(\cdot))^\gamma$ is retained (shown in Eqs. (18) and (26)), ensuring that the relaxed formulation preserves the focal mechanism's ability to target difficult examples. The effect is thus *smoother but not weaker*, maintaining adaptive balancing between well-fitted and under-fitted cases.

**Effectiveness of performance weighting.** Performance weighting ($w_{\text{perf}}$) yields consistent gains across tasks. It stabilizes optimization (Fig. 5b), mitigates overfitting (Figs. 5c and 5d), and improves

both overall comparisons (Tab. 1) and ablation results (Tab. 3), demonstrating its necessity and effectiveness in preference learning. The unified RL analysis in Sec. E.1 further demonstrates that the performance weighting remains a theoretically sound addition to preference learning.

## F    RELATED WORK

### F.1    LARGE LANGUAGE MODELS

Recent rapid progress in large language models (LLMs) (Touvron et al., 2023; Achiam et al., 2023; AI@Meta, 2024b; Hurst et al., 2024; Yang et al., 2024a; AI@Meta, 2024a; Yang et al., 2025a; Brown et al., 2020; Team et al., 2024a; Anthropic, 2023b;a; 2024; Liu et al., 2024a; Guo et al., 2025; Li et al., 2025) has accelerated the path toward artificial general intelligence (AGI). In particular, combining extensive self-supervised pre-training on massive text corpora with subsequent high-quality supervised fine-tuning (SFT) has endowed LLMs with surprising emergent properties in real-world understanding. Building on these milestones, current efforts focus on refining the alignment between model outputs and human preferences, thereby extending the utility of LLMs across a wider array of real-world applications (Ouyang et al., 2022).

In recent years, vision-language models (VLMs) have achieved remarkable progress (Radford et al., 2021; Shao et al., 2024a; Wang et al., 2025; Tian et al., 2019; Liu et al., 2024d; Yang et al., 2024d; Team et al., 2025). With the advancement of LLMs, multimodal large language models (MLLMs) have achieved remarkable alignment of visual and textual representations through cross-modal feature integration, marking a crucial milestone toward truly general-purpose AI systems (Wang et al., 2024d; Bai et al., 2023; 2025; Liu et al., 2023; 2024b;c; Dai et al., 2023; OpenAI, 2023; Zhu et al., 2023; Qu et al., 2025; Yang et al., 2023b; Zhong et al., 2024; Yang et al., 2023a; 2024c; Lai et al., 2024a; Peng et al., 2025; Hou et al., 2026). However, the challenge of building models that are useful, reliable, and harmless remains a central concern. In this paper, we propose a new algorithm, Uni-DPO, to advance the reliability of MLLMs.

### F.2    REINFORCEMENT LEARNING FROM HUMAN FEEDBACK

RLHF is a powerful paradigm to align LLMs with human judgments and values (Christiano et al., 2017; Ziegler et al., 2019; Kaufmann et al., 2023; Stiennon et al., 2020; Touvron et al., 2023; Wang et al., 2024a; Ouyang et al., 2022; Bai et al., 2022). In the classical RLHF workflow, a reward model is first trained on human preference data (Gao et al., 2023b; Luo et al., 2023; Chen et al., 2024b; Lightman et al., 2023; Havrilla et al., 2024b; Lambert et al., 2024), and this learned reward is then used to optimize the policy, typically via reinforcement learning algorithms such as PPO (Schulman et al., 2017; Ouyang et al., 2022; Ramamurthy et al., 2022; Anthony et al., 2017). RLHF has demonstrated its utility in a wide range of settings, from toxicity mitigation (Amini et al., 2024; Korbak et al., 2023; Zheng et al., 2023a; Li et al., 2024b; Chaudhary et al., 2024) to enhancing helpfulness (Tian et al., 2024; Wang et al., 2024b; Tan et al., 2025), reducing hallucination phenomena (Peng et al., 2025; Zhao et al., 2023; Xiao et al., 2024; Zhou et al., 2024b; Yu et al., 2024; He et al., 2024b), and bolstering reasoning skills (Zhang et al., 2025a; Pang et al., 2024; Havrilla et al., 2024a; Lai et al., 2024b; Yang et al., 2025b).

### F.3    DIRECT PREFERENCE OPTIMIZATION

**DPO.**    Although RLHF has been successfully applied across many domains, online preference optimization algorithms are often complex, hard to optimize efficiently, and may compromise the stability and reliability of the learned reward model (Casper et al., 2023; Gao et al., 2023b; Chaudhari et al., 2024; Zheng et al., 2023c; Santacroce et al., 2023). This has motivated the development of simpler offline alternatives such as Direct Preference Optimization (DPO) (Rafailov et al., 2023). DPO dispenses with an explicit reward model and instead learns a policy directly from the annotated preference data. However, without an explicit reward model, DPO cannot fully utilize preference pairs by the improved policy after several steps of training (Pan et al., 2025; Gorbatovski et al., 2024; Liu et al., 2024e; Gao et al., 2023a; Kong et al., 2024). To address this limitation, several works adopt an iterative scheme in which the reference policy is periodically replaced with the latest policy parameters or fresh preference pairs are sampled at each iteration (Dong et al., 2024;

Table F.1: **Comparison of Uni-DPO with $\beta$-DPO and D$^2$PO**. Reported results for $\beta$-DPO and D$^2$PO are taken from their original papers. Uni-DPO consistently achieves superior performance across benchmarks, demonstrating its effectiveness.

| Model | Method | AlpacaEval2 | | Arena-Hard | IFEval | | SedarEval |
|-------|--------|------|------|------|------|------|------|
| | | LC(%) | WR(%) | WR(%) | Strict Acc.(%) | Loose Acc.(%) | Overall(%) |
| Llama3-8B Instruct | SFT | 28.5 | 28.4 | 25.7 | 59.4 | 62.2 | 40.93 |
| | DPO | 43.7 | 41.7 | 32.6 | - | - | - |
| | SimPO | 44.7 | 40.4 | 33.8 | 60.8 | 68.6 | 44.81 |
| | $\beta$-DPO | 43.4 | 38.2 | - | - | - | - |
| | D$^2$PO | 43.0 | 43.5 | - | 65.6 | - | - |
| | Uni-DPO | **47.2** | **44.2** | **37.3** | **66.9** | **73.2** | 46.50 |

Kim et al., 2024; Rosset et al., 2024; Xiong et al., 2024; Yuan et al., 2024). In contrast, Uni-DPO incorporates a dual-adaptive weighting mechanism that dynamically accounts for both the model's learning dynamics and the effectiveness of training data. Consequently, it runs entirely in a single pass without any iterative retraining and achieves superior performance (see Sec. D.2).

**Limitations of standard DPO.** DPO has emerged as a cornerstone technique within RLHF due to its simplicity and efficiency. However, several limitations of DPO have been identified. For example, DPO can reduce the probabilities of preferred samples $y_w$ (Feng et al., 2024; Rafailov et al., 2024), leading LLMs to struggle with generating human-preferred outputs. To mitigate this issue, Pal et al. (2024); Pang et al. (2024) propose incorporating a negative log-likelihood loss to balance learning. Building on this idea, Uni-DPO applies a calibrated NLL term selectively to high-quality and challenging positive samples, enabling more focused updates and yielding superior performance.

**Length bias and normalization.** Another line of research highlights that DPO could induce a length bias, where the policy model tends to favor longer sequences (Meng et al., 2024; Park et al., 2024; Lu et al., 2024; Dubois et al., 2024; Singhal et al., 2023; Wang et al., 2023). Length normalization (LN) has been shown to alleviate this problem. Uni-DPO integrates LN into the training objective, and our ablation experiments confirm its effectiveness in improving training stability.

**Adaptive weighting strategies.** Beyond these refinements, other DPO variants explore different weighting strategies. For instance, $\beta$-DPO (Wu et al., 2024b) introduces a dynamic weighting factor to rebalance training samples by applying it to both sample filtering and loss scaling. Extending this idea further, Uni-DPO unifies the off-policy aspect (intrinsic data quality) and the on-policy aspect (model learning dynamics) within a single adaptive weighting scheme. Similarly, D$^2$PO (Shao et al., 2025) argues that uniformly weighting all tokens underestimates the fact that earlier tokens contribute more to sequence-level preference optimization. Uni-DPO generalizes this principle to the sequence level, allowing each example's weight to adapt jointly to its judge-assigned quality and its difficulty for the current policy. The comparison results in Tab. F.1 demonstrate that Uni-DPO consistently outperforms these variants across different models and benchmarks.

Besides, several recent works propose improvements to DPO on sample-wise weighting. RDO (Wang et al., 2024e) observes that current offline RLHF mainly captures the ordinal relationship between responses, while overlooking how much one response is preferred over another. Thus, RDO directly predicts a differential score between two responses and uses this margin to more precisely guide the alignment. WPO (Zhou et al., 2024a) reweights preference pairs according to their likelihood under the policy, so that off-policy data are adjusted to better match the on-policy distribution. MWPO (Xu et al., 2025b) proposes a DPO-based multi-weighting scheme that jointly considers implicit reward margins and response length margins. These two factors are geometrically combined into synthetic weights, which are then used to rescale preference pairs for optimization. RPO (Yin et al., 2024) introduces a contrastive weighting mechanism that allows the model to train on a broader range of preference data, including both paired and unpaired examples.

**Comparison with SimPO.** Among recent variants, SimPO (Meng et al., 2024) stands out for eliminating the reference model, thereby achieving improved training efficiency and strong performance. Our Uni-DPO combines fine-grained data utilization, length normalization, and unified weighting into a coherent framework, leading to further gains across diverse tasks.

## G    CASE STUDIES

This section presents two types of studies illustrating the effectiveness of Uni-DPO: one in textual understanding, which demonstrates the model's ability to generate coherent, instruction-following responses, as illustrated in Figs. G.1 to G.4, and one in mathematical reasoning tasks, which showcases its proficiency in solving complex problems, as shown in Figs. G.5 to G.7.

## H    BROADER IMPACT

This work contributes to the advancement of preference learning in large language models. While the proposed methods may have broader societal implications, we believe the primary impact lies in improving the alignment between model outputs and human preferences. We have carefully considered potential concerns and have not identified specific issues that require immediate attention beyond standard considerations for language model deployment.

## I    LLM USAGE STATEMENT

In this work, large language models (LLMs) were used as general-purpose writing assistants. Specifically, we employed LLMs to improve the grammar and readability of the manuscript through language polishing and minor rephrasing.

Figure G.1: An example Arena-Hard (Li et al., 2024a) response generated by the Qwen2.5-7B (Yang et al., 2024a) model after preference learning. The outputs are compared against those of the SimPO-trained model and the proprietary GPT-4 model.

| | |
|---|---|
| **Input** → | Q: Write a function in Scheme that reverses a list of strings? |

**SimPO** → You can write a function in Scheme to reverse a list of strings using recursion. Here's an example:

```scheme
(define (reverse-list lst)
  (if (null? lst)
    '()
    (append (reverse-list (cdr lst)) (list (car lst)))))
```

This function takes a list `lst` as input and recursively appends the car (first element) of the list to the reversed rest of the list. When the list is empty, it returns an empty list.

You can use this function like this:

```scheme
(reverse-list '(hello world)) ; Output: (world hello)
```

**Uni-DPO** → Here's a Scheme function that reverses a list of strings:

```scheme
(define (reverse-list lst)
  (if (null? lst)
    '()
    (append (reverse-list (cdr lst)) (list (car lst)))))
```

**However, this function uses `append`, which is not efficient for reversing lists, as it creates new lists and copies elements. A more efficient approach would be to use recursion with an accumulator:**

```scheme
(define (reverse-list lst)
  (define (reverse-helper lst acc)
    (if (null? lst)
      acc
      (reverse-helper (cdr lst) (cons (car lst) acc))))
  (reverse-helper lst '()))
```

This function uses an inner helper function `reverse-helper` that takes the original list and an accumulator list. **It recursively processes the input list, adding each element to the front of the accumulator list.** Finally, it returns the reversed list when the input list is empty.

**GPT-4** → Here's a function in Scheme that reverses a list of strings:

```scheme
(define (reverse-list lst)
  (if (null? lst)
    '()
    (append (reverse-list (cdr lst)) (list (car lst)))))

(define (reverse-string-list string-list)
  (reverse-list string-list))
```

You can use the `reverse-string-list` function to reverse a list of strings. The `reverse-list` function is a general function that reverses any list, and `reverse-string-list` is a specific function for reversing a list of strings.

Example usage:

```scheme
(reverse-string-list '("hello" "world" "scheme"))
; Output: ("scheme" "world" "hello")
```

Figure G.2: An example Arena-Hard response generated by the Qwen2.5-7B model after preference learning. The outputs are compared against those of the SimPO-trained model and the proprietary GPT-4 model.

| | |
|---|---|
| **Input** → | Q: Write a python function to find the nth occurrence of a specific character in a string. |

**SimPO** →

You can create a Python function to find the nth occurrence of a specific character in a string using the following approach:

```python
def find_nth_occurrence(string, char, n):
    count = 0
    for i in range(len(string)):
        if string[i] == char:
            count += 1
            if count == n:
                return i
    return -1 # Return -1 if the character doesn't occur n times

# Example usage:
string = "Hello World!"
char = "o"
n = 2
result = find_nth_occurrence(string, char, n)
print(f"The {n}th occurrence of '{char}' is at index {result}")
```

This function iterates through the string, counting occurrences of the specified character. When it reaches the nth occurrence, it returns the index. If the character does not occur n times, it returns -1.

**Uni-DPO** →

Here's a Python function that finds the nth occurrence of a specific character in a string:

```python
def find_nth_occurrence(string, char, n):
count = 0
    for i, c in enumerate(string):
        if c == char:
            count += 1
            if count == n:
                return i
    return -1 # Return -1 if the character is not found n times

# Example usage:
string = "Hello, World!"
char = "o"
n = 2

index = find_nth_occurrence(string, char, n)
if index != -1:
    print(f"The {n}th occurrence of '{char}' is at index {index}.")
else:
    print(f"The character '{char}' does not occur {n} times in the string.")
```

This function iterates through the string, counting the occurrences of the specified character. When the nth occurrence is found, it returns the index. **If the character is not found n times**, it returns -1.

**GPT-4** →

Here's a Python function that finds the nth occurrence of a specific character in a string:

```python
def find_nth_occurrence(character, string, n):
    count = 0
    for index, char in enumerate(string):
        if char == character:
            count += 1
            if count == n:
                return index
    return -1

# Example usage:
string = "hello world, how are you?"
character = 'o'
n = 3
result = find_nth_occurrence(character, string, n)
print(f"The {n}rd occurrence of '{character}' is at index {result}.")
```

This function takes three arguments:...  (Omitted for brevity.)

Figure G.3: An example Arena-Hard (Li et al., 2024a) response generated by Llama3-8B-Base (AI@Meta, 2024b) after preference learning. The outputs are compared against those of the SimPO-trained model and the proprietary GPT-4 model.

| | |
|---|---|
| **Input** → | Q: How do i do a tuple comprehension in python? |

**SimPO** → **Tuple comprehensions are not available in Python, as they are specific to Python 3.8 and later versions.**

In Python, you can create tuples using list comprehensions and then convert them to tuples using the `tuple()` function.

Here's an example:

```python
# Example:  Creating a tuple of squares from a list of numbers

numbers = [1, 2, 3, 4, 5]

# Using list comprehension

squares_list = [x**2 for x in numbers]

# Converting the list comprehension result to a tuple
squares_tuple = tuple(squares_list)

print(squares_tuple) # Output:  (1, 4, 9, 16, 25)
```

In this example, we first create a list comprehension that calculates the squares of each number in the `numbers` list.  Then, we convert this list comprehension result to a tuple using the `tuple()` function.

**Note that tuple comprehensions are not available in Python versions prior to 3.8, but you can achieve similar functionality using a list comprehensions and converting them to tuples as shown above.**

**Uni-DPO** → **Python does not have tuple comprehensions, but you can use a generator expression to create a tuple.** Here's an example:

```python
# Example:  Create a tuple of squares from a range of numbers

numbers = range(1, 6)
square_tuple = tuple(x**2 for x in numbers)

print(square_tuple) # Output:  (1, 4, 9, 16, 25)
```

In this example, we use a generator expression `(x**2 for x in numbers)` to calculate the squares of each number in the `numbers` iterable, and then convert the resulting generator object to a tuple using the `tuple()` function.

**Note that tuple comprehensions are not a feature in Python, but generator expressions can be used to achieve similar results.** If you want to create a tuple with a simple transformation, generator expressions are a concise and readable way to do it.

**GPT-4** → In Python, you can create a tuple comprehension using a generator expression inside a tuple constructor.  Here's an example:

```python
# Generator expression inside a tuple constructor
tuple_comprehension = tuple(x * 2 for x in range(5))

print(tuple_comprehension) # Output:  (0, 2, 4, 6, 8)
```

In this example, we create a tuple comprehension that contains the double of each number in the range from 0 to 4.  The generator expression `x * 2 for x in range(5)` is used to generate the doubled values, and the `tuple()` constructor is used to convert the generator expression into a tuple.

Figure G.4: An example Arena-Hard (Li et al., 2024a) response generated by the Llama3-8B-Instruct (AI@Meta, 2024b) model after preference learning. The outputs are compared against those of the baseline model.

---

**Input →** Q:I have part of a Javascript function that I want to rewrite. Currently it searches every property Matches to find the minimum, and makes Player2 always be the first member. Instead, I want Player1 to be the lowest result sorting by Matches, and Player2 to be random each time the code is run.

```
function elo(data) {
  // Find the two players with the fewest matches.
  let minMatches = Number.MAX_SAFE_INTEGER;
  let Player1 = null;
  let Player2 = null;
  for (let player of data) {
    if (player.Matches < minMatches) {
      minMatches = player.Matches;
      Player1 = player;
      Player2 = data.find(p => p !== Player1);
    }
  }
}
```

---

**Baseline →** Here's a revised version of your function that sorts the players by the number of matches and then assigns Player1 as the player with the fewest matches and Player2 as a random player from the rest:

```javascript
function elo(data) {
  // Sort the players by the number of matches
  data.sort((a, b) => a.Matches - b.Matches);

  // Assign Player1 as the player with the fewest matches
  let Player1 = data[0];

  // Assign Player2 as a random player from the rest
  let Player2 = data[Math.floor(Math.random() * (data.length - 1)) + 1];

  // Make sure Player2 is not the same as Player1
  while (Player2 === Player1) {
    Player2 = data[Math.floor(Math.random() * (data.length - 1)) + 1];
  }
}
```

This function first sorts the players by the number of matches in ascending order. Then it assigns the first player in the sorted array as Player1. It assigns a random player from the rest of the array as Player2, making sure that Player2 is not the same as Player1.

---

**Uni-DPO →** Here's the rewritten function:

```javascript
function elo(data) {
  // Find the player with the fewest matches (Player1)
  let minMatches = Math.min(...data.map(player => player.Matches));
  let Player1 = data.find(player => player.Matches === minMatches);
  // Randomly select Player2 from the remaining players
  let playerOptions = data.filter(p => p !== Player1);
  let Player2=playerOptions[Math.floor(Math.random()*playerOptions.length)];
}
```

Here's what's changed:

1. Instead of iterating through the data to find the minimum `Matches` value, **we use the `Math.min()` function with the spread operator (`...`) to find the minimum value in a concise way.**
2. **We use `data.find()` to retrieve the player with the minimum `Matches` value**, which becomes `Player1`.
3. To select `Player2` randomly, we filter out `Player1` from the original data array **using `data.filter()`**, and then **use `Math.random()` to select an index from the remaining players**. We convert the random index to an integer **using `Math.floor()`** and use it to access the corresponding player from the filtered array.
Note that this implementation assumes that `data` is an array of objects with a `Matches` property, and there are at least two players in the data. If these assumptions don't hold, you may need to add additional error handling or checks.

---

Figure G.5: An example AIME2024 response generated by the Qwen2.5-Math-1.5B (Yang et al., 2024b) model after preference learning. The outputs are compared against those of the SimPO-trained model.

| Input → | Q: Jen enters a lottery by picking 4 distinct numbers from $S = \{1, 2, \cdots, 9, 10\}$. 4 numbers are randomly chosen from $S$. She wins a prize if at least two of her numbers were 2 of the randomly chosen numbers, and wins the grand prize if all four of her numbers are the randomly chosen numbers. The probability of her winning the grand prize given that she won a prize is $\frac{m}{n}$ where $m$ and $n$ are relatively prime positive integers. Find $m+n$. |
|---|---|
| SimPO → | To solve this problem, we need to calculate the probability of Jen winning the grand prize given that she won a prize. We can break this down into the following steps:
1. Calculate the total number of ways to choose 4 distinct numbers from the set $S = \{1, 2, 3, \ldots, 10\}$.
2. Calculate the number of ways to win a prize (i.e., at least 2 of her numbers are among the 4 randomly chosen numbers).
3. Calculate the number of ways to win the grand prize (i.e., all 4 of her numbers are among the 4 randomly chosen numbers).
4. Use the conditional probability formula to find the probability of winning the grand prize given that she won a prize.
Let's start by calculating the total number of ways to choose 4 distinct numbers from the set $S$. This is given by the combination formula $\binom{10}{4}$.
Next, we need to calculate the number of ways to win a prize. This can be done by considering the cases where exactly 2, 3, or 4 of her numbers are among the 4 randomly chosen numbers.
Finally, we need to calculate the number of ways to win the grand prize, which is the case where all 4 of her numbers are among the 4 randomly chosen numbers. This is given by the combination formula $\binom{4}{4}$.
...(Omitted for brevity.)
The probability of Jen winning the grand prize given that she won a prize is $\frac{1}{11}$. Therefore, the values of $m$ and $n$ are 1 and 11, respectively. The sum $m+n$ is $1+11=12$.
Thus, the final answer is $\boxed{12}$. |
| Uni-DPO → | To solve this problem, we need to calculate the probability of Jen winning the grand prize given that she won a prize. We can break this down into the following steps:
1. Calculate the total number of ways to choose 4 distinct numbers from the set $S = \{1, 2, 3, \ldots, 10\}$.
2. Calculate the number of ways to win a prize (i.e., at least 2 of her numbers match the 4 randomly chosen numbers).
3. Calculate the number of ways to win the grand prize (i.e., all 4 of her numbers match the 4 randomly chosen numbers).
4. Use the conditional probability formula to find the probability of winning the grand prize given that she won a prize.
Let's start with the calculations:
1. The total number of ways to choose 4 distinct numbers from 10 is given by the combination formula $\binom{10}{4}$:
$$\binom{10}{4} = \frac{10!}{4!(10-4)!} = 210$$
2. To calculate the number of ways to win a prize, we need to consider the cases where exactly 2, 3, or 4 of her numbers match the 4 randomly chosen numbers:
– Exactly 2 matches: Choose 2 from her 4 numbers and 2 from the remaining 6 numbers.
$$\binom{4}{2} \cdot \binom{6}{2} = 6 \cdot 15 = 90$$
– Exactly 3 matches: Choose 3 from her 4 numbers and 1 from the remaining 6 numbers.
$$\binom{4}{3} \cdot \binom{6}{1} = 4 \cdot 6 = 24$$
– Exactly 4 matches: Choose all 4 from her 4 numbers.
$$\binom{4}{4} \cdot \binom{6}{0} = 1 \cdot 1 = 1$$
So, the total number of ways to win a prize is:
$$90 + 24 + 1 = 115$$
3. The number of ways to win the grand prize is the number of ways to choose all 4 of her numbers to match the 4 randomly chosen numbers, which is:
$$\binom{4}{4} \cdot \binom{6}{0} = 1 \cdot 1 = 1$$
4. The probability of winning the grand prize given that she won a prize is the ratio of the number of ways to win the grand prize to the number of ways to win a prize:
$$P(\text{grand prize} \,|\, \text{prize}) = \frac{1}{115}$$
So, the probability is $\frac{1}{115}$, and the values of $m$ and $n$ are 1 and 115, respectively. Therefore, $m+n = 1+115 = 116$.
The final answer is $\boxed{116}$. |

Figure G.6: An example AMC2023 response generated by the Qwen2.5-Math-1.5B (Yang et al., 2024b) model after preference learning. The outputs are compared against those of the SimPO-trained model.

| | |
|---|---|
| **Input** $\rightarrow$ | Q: Mrs. Jones is pouring orange juice into four identical glasses for her four sons. She fills the first three glasses completely but runs out of juice when the fourth glass is only $\frac{1}{3}$ full. What fraction of a glass must Mrs. Jones poured from each of the first three glasses into the fourth glass so that all four glasses will have the same amount of juice? The final answer can be written in the form $\frac{m}{n}$, where $m$ and $n$ are relatively prime positive integers. What is $m + n$? |
| **SimPO** $\rightarrow$ | To solve this problem, we need to determine how much juice Mrs. Jones must pour from each of the first three glasses into the fourth glass so that all four glasses have the same amount of juice. 1. Let's assume each glass has a capacity of 1 unit of juice. 2. The first three glasses are completely filled, so each of them has 1 unit of juice. 3. The fourth glass is only $\frac{1}{3}$ full, so it has $\frac{1}{3}$ unit of juice. 4. The total amount of juice is $1 + 1 + 1 + \frac{1}{3} = 3 + \frac{1}{3} = \frac{10}{3}$ units. 5. Since there are four glasses, each glass should have $\frac{10}{3} \div 4 = \frac{10}{12} = \frac{5}{6}$ units of juice. 6. The fourth glass currently has $\frac{1}{3}$ unit of juice, so it needs $\frac{5}{6} - \frac{1}{3} = \frac{5}{6} - \frac{2}{6} = \frac{3}{6} = \frac{1}{2}$ unit of juice to reach $\frac{5}{6}$ units. 7. Since the first three glasses are completely filled, each of them needs to lose $\frac{1}{2}$ unit of juice to the fourth glass. So, Mrs. Jones must pour $\frac{1}{2}$ unit of juice from each of the first three glasses into the fourth glass. The fraction of a glass that must be poured from each of the first three glasses into the fourth glass is $\frac{1}{2}$. The final answer is $\frac{1}{2}$, and in the form $\frac{m}{n}$, we have $m = 1$ and $n = 2$. Therefore, $m + n = 1 + 2 = 3$.

The final answer is $\boxed{3}$. |
| **Uni-DPO** $\rightarrow$ | To solve this problem, we need to determine how much juice Mrs. Jones must pour from each of the first three glasses into the fourth glass so that all four glasses have the same amount of juice. 1. Let's assume each glass has a capacity of 1 unit of juice. 2. The first three glasses are completely filled, so each of them has 1 unit of juice. 3. The fourth glass is only $\frac{1}{3}$ full, so it has $\frac{1}{3}$ unit of juice. 4. The total amount of juice is $1 + 1 + 1 + \frac{1}{3} = 3 + \frac{1}{3} = \frac{10}{3}$ units. 5. Since there are four glasses, each glass should have $\frac{10}{3} \div 4 = \frac{10}{12} = \frac{5}{6}$ units of juice. 6. The fourth glass currently has $\frac{1}{3}$ unit of juice, so it needs $\frac{5}{6} - \frac{1}{3} = \frac{5}{6} - \frac{2}{6} = \frac{3}{6} = \frac{1}{2}$ unit of juice to reach $\frac{5}{6}$ units. 7. Since the juice is being poured from the first three glasses into the fourth glass, each of the first three glasses must lose $\frac{1}{2} \div 3 = \frac{1}{6}$ unit of juice. So, Mrs. Jones must pour $\frac{1}{6}$ unit of juice from each of the first three glasses into the fourth glass. The fraction of a glass that must be poured from each of the first three glasses into the fourth glass is $\frac{1}{6}$. The final answer is $\frac{1}{6}$, and $m + n = 1 + 6 = 7$.
The final answer is $\boxed{7}$. |

Figure G.7: An example AIME2024 response generated by the Qwen2.5-Math-7B (Yang et al., 2024b) model after preference learning. The outputs are compared against those of the SimPO-trained model.

---

**Input** → Q: A list of positive integers has the following properties:
· The sum of the items in the list is 30.
· The unique mode of the list is 9.
· The median of the list is a positive integer that does not appear in the list itself.
Find the sum of the squares of all the items in the list.

---

**SimPO** → Given these constraints, let's start by considering the possible lengths of the list. Since 9 is the unique mode, it must appear at least twice. If the list has an odd number of elements, the median will be the middle element, and it cannot be 9. If the list has an even number of elements, the median will be the average of the two middle elements, and again, it cannot be 9.
Let's start by considering the smallest possible list that satisfies these conditions. The list must have at least 5 elements (since 9 must appear at least twice and the median cannot be 9). Let's try a list with 5 elements: [9, 9, a, b, c]. The sum of these elements must be 30, so 9 + 9 + a + b + c = 30, which simplifies to a + b + c = 12. The median of this list is the third element when sorted, so it must be one of a, b, or c, and it cannot be 9. Let's try a = 1, b = 2, and c = 9. This gives us the list [1, 2, 9, 9, 9], but 9 is the mode, and the median is 9, which is not allowed. So, let's try another combination.

...(Omitted for brevity.)

Let's try a list with 7 elements: [9, 9, a, b, c, d, e]. The sum of these elements must be 30, so 9 + 9 + a + b + c + d + e = 30, which simplifies to a + b + c + d + e = 12. The median of this list is the fourth element when sorted, so it must be one of the numbers in the list, and it cannot be 9. Let's try a = 1, b = 2, c = 3, d = 4, and e = 2. This gives us the list [1, 2, 2, 3, 4, 9, 9], and the median is 3, which is not 9 and appears only once. This list satisfies all the conditions. **Comment: 9 is not the unique mode.**
Now, let's calculate the sum of the squares of all the items in the list: $1^2 + 2^2 + 2^2 + 3^2 + 4^2 + 9^2 + 9^2 = 1 + 4 + 4 + 9 + 16 + 81 + 81 = 196$.

The sum of the squares of all the items in the list is $\boxed{196}$.

---

**Uni-DPO** → Let's denote the list of integers as $a_1, a_2, \ldots, a_n$, where $a_i$ are positive integers and $n$ is the number of elements in the list. Since the mode is 9 and it is unique, the number 9 must appear more frequently than any other number in the list. Let's assume that 9 appears $k$ times in the list, where $k \geq 2$.
Since the median is a positive integer that does not appear in the list itself, the list must have an even number of elements (otherwise, the median would be the middle element, which would appear in the list). Let's assume the list has $2m$ elements. The median would then be the average of the $m$-th and $(m + 1)$-th elements when the list is sorted in ascending order. Since the median does not appear in the list, it must be a different number.
Let's start by considering the simplest case where the list has 4 elements. The median would be the average of the 2nd and 3rd elements. Since the median does not appear in the list, it must be a different number. Let's denote the list as $a, b, c, d$ in ascending order. The median would be $\frac{b+c}{2}$, and it must be a different number from $a, b, c, d$.
Given that the sum of the elements is 30 and the mode is 9, let's assume the list is $a, b, 9, 9$. The sum of these elements is $a + b + 9 + 9 = 30$, so $a + b = 12$. The median would be $\frac{b+9}{2}$, and it must be a different number from $a, b, 9$. Let's try different values for $a$ and $b$.
If $a = 3$ and $b = 9$, the list would be $3, 9, 9, 9$, but the mode would not be unique. If $a = 4$ and $b = 8$, the list would be $4, 8, 9, 9$, and the median would be $\frac{8+9}{2} = 8.5$, which is not an integer. If $a = 5$ and $b = 7$, the list would be $5, 7, 9, 9$, and the median would be $\frac{7+9}{2} = 8$, which is a different number and does not appear in the list.
So, the list $5, 7, 9, 9$ satisfies all the conditions. The sum of the squares of the elements in the list is $5^2 + 7^2 + 9^2 + 9^2 = 25 + 49 + 81 + 81 = 236$.

The sum of the squares of all the items in the list is $\boxed{236}$.

---

