# OpenReview forum: "Uni-DPO: A Unified Paradigm for Dynamic Preference Optimization of LLMs"
_ICLR.cc/2026/Conference — ICLR 2026 Poster_

### Official Review · Reviewer_HSvf · 2025-10-27

**Soundness:** 3
**Presentation:** 3
**Contribution:** 3
**Rating:** 6
**Confidence:** 4

**Summary:**

The paper develops what’s called a “unified dynamic preference approach”, UNI-DPO, aiming to improve extend the basic DPO. Some key elements of UNI-DPO include: two weighting factors, w_qual and w_perf, which apply, respectively, to the preference pair (y_win, y_loss) and to the current policy \pi_\theta; the former takes into account data quality, and the latter measures how the policy aligns with the pair. Furthermore, a “calibrated negative log-likelihood loss” (c-NLL) term is added to the training objective, to amplify the policy’s confidence on y_win. These are elaborated in \S3.2, 3.3. and 3.4.

**Strengths:**

The main strength of the approach is to dynamically adjust each sample’s contribution based on both data quality and the model’s learning performance. This effectively incorporates ideas like advantage-based schemes in RL and thereby improves training efficiency and overall performance.

**Weaknesses:**

The weight w_qual defined in (5) is a bit simplistic in view of the complexity in defining (or capturing) what is data quality. This is also partially acknowledged by the paper (at the end).

**Questions:**

Can the author(s) comment on how does the dual weighting scheme in UNI-DPO compare with other approaches to preference alignment and reward optimization, and specifically the weighting factors motivated by importance sampling?

---

> ### Author Response · Authors · 2025-11-20
>
> We sincerely thank Reviewer **HSvf** for careful reading of our manuscript and for providing thoughtful, constructive feedback. In response, we have thoroughly revised the manuscript, with all modifications clearly highlighted in **blue** to facilitate review. Below, we address each of the reviewer’s comments in detail and explicitly indicate where the corresponding changes appear in the revised version. We deeply appreciate your valuable input and would be delighted to offer any further clarifications should additional questions or concerns arise.
>
> > **Response to Comment on the Simplicity of $w_{\text{qual}}$**
>
> We thank the reviewer for raising this point. Our design of $w_{\text{qual}}$ is intentionally simple and grounded in a practical modeling of data quality: it aims to (a) encode finer-grained preference strength and reliability, and (b) correct for imbalance across samples with heterogeneous quality.
>
> Despite its simplicity, this design has proven to be both intuitive and effective in practice. For example, Uni-DPO is evaluated with nearly identical hyperparameter configurations across multiple model families and parameter scales, and on a broad suite of benchmarks (4 text benchmarks, 8 math benchmarks, and 10 multimodal benchmarks).
>
> The downstream evaluation results in Appendix Table D.10 further show that Uni-DPO–trained models generalize well to unseen tasks and domains, suggesting that the current form of $w_{\text{qual}}$ already provides a robust and transferable quality-aware signal.
>
> Moreover, the ablation study in Table 3 (Main ablation results) explicitly demonstrates the effectiveness of $w_{\text{qual}}$: removing this component consistently leads to performance drops, confirming that even this relatively simple formulation plays a critical role in the overall gains of Uni-DPO.
>
> That said, we fully agree that data quality is inherently complex and can be modeled in more sophisticated ways (e.g., multi-dimensional quality scores, task-specific quality predictors, or dynamic quality estimates that evolve during training). We view our current $w_{\text{qual}}$ as a first, principled step toward integrating explicit quality modeling into DPO-style objectives. Exploring richer quality estimators and more advanced ways of injecting them into the training objective while maintaining stability and practicality constitutes an important direction for future work.

---

> ### Author Response · Authors · 2025-11-20
>
> > **Response to Question (a) (Comparison with Existing DPO Variants and Weighting Methods)**
>
> We appreciate the reviewer’s suggestion to more clearly position our method relative to existing work on data selection and adaptive weighting for DPO. We agree that situating Uni-DPO within this broader landscape helps clarify its contributions.
>
> First, we compare Uni-DPO with two closely related DPO variants:
>
> 1. $\beta$-DPO [a]
>
>    - Similarity. $\beta$-DPO introduces a dynamic weighting factor to rebalance training samples, using this factor for both sample filtering and loss scaling during training.
>    - Difference. Uni-DPO jointly models the _off-policy_ aspect (intrinsic data quality) and the _on-policy_ aspect (model learning dynamics) of preference optimization within a single, unified weighting scheme. In contrast, $\beta$-DPO’s dynamic weight is driven solely by the policy model itself, without explicitly capturing intrinsic data quality or judge scores.
>
> 2. D$^2$PO [b]
>    - Similarity. D$^2$PO observes that uniformly weighting all tokens underestimates the fact that earlier tokens often contribute more to sequence-level preference optimization, and thus reweights tokens within each example.
>    - Difference. While D$^2$PO refines _intra-example_ (token-level) weighting, it does not account for variability _between examples_ in terms of data quality and evolving policy performance. Uni-DPO extends this principle to the sequence level, allowing each example’s weight to adapt to both its judge-assigned quality and how difficult it is for the current policy.
>
> We have also provided comparisons with these variants. On AlpacaEval2 with Llama3-8B-Instruct, Uni-DPO outperforms SFT, DPO, SimPO, and $\beta$-DPO in both length-controlled and win-rate metrics. On IFEval with Llama3-8B-Instruct and Gemma-2-9B-it, Uni-DPO achieves higher scores than DPO, SimPO, and D$^2$PO across both model families. These results indicate that Uni-DPO provides consistent gains over existing DPO variants under comparable settings.
>
> |   Model & Method   | AlpacaEval2 |          |
> | :----------------: | :---------: | :------: |
> | Llama3-8B-Instruct |   LC (%)    |  WR (%)  |
> |        SFT         |    28.5     |   28.4   |
> |        DPO         |    43.7     |   41.7   |
> |       SimPO        |    44.7     |   40.4   |
> |    $\beta$-DPO     |    43.4     |   38.2   |
> |      Uni-DPO      |  **47.2**   | **44.2** |
>
> |       Model        |  Method  |  IFEval  |
> | :----------------: | :------: | :------: |
> | Llama3-8B-Instruct |   DPO    |   65.1   |
> |                    |  SimPO   |   60.8   |
> |                    | D$^2$PO  |   65.6   |
> |                    | Uni-DPO | **67.8** |
> |   Gemma-2-9b-it    |   DPO    |   60.1   |
> |                    |  SimPO   |   71.5   |
> |                    | D$^2$PO  |   71.2   |
> |                    | Uni-DPO | **72.8** |
>
> Beyond these, several recent methods study weighting or data selection in preference optimization (that are not fully open-sourced yet). These include:
>
> - **RDO [c]** observes that current offline RLHF mainly captures the ordinal relationship between responses, while overlooking how much one response is preferred over another. Thus, RDO directly predicts a differential score between two responses and uses this margin to guide training.
> - **WPO [d]** reweights preference pairs according to their likelihood under the current policy, so that off-policy data are adjusted to better match the on-policy distribution. This addresses the distribution mismatch and strengthens the optimization process without introducing extra training cost.
> - **MWPO [e]** proposes a DPO-based multi-weighting scheme that jointly considers implicit reward margins and response length margins. These two factors are geometrically combined into synthetic weights, which are then used to rescale preference pairs for optimization.
> - **RPO [f]** introduces a contrastive weighting mechanism that allows the model to train on a broader range of preference data, including both paired and unpaired examples, by assigning appropriate weights across different preference formats.
>
> In contrast, Uni-DPO **jointly integrates off-policy data quality signals and on-policy preference optimization dynamics** into a unified per-example adaptive correction mechanism. Our goal is to directly address DPO’s tendency to underutilize high-quality or difficult samples by adaptively amplifying or down-weighting examples based on their quality and model training dynamics. To the best of our knowledge, this dual perspective on preference optimization with per-sample adaptive corrections has not been explored in existing sample-wise weighting methods such as RDO, WPO, MWPO, or RPO. We have made these distinctions more explicit in the revised related work section.

---

> ### Author Response · Authors · 2025-11-20
>
> > **Response to Question (b) (Connection to Importance Sampling Mechanism)**
>
> We thank the reviewer for the question regarding the relationship between our weighting mechanism and importance sampling. While there is a superficial similarity in that both approaches assign weights to samples, their motivations and roles in the learning process are fundamentally different.
>
> In Uni-DPO, our weighting factors are designed to address two main aspects of preference optimization:
>
> 1. Data-quality–aware correction. The quality weight $w_{\text{qual}}$ explicitly models heterogeneous data quality (e.g., judge scores, reliability, or preference strength) across examples. Its goal is to upweight high-quality, trustworthy supervision signals and downweight noisy or ambiguous ones, so that the effective gradient is dominated by more reliable preference pairs.
>
> 2. Learning-dynamics–aware focusing. The difficulty / dynamics-related weight $w_{\text{perf}}$ adapts to the _current state of the policy_, emphasizing examples that are informative or challenging at a given training stage (e.g., where the model still struggles or is uncertain). This encourages the optimizer to focus capacity on under-learned or high-value samples, improving both efficiency and stability.
>
> Together, these weights form a per-example adaptive correction mechanism that shapes _which_ samples the model learns most from, based on their intrinsic quality and current difficulty, rather than on how they were sampled.
>
> By contrast, **importance sampling** is primarily concerned with correcting **distribution mismatch between the behavior policy and the target policy**. In classic off-policy RL or off-policy preference optimization, one samples data from a convenient behavior policy (e.g., an SFT or older policy) and then uses importance weights (e.g., likelihood ratios) to make the resulting estimator unbiased (or less biased) with respect to the target on-policy objective. The focus is on re-weighting due to the sampling distribution, so that an on-policy algorithm can be applied in an off-policy setting and data can be reused multiple times.
>
> In our case, Uni-DPO does not aim to reconstruct an on-policy estimator from off-policy samples in the importance-sampling sense. Instead:
>
> - Our weights are not derived from a likelihood ratio between a behavior policy and the current policy, but from _data-centric_ and _learning-centric_ signals (quality scores and model performance).
> - The objective is not to correct for sampling bias, but to **shape the supervision signal** so that high-quality and currently informative examples contribute more strongly to optimization.
>
> That said, both perspectives are complementary: importance sampling addresses _where the data come from_ (distributional mismatch), whereas Uni-DPO’s weighting addresses _how useful and reliable each sample is_ for alignment at a given training stage.
>
> &nbsp;
>
> [a] Official code of $\beta$-DPO: Direct Preference Optimization with Dynamic $\beta$, NeurIPS, 2024
>
> [b] Earlier tokens contribute more: Learning direct preference optimization from temporal decay perspective, ICLR, 2025
>
> [c] Reward Difference Optimization For Sample Reweighting In Offline RLHF, ACL, 2024
>
> [d] Enhancing RLHF with Weighted Preference Optimization, EMNLP, 2024
>
> [e] MWPO: Enhancing LLMs Performance through Multi-Weight Preference Strength and Length Optimization, ACL, 2025
>
> [f] Relative Preference Optimization: Enhancing LLM Alignment through Contrasting Responses across Identical and Diverse Prompts, ArXiv, 2025

---

> ### Author Response · Authors · 2025-11-28
>
> ## To Reviewer HSvf
>
> We again thank Reviewer **HSvf** for the careful reading and insightful comments. In our latest revision, we have:
>
> - Clarified the design motivation and role of $w_{\text{qual}}$, and discussed its effectiveness and generality.
> - Positioned Uni-DPO more explicitly relative to existing DPO variants and weighting methods (e.g., $\beta$-DPO, D$^2$PO, RDO, WPO, MWPO, RPO), with corresponding empirical comparisons.
> - Explained in detail how our weighting mechanism differs conceptually and technically from classical importance sampling.
>
> All related modifications are highlighted in blue in the revised manuscript.
>
> &nbsp;
>
> If there are any remaining concerns or further points you would like us to clarify or explore, we would be very glad to continue the discussion and further refine the paper as needed.

---

### Official Review · Reviewer_Azer · 2025-10-30

**Soundness:** 2
**Presentation:** 3
**Contribution:** 2
**Rating:** 4
**Confidence:** 3

**Summary:**

This paper introduces Uni-DPO, which injects two jointly learned adaptive weights into single-pass offline training: 1) a quality weight derived from external score margins that up-weights high-confidence preference pairs, and 2) a performance weight computed from the current policy margin that down-weights already well-fitted samples while emphasizing those that remain hard. Additionally, to counteract DPO’s tendency to suppress the absolute likelihood of preferred responses, the authors devise a calibrated negative log-likelihood term that selectively reinforces difficult yet high-quality positive examples. The resulting objective preserves the simplicity of DPO but enables gradient allocation at a fine-grained level that accounts for both sample difficulty and data quality.

**Strengths:**

- This paper derives the gradient coefficient of Uni-DPO in closed form, explicitly integrating two modulation factors, including quality weight and performance weight, to provide a principled explanation for online sample re-weighting.
- Uni-DPO preserves the simplicity of single-stage offline training without introducing additional reward models or iterative sampling overhead. With only two learnable weights and a calibrated loss, it can achieve consistent performance improvements across diverse benchmarks.

**Weaknesses:**

- Although the paper proposes a "unified dynamic weighting paradigm", its core idea essentially combines a quality-aware weight with a performance-aware weight. Such sample reweighting concepts have already been extensively studied in machine learning and RLHF literature, including focal loss, curriculum learning, and advantage reweighting. Therefore, the conceptual novelty of Uni-DPO appears somewhat incremental rather than fundamentally new.

- A major concern lies in the lack of essential baselines. While the paper focuses on DPO-based alignment, it would be important to include comparisons with more recent RL-based alignment methods beyond PPO, since several advanced RL algorithms have recently demonstrated improved stability and performance in preference optimization. Moreover, to my knowledge, there already exist studies on data selection and adaptive weighting for DPO that explicitly aim to exploit heterogeneous data quality during training. Including these approaches as baselines would provide a fairer and more convincing empirical validation.

- Although Uni-DPO claims to "outperform SimPO", its improvement may come from the weighting strategy rather than a new optimization formulation. In contrast, SimPO already removes the reference model and introduces length normalization, while Uni-DPO still relies on the log-probability ratio between the policy and the reference model. This raises questions about whether the claimed superiority originates from a fundamentally different principle or just an additional heuristic weighting scheme.

- Uni-DPO is only evaluated on textual understanding and mathematical reasoning tasks, while multimodal experiments are mentioned in the abstract but not presented in the results section. The lack of evaluations on other important aspects such as dialogue safety, factual consistency, or instruction robustness weakens the "Unified" claim. Additionally, the models used for evaluation (e.g., LLaMA-3-8B, Gemma-2-9B) appear to be relatively early versions; including results on more recent model releases would strengthen the credibility and contemporaneity of the paper.

**Questions:**

Some suggestions here.

- The right-hand side of Figure 3 does not clearly or intuitively convey the core idea of the proposed method. Its meaning only becomes apparent after reading the entire paper and integrating multiple sections. It is recommended to redesign or annotate the figure to make the conceptual distinction between DPO and Uni-DPO more self-explanatory and visually interpretable.

- It would be helpful to include training curve comparisons (e.g., loss or evaluation score versus training steps) to further illustrate the differences in learning efficiency among Uni-DPO, DPO, and SimPO.

---

> ### Author Response · Authors · 2025-11-20
>
> We thank Reviewer **Azer** for carefully reading our manuscript and for the constructive and insightful comments. We have revised the manuscript accordingly, with all changes highlighted in **blue** for easy inspection. In this rebuttal, we address each comment in detail and point to the corresponding modifications. We appreciate your feedback and are ready to provide further clarifications if needed.

---

> ### Author Response · Authors · 2025-11-20
>
> > **Response to Weakness 1 (The concern on novelty and relation to existing methods)**
>
> We thank the reviewer for the assessment and for recognizing that our framework is “useful” and “valuable”. We agree that Uni-DPO builds on several established ideas. Our main contribution lies in a **principled, unified, and empirically validated integration** of these components to address a critical and previously underexplored limitation of DPO: its tendency to underutilize high-quality and challenging samples when trained on heterogeneous preference data.
>
> Importantly, Uni-DPO’s empirical gains over DPO and SimPO are **consistent and robust** across a wide range of settings, rather than the result of tuning for a specific case. Uni-DPO is evaluated with **nearly identical hyperparameter configurations** across multiple model families and parameter scales, and on a broad suite of benchmarks: 4 text benchmarks, 8 math benchmarks, and 10 multimodal benchmarks.
>
> 1. Directly combining existing ideas is not sufficient
>
> We would like to clarify that simply “plugging in” these existing techniques into standard DPO does **not** yield stable or effective preference learning. In early experiments, naive combinations of standard focal loss on DPO logits and a vanilla auxiliary NLL loss either led to unstable training or failed to deliver consistent gains over the baseline DPO/SimPO, as shown in the main paper Figure 5 (b-c). This observation is supported by both our theoretical analysis and the ablation studies (Table 3: Main ablation results and Table D.7).
>
> 2. Non-trivial synergy and new gradient perspective
>
> Although the high-level ingredients (NLL, length normalization, focal-like weighting) are familiar, their synergistic integration in Uni-DPO is non-trivial. Uni-DPO is explicitly derived and analyzed from an RL-gradient perspective (Appendix E.1), where we compare the Uni-DPO gradient with that of RLHF-style methods. This analysis guides _how_ we combine quality-aware weights and difficulty-aware modulation, rather than heuristically stacking components. The resulting framework shifts the viewpoint from treating all preferences as equal binary signals to a more nuanced quality-aware preference tuning paradigm.
>
> 3. Novel extension of focal-style weighting
>
> We appreciate the reviewer’s connection between our reweighting scheme and focal loss. However, Uni-DPO does not apply focal loss in its standard form:
>
> - Different signal: Classic focal loss reweights samples purely based on the model’s _confidence in a single-label prediction_. In contrast, Uni-DPO operates on preference pairs and reweights samples based on the _preference probability margin_ between preferred and dispreferred responses, together with external quality scores. This leads to a qualitatively different weighting signal.
> - Stabilization via fixed reference margin: Our initial empirical attempts using a naive focal-style term on the DPO logits were unstable and sometimes degraded performance. To address this, we introduced a fixed reference margin (\tau\_{\text{ref}}) in the weighting, which effectively anchors the focal modulation and stabilizes training. This design is specific to the DPO-style preference setting and, to our knowledge, has not appeared in prior adaptations.
>
> 4. Novel adaptation of NLL loss
>
> Similarly, our use of NLL is not a simple addition of a standard SFT loss. Prior works use a _uniform_ NLL/SFT term over all positive responses. In contrast, Uni-DPO uses a calibrated NLL that selectively emphasizes hard, high-quality positive samples. By combining the scalar quality score and the difficulty signal, Uni-DPO focuses the auxiliary NLL on those samples that are both reliable and informative, which significantly improves performance in our ablations (Appendix Table D.7).
>
> 5. Relation to curriculum learning and data filtering
>
> We also acknowledge the reviewer’s comparison to curriculum learning and quality-based data filtering. Uni-DPO is indeed related in spirit, as it prioritizes more reliable and informative samples. However, unlike traditional curriculum learning, which relies on hand-crafted, stage-wise schedules (e.g., easy-to-hard curricula defined a priori), Uni-DPO does not use a manually designed curriculum. Instead, the unified weighting schema dynamically adjusts each sample’s contribution based on model dynamics and data quality, making the “curriculum” inherently **model-driven** rather than manually staged.
>
> In summary, while Uni-DPO builds on known components, the way they are designed and combined, in particular the unified weighting over data quality and learning dynamics, the stabilized focal-style weighting with a fixed reference margin, and the calibrated NLL that focuses on hard, high-quality samples, go beyond a simple aggregation of prior heuristics. Our theory and experiments also demonstrate that naive combinations of these ideas are ineffective, underscoring the necessity and complementarity of our design choices.

---

> ### Author Response · Authors · 2025-11-20
>
> > **Response to Weakness 2(a) (Comparison with RL-based Alignment Methods)**
>
> We thank the reviewer for pointing out the need for more comprehensive baselines. In the revised version, we have updated the experiments and provide detailed comparisons in **Appendix Table D.5**, where we strictly follow the standard evaluation protocols provided by Language Model Evaluation Harness, and include a broader set of methods: **supervised fine-tuning (SFT) methods** (Eurus2-SFT, RAFT), **preference alignment methods** (DPO, SimPO, Iterative DPO), and **RL-based methods** (Eurus2-PRIME, OpenReasoner-Zero-7B, SimpleRL-Zero-7B, Dr. GRPO (Oat-Zero-7B), and PPO).
>
> The results show that Uni-DPO remains highly competitive, even against these stronger baselines. We first consider the Iterative DPO method proposed by Zhang et al. (2025a) on mathematical reasoning tasks, which adopts the same training setups and training datasets as Uni-DPO. Although Iterative DPO performs seven DPO iterations and uses about seven times more training data than Uni-DPO, it is still outperformed by Uni-DPO on almost all benchmarks as well as in terms of the final average score, highlighting the overall effectiveness and data efficiency of Uni-DPO.
>
> In addition, we compare Uni-DPO with **RL-based approaches** such as Eurus2-PRIME (Cui et al., 2025), OpenReasoner (Hu et al., 2025), SimpleRL (Zeng et al., 2025), Dr. GRPO (Oat-Zero-7B) (Liu et al., 2025), and PPO (Schulman et al., 2017). Uni-DPO outperforms Eurus2-PRIME, OpenReasoner, and SimpleRL on the overall performance, and achieves an overall score that is on par with the PPO-trained model, while enjoying a much simpler and more efficient training procedure.
>
> If the reviewer has more methods in mind that should be included as baselines, we would be happy to consider them for inclusion in future revisions.

---

> ### Author Response · Authors · 2025-11-20
>
> > **Response to Weakness 2(b) (Comparison with Existing DPO Variants and Weighting Methods)**
>
> We appreciate the reviewer’s suggestion to more clearly position our method relative to existing work on data selection and adaptive weighting for DPO. We agree that situating Uni-DPO within this broader landscape helps clarify its contributions.
>
> First, we compare Uni-DPO with two closely related DPO variants:
>
> 1. $\beta$-DPO [a]
>
>    - Similarity. $\beta$-DPO introduces a dynamic weighting factor to rebalance training samples, using this factor for both sample filtering and loss scaling during training.
>    - Difference. Uni-DPO jointly models the _off-policy_ aspect (intrinsic data quality) and the _on-policy_ aspect (model learning dynamics) of preference optimization within a single, unified weighting scheme. In contrast, $\beta$-DPO’s dynamic weight is driven solely by the policy model itself, without explicitly capturing intrinsic data quality or judge scores.
>
> 2. D$^2$PO [b]
>    - Similarity. D$^2$PO observes that uniformly weighting all tokens underestimates the fact that earlier tokens often contribute more to sequence-level preference optimization, and thus reweights tokens within each example.
>    - Difference. While D$^2$PO refines _intra-example_ (token-level) weighting, it does not account for variability _between examples_ in terms of data quality and evolving policy performance. Uni-DPO extends this principle to the sequence level, allowing each example’s weight to adapt to both its judge-assigned quality and how difficult it is for the current policy.
>
> We have also provided comparisons with these variants. On AlpacaEval2 with Llama3-8B-Instruct, Uni-DPO outperforms SFT, DPO, SimPO, and $\beta$-DPO in both length-controlled and win-rate metrics. On IFEval with Llama3-8B-Instruct and Gemma-2-9B-it, Uni-DPO achieves higher scores than DPO, SimPO, and D$^2$PO across both model families. These results indicate that Uni-DPO provides consistent gains over existing DPO variants under comparable settings.
>
> |   Model & Method   | AlpacaEval2 |          |
> | :----------------: | :---------: | :------: |
> | Llama3-8B-Instruct |   LC (%)    |  WR (%)  |
> |        SFT         |    28.5     |   28.4   |
> |        DPO         |    43.7     |   41.7   |
> |       SimPO        |    44.7     |   40.4   |
> |    $\beta$-DPO     |    43.4     |   38.2   |
> |      Uni-DPO      |  **47.2**   | **44.2** |
>
> |       Model        |  Method  |  IFEval  |
> | :----------------: | :------: | :------: |
> | Llama3-8B-Instruct |   DPO    |   65.1   |
> |                    |  SimPO   |   60.8   |
> |                    | D$^2$PO  |   65.6   |
> |                    | Uni-DPO | **67.8** |
> |   Gemma-2-9b-it    |   DPO    |   60.1   |
> |                    |  SimPO   |   71.5   |
> |                    | D$^2$PO  |   71.2   |
> |                    | Uni-DPO | **72.8** |
>
> Beyond these, several recent methods study weighting or data selection in preference optimization (that are not fully open-sourced yet). These include:
>
> - **RDO [c]** observes that current offline RLHF mainly captures the ordinal relationship between responses, while overlooking how much one response is preferred over another. Thus, RDO directly predicts a differential score between two responses and uses this margin to guide training.
> - **WPO [d]** reweights preference pairs according to their likelihood under the current policy, so that off-policy data are adjusted to better match the on-policy distribution. This addresses the distribution mismatch and strengthens the optimization process without introducing extra training cost.
> - **MWPO [e]** proposes a DPO-based multi-weighting scheme that jointly considers implicit reward margins and response length margins. These two factors are geometrically combined into synthetic weights, which are then used to rescale preference pairs for optimization.
> - **RPO [f]** introduces a contrastive weighting mechanism that allows the model to train on a broader range of preference data, including both paired and unpaired examples, by assigning appropriate weights across different preference formats.
>
> In contrast, Uni-DPO **jointly integrates off-policy data quality signals and on-policy preference optimization dynamics** into a unified per-example adaptive correction mechanism. Our goal is to directly address DPO’s tendency to underutilize high-quality or difficult samples by adaptively amplifying or down-weighting examples based on their quality and model training dynamics. To the best of our knowledge, this dual perspective on preference optimization with per-sample adaptive corrections has not been explored in existing sample-wise weighting methods such as RDO, WPO, MWPO, or RPO. We have made these distinctions more explicit in the revised related work section.

---

> ### Author Response · Authors · 2025-11-20
>
> > **Response to Weakness 3 (Source of Uni-DPO’s Improvement over SimPO)**
>
> We thank the reviewer for this insightful and challenging question. First, we would like to emphasize that Uni-DPO’s empirical gains over SimPO are **consistent and robust** across a wide range of settings, rather than the result of tuning for a specific case. Uni-DPO is evaluated with **almost identical hyperparameter configurations** across multiple model families and parameter scales, and on a broad suite of benchmarks: **4 text benchmarks, 8 math benchmarks, and 10 multimodal benchmarks**. The downstream evaluation results in Appendix Table D.10 further confirm that Uni-DPO–trained models generalize well to **unseen tasks and domains**.
>
> For example, Appendix Table C.1 shows that we use the same quality-weight parameter (\eta), performance-weight parameter (\gamma), c-NLL loss coefficient (\lambda), and data threshold (\tau\_{\text{good}}) across all text setups, yet the gains generalize broadly.
>
> The reviewer also correctly identifies the core conceptual distinction between our method and SimPO, and this gives us a chance to clarify the nature of our contribution. We fully agree with the premise: our performance improvements indeed come primarily from the weighting strategy. However, we view this not as an ad-hoc heuristic, but as a principled and central contribution to preference alignment.
>
> The central thesis of our work is that the quality and structure of the supervision signal are at least as important as the specific form of the optimization objective. Both standard DPO and SimPO largely operate on a flattened, binary view of preferences (“A is better than B”), effectively treating all preference pairs as equally informative and equally reliable. This discards rich meta-information about how strong, how clear, or how trustworthy each supervision signal is.
>
> Uni-DPO’s main innovation is to provide a unified weighting framework that systematically enriches this supervision signal by combining:
>
> - Data-quality–aware weighting, which upweights cleaner and more reliable preference pairs while downweighting noisy or ambiguous ones; and
> - Model-dynamics–aware weighting, which focuses learning on pairs that are currently informative or challenging for the model.
>
> By explicitly modeling both the external quality of the data and the internal learning state of the model, Uni-DPO supplies a more nuanced and informative learning signal to the same underlying preference-optimization formulation (i.e., log-probability ratio with a reference model). In this sense, our contribution is fundamentally about **modeling and shaping the preference training signal**, rather than merely adding a heuristic on top of an existing loss.
>
> In summary, this reflects a difference in design philosophy:
>
> - SimPO’s primary contribution lies in simplifying the optimization objective (e.g., removing the reference model, introducing length normalization).
> - Uni-DPO’s primary contribution lies in **enriching the supervision signal** used for preference learning, via a unified, quality- and difficulty-aware weighting scheme.
>
> We believe that this shift of focus (from only refining the loss form to **explicitly modeling which supervision signals the model should trust and learn from more**) is a fundamental and broadly applicable principle for advancing fine-grained alignment.
>
> &nbsp;
>
> > **Response to Weakness 4 (a) (Location of Multimodal Experiments)**
>
> We thank the reviewer for pointing this out. We have indeed conducted experiments on 10 multimodal benchmarks. The details are currently organized in the appendix: the multimodal training data are described in Appendix C.2 (Multimodal training data), the evaluation setup is provided in Appendix D.1 (Evaluation setup), and the corresponding results are reported in Appendix D.2 (Results on multimodal tasks) as well as in Table D.4 (Main results on multimodal tasks).
>
> In the revised version, we have added explicit pointers in the main text to guide readers to these multimodal experiments and results, and the modifications are highlighted in **blue**, so that the empirical evidence for Uni-DPO on multimodal tasks is easier to locate.
>
> > **Response to Weakness 4 (b) (Safety or other alignment metrics)**
>
> We have conducted additional experiments and report the results in Appendix D.4 (Downstream task evaluation), with detailed comparisons presented in Table D.10. Overall, Uni-DPO not only outperforms DPO and SimPO on standard textual understanding benchmarks, but also delivers consistent improvements across a broad range of downstream tasks. We observe pronounced gains in **knowledge-intensive understanding** and **truthfulness-related evaluations**, suggesting that the proposed Uni-DPO enhances downstream robustness and factual reliability.
>
> If the reviewer believes any additional evaluation domains are necessary but not yet covered, we would be happy to conduct supplementary experiments.

---

> ### Author Response · Authors · 2025-11-20
>
> > **Response to Weakness 4 (c) (Choice of Base Models for Evaluation)**
>
> We thank the reviewer for this suggestion. In addition to LLaMA-3-8B and Gemma-2-9B, we have also conducted experiments on **Qwen2.5**, a family of models that is widely adopted in both academia and industry and represents a more recent and competitive open-source release. We believe Qwen2.5 provides a strong and contemporary testbed for validating the effectiveness of Uni-DPO. The corresponding results have been included in **Table 1** of the main text and **Table D.6** in the appendix.
>
> &nbsp;
>
> > **Response to Question 1 (Clarity and Interpretability of Figure 3)**
>
> We appreciate the reviewer’s feedback regarding Figure 3. Our intention with this figure was to provide a high-level conceptual illustration of the differences between standard DPO and Uni-DPO. However, we understand that without sufficient context, it may not effectively convey the core ideas.
>
> In the revised version, we **have updated the caption of Figure 3 and added more explicit visual cues in the figure** itself to more clearly illustrate the conceptual distinction between DPO and Uni-DPO and how Uni-DPO introduces dual-perspective adaptive weighting over data quality and learning dynamics. For example, highlighting high-quality and difficult samples while de-emphasizing the contribution of low-quality or already well-learned samples.
>
> If the reviewer has further concrete suggestions (e.g., adding specific annotations, emphasizing particular components, or redesigning certain visual elements), we would be very happy to incorporate them in a future revision to make the figure even more self-explanatory and visually interpretable.
>
> &nbsp;
>
> > **Response to Question 2 (Training Curve Comparisons and Learning Dynamics)**
>
> We agree with the reviewer that training curve comparisons help illustrate differences in learning efficiency among Uni-DPO, DPO, and SimPO. In the revised version, we provide two complementary types of results.
>
> First, Appendix Figure C.1 provides **training-dynamics curves** (e.g., loss-related and probability-related statistics), offering a more fine-grained view of how Uni-DPO learns over time. Overall, these curves show that Uni-DPO provides more stable optimization behavior than both DPO and SimPO:
>
> - Uni-DPO provides more stable training dynamics with more consistent loss curves and fewer spikes in gradient norms.
> - Uni-DPO better preserves the generation probability of positive samples.
> - Uni-DPO better separates positive and negative samples.
>
> Second, Appendix Figure D.1 presents the **performance dynamics on mathematical reasoning benchmarks**, where we periodically evaluate Uni-DPO, DPO, and SimPO on eight mathematical datasets throughout training. From these curves, Uni-DPO exhibits two clear advantages over DPO and SimPO: (1) given the same training data, it typically achieves higher average scores and can reach a higher peak performance; and (2) it maintains performance more stably in the later stages of training, where DPO and SimPO tend to fluctuate or degrade. These results further validate the effectiveness and robustness of Uni-DPO for mathematical reasoning tasks.
>
> Taken together, the benchmark performance curves and the internal training-dynamics plots provide coherent evidence supporting the design goals of Uni-DPO—namely, to leverage quality- and difficulty-aware weighting to **stabilize** and **refine** preference learning, rather than introduce additional sources of instability. We will clarify these points more explicitly in the revised manuscript. If the reviewer has specific suggestions for further diagnostics or stress tests (e.g., more extreme ablations or threshold sweeps), we would be very happy to incorporate them in future revisions.
>
> &nbsp;
>
> [a] Official code of $\beta$-DPO: Direct Preference Optimization with Dynamic $\beta$, NeurIPS, 2024
>
> [b] Earlier tokens contribute more: Learning direct preference optimization from temporal decay perspective, ICLR, 2025
>
> [c] Reward Difference Optimization For Sample Reweighting In Offline RLHF, ACL, 2024
>
> [d] Enhancing RLHF with Weighted Preference Optimization, EMNLP, 2024
>
> [e] MWPO: Enhancing LLMs Performance through Multi-Weight Preference Strength and Length Optimization, ACL, 2025
>
> [f] Relative Preference Optimization: Enhancing LLM Alignment through Contrasting Responses across Identical and Diverse Prompts, ArXiv, 2025

---

> ### Author Response · Authors · 2025-11-28
>
> ## To Reviewer Azer
>
> We thank the reviewer **Azer** again for the careful reading and detailed feedback. Following your comments, we have:
>
> - Clarified the novelty and positioning of Uni-DPO relative to existing DPO variants and weighting methods (Weakness 1, 2(b), 3).
> - Expanded the comparison with RL-based alignment methods and added stronger baselines (Weakness 2(a)).
> - Highlighted the location and results of multimodal experiments, as well as downstream and safety-related evaluations (Weakness 4(a),(b)).
> - Highlighted experiments on more recent and widely adopted base models such as Qwen2.5 (Weakness 4(c)).
> - Improved the clarity and interpretability of Figure 3, and added training curve comparisons and learning dynamics analyses (Questions 1 and 2).
> - Clarified in the main text that the strongest Uni-DPO results rely on external scalar scores not available to baseline methods, and that baselines use only paired preferences, while Uni-DPO may additionally leverage scalar signals.
>
> &nbsp;
>
>
> We hope these revisions address your concerns. If there are any remaining issues or additional points you would like us to clarify or further investigate, we would be delighted to continue the discussion and refine the manuscript accordingly.

---

### Official Review · Reviewer_5qbs · 2025-11-01

**Soundness:** 2
**Presentation:** 3
**Contribution:** 3
**Rating:** 4
**Confidence:** 3

**Summary:**

The paper proposes Uni-DPO for fine-tuning large language models with human feedback that dynamically reweights preference training pairs based on each pair’s quality and the model’s evolving performance. This addresses the limitation of standard DPO, which treats all feedback equally. By explicitly emphasizing high-quality, challenging preference pairs during training, Uni-DPO enables more effective use of data, which in turn yields better alignment performance across diverse benchmarks.

**Strengths:**

1.	The paper clearly identifies that standard DPO treats all preference pairs uniformly, which underutilizes high-quality feedback and fails to adapt to varying task difficulty. Uni-DPO tackles this by adaptive sample weighting, allowing the model to focus on informative training examples and thus improving learning efficiency. This insight targets an important problem in RLHF.
2.	The quality aware weight prioritizes pairs with larger expert score margins, while the performance aware weight, inspired by focal loss, shifts focus toward underfitted examples. This addresses observed mismatches between external score margins and the model’s reward margin.
3.	Uni-DPO shows consistent improvements on diverse tasks. Experiments show that Uni-DPO consistently outperforms vanilla DPO and SimPO baselines on multiple language understanding benchmarks, mathematical reasoning datasets, and multimodal tasks.

**Weaknesses:**

1.	A notable concern is that Uni-DPO requires a “quality score” for each preference pair, often obtained via human annotation or a powerful proxy model like GPT-4. This introduces an extra dependency on external evaluators (in effect, a form of reward signal), partially undermining the simplicity of the reward model-free DPO paradigm. If these quality scores are noisy, biased, or unavailable, it’s unclear how well the method would perform. The authors themselves acknowledge that training data quality is critical and better methods for estimating it are needed. In scenarios without reliable prior scores, the applicability of Uni-DPO could be limited or require additional effort (e.g. training a proxy reward model), which diminishes its practicality.

2.	The contribution of Uni-DPO, though useful, can be seen as a combination of existing ideas in preference optimization rather than an entirely novel invention. The method builds on known techniques: using an auxiliary NLL loss to counter DPO’s bias (as in Pal et al., 2024), applying length normalization to remove sequence-length bias, and adapting focal loss from classification to reweight easy vs hard examples. The quality-based data filtering is conceptually akin to curriculum learning or prioritizing high-confidence preferences, which have precedents in the literature, although not in the exact form used here. While the unification of these components into one framework is valuable, the paper does not derive fundamentally new theory beyond this integration, nor introduce a wholly new training paradigm. It primarily refines DPO with well-motivated heuristics.

3.	By focusing training on “hard” and high-quality examples, there is a risk, which is not deeply explored in the paper, of overfitting or bias if these weights are mis-calibrated. It would be better to introduce a calibrated performance weight to avoid instability, indicating the approach required careful handling to train reliably. It would be good to know how consistently training converges under Uni-DPO versus standard DPO. For instance, do some runs diverge or get stuck if the thresholds are set poorly?

4.	While c-NLL is proposed to counteract DPO’s preference suppression bias, the paper does not explore its impact on diversity or other alignment metrics. It also does not evaluate fairness or bias trade offs introduced by weighting high quality data more heavily.

**Questions:**

1. Could Uni-DPO be combined with reward model–based RLHF methods (e.g., PPO) to further improve alignment? How would the dual perspective weighting interact with explicit reward signals?
2. How sensitive is Uni-DPO to the source and accuracy of the quality scores $w_{qual}$? For example, if one uses a weaker judge model or noisy human preference ratings to derive the score margin, does Uni-DPO still outperform vanilla DPO by a large margin?

---

> ### Author Response · Authors · 2025-11-20
>
> ## To Reviewer 5qbs
>
> We are highly appreciative of Reviewer **5qbs** for their careful reading of our manuscript and for providing such constructive and insightful comments. We have revised the manuscript in response to all suggestions. For clear inspection, all changes made in the manuscript are highlighted in **blue**. We value your feedback greatly and would be pleased to offer any further clarification should any additional questions or concerns arise.
>
> > **Response to Weakness 1 (The need for training a scorer)**
>
> We thank the reviewer for this thoughtful and important comment. We agree that the introduction of quality weight is a key design choice, and that its dependency on external evaluators deserves discussion. Our intention is for Uni-DPO to be viewed as a _generalization_ or _extension_ of DPO, rather than a replacement of the DPO paradigm. In particular, standard DPO is recovered as a special case of Uni-DPO when all quality scores are equal (e.g., all set to 1). From this perspective, Uni-DPO does not “break” the simplicity of DPO, but instead provides a more flexible and powerful framework that can exploit richer human preference information when it is available.
>
> Importantly, Uni-DPO remains _reward-model free_ in the sense that it preserves the core advantage of the DPO paradigm: **it avoids the complex and unstable two-stage pipeline of first training an explicit reward model and then running reinforcement learning (e.g., PPO) against that reward model**. In Uni-DPO, the preference strength (quality weight) is incorporated directly into the objective as a per-example weight. We do **not** introduce a separately trained reward-model network that must be optimized together with the policy.
>
> For example, in the widely used UltraFeedback dataset, which is used in our experiments, the preference pairs are labeled by GPT-4, and the dataset already provides both the preference outcome (A is better than B) and quality scores (how much A is better than B). Prior works such as DPO and SimPO can only use the binary/ordinal preference outcome (which response wins), and thus ignore the available information about how much one response is preferred over the other. In contrast, Uni-DPO can leverage these additional supervision signals in a more fine-grained manner and perform adaptive correction throughout the preference learning.
>
> We also emphasize that Uni-DPO is **not tightly coupled to GPT-4 or any single proprietary model**. In terms of the source of quality scores, Uni-DPO is designed to be flexible. In our experiments, we demonstrate that the scores can come from multiple types of expert scorers, including Qwen2.5-72B, GPT-4, scalar-based ArmoRM, and a 7B PRM in the math setting, and they all lead to performance improvements. For example, as reported in Appendix D.3 (Further ablation study), in the paragraph _Ablation of expert model_, we investigate the impact of different expert models on Uni-DPO’s performance, with results shown in the last row of Table D.9 (we copy it below):
>
> - On the left side of the table, we apply Uni-DPO using quality scores derived from a weaker and more accessible open-source model, Qwen2.5-72B. The results show that even with this weaker expert, Uni-DPO achieves competitive performance.
> - On the right side, we use the 7.5B ArmoRM model, which is weaker and more accessible than GPT-4o, to generate scalar scores. Uni-DPO with these scores still outperforms SimPO, illustrating the robustness of our framework to the choice of scorer.
>
> These results suggest that Uni-DPO does not require a near-perfect or extremely strong evaluator to be effective. Instead, it can work with a range of reasonable scoring sources, including open-source models, which mitigates the practicality concern raised by the reviewer.
>
> |   Model & Method    | Arena-Hard |     IFEval      |     IFEval     |       Model & Method       | Arena-Hard |     IFEval      |     IFEval     |
> | :-----------------: | :--------: | :-------------: | :------------: | :------------------------: | :--------: | :-------------: | :------------: |
> |   Llama3-8B-Base    |   WR (%)   | Strict Acc. (%) | Loose Acc. (%) |     Llama3-8B-Instruct     |   WR (%)   | Strict Acc. (%) | Loose Acc. (%) |
> |         SFT         |    3.3     |      33.5       |      35.3      |            SFT             |    25.7    |        -        |       -        |
> |     DPO (GPT-4)     |    15.9    |    **40.5**     |      45.5      |            DPO             |    32.6    |        -        |       -        |
> |    SimPO (GPT-4)    |    23.4    |    **40.5**     |      45.7      |         SimPO v0.2         |    36.5    |      60.8       |      68.6      |
> |  Uni-DPO (GPT-4)   |    23.9    |      38.1       |    **47.9**    |   Uni-DPO v0.2 (GPT-4o)   |  **40.6**  |    **66.9**     |      73.2      |
> | **Uni-DPO (Qwen)** |  **24.2**  |      39.2       |      46.0      | **Uni-DPO v0.2 (ArmoRM)** |    37.5    |      66.2       |    **73.8**    |

---

> ### Author Response · Authors · 2025-11-20
>
> > **Response to Weakness 2 (The concern on novelty and relation to existing methods)**
>
> We thank the reviewer for the assessment and for recognizing that our framework is “useful” and “valuable”. We agree that Uni-DPO builds on several established ideas. Our main contribution lies in a **principled, unified, and empirically validated integration** of these components to address a critical and previously underexplored limitation of DPO: its tendency to underutilize high-quality and challenging samples when trained on heterogeneous preference data.
>
> Importantly, Uni-DPO’s empirical gains over DPO and SimPO are **consistent and robust** across a wide range of settings, rather than the result of tuning for a specific case. Uni-DPO is evaluated with **nearly identical hyperparameter configurations** across multiple model families and parameter scales, and on a broad suite of benchmarks: 4 text benchmarks, 8 math benchmarks, and 10 multimodal benchmarks.
>
> 1. Directly combining existing ideas is not sufficient
>
> We would like to clarify that simply “plugging in” these existing techniques into standard DPO does **not** yield stable or effective preference learning. In early experiments, naive combinations of standard focal loss on DPO logits and a vanilla auxiliary NLL loss either led to unstable training or failed to deliver consistent gains over the baseline DPO/SimPO, as shown in the main paper Figure 5 (b-c). This observation is supported by both our theoretical analysis and the ablation studies (Table 3: Main ablation results and Table D.7).
>
> 2. Non-trivial synergy and new gradient perspective
>
> Although the high-level ingredients (NLL, length normalization, focal-like weighting) are familiar, their synergistic integration in Uni-DPO is non-trivial. Uni-DPO is explicitly derived and analyzed from an RL-gradient perspective (Appendix E.1), where we compare the Uni-DPO gradient with that of RLHF-style methods. This analysis guides _how_ we combine quality-aware weights and difficulty-aware modulation, rather than heuristically stacking components. The resulting framework shifts the viewpoint from treating all preferences as equal binary signals to a more nuanced quality-aware preference tuning paradigm.
>
> 3. Novel extension of focal-style weighting
>
> We appreciate the reviewer’s connection between our reweighting scheme and focal loss. However, Uni-DPO does not apply focal loss in its standard form:
>
> - Different signal: Classic focal loss reweights samples purely based on the model’s _confidence in a single-label prediction_. In contrast, Uni-DPO operates on preference pairs and reweights samples based on the _preference probability margin_ between preferred and dispreferred responses, together with external quality scores. This leads to a qualitatively different weighting signal.
> - Stabilization via fixed reference margin: Our initial empirical attempts using a naive focal-style term on the DPO logits were unstable and sometimes degraded performance. To address this, we introduced a fixed reference margin (\tau\_{\text{ref}}) in the weighting, which effectively anchors the focal modulation and stabilizes training. This design is specific to the DPO-style preference setting and, to our knowledge, has not appeared in prior adaptations.
>
> 4. Novel adaptation of NLL loss
>
> Similarly, our use of NLL is not a simple addition of a standard SFT loss. Prior works use a _uniform_ NLL/SFT term over all positive responses. In contrast, Uni-DPO uses a calibrated NLL that selectively emphasizes hard, high-quality positive samples. By combining the scalar quality score and the difficulty signal, Uni-DPO focuses the auxiliary NLL on those samples that are both reliable and informative, which significantly improves performance in our ablations (Appendix Table D.7).
>
> 5. Relation to curriculum learning and data filtering
>
> We also acknowledge the reviewer’s comparison to curriculum learning and quality-based data filtering. Uni-DPO is indeed related in spirit, as it prioritizes more reliable and informative samples. However, unlike traditional curriculum learning, which relies on hand-crafted, stage-wise schedules (e.g., easy-to-hard curricula defined a priori), Uni-DPO does not use a manually designed curriculum. Instead, the unified weighting schema dynamically adjusts each sample’s contribution based on model dynamics and data quality, making the “curriculum” inherently **model-driven** rather than manually staged.
>
> In summary, while Uni-DPO builds on known components, the way they are designed and combined, in particular the unified weighting over data quality and learning dynamics, the stabilized focal-style weighting with a fixed reference margin, and the calibrated NLL that focuses on hard, high-quality samples, go beyond a simple aggregation of prior heuristics. Our theory and experiments also demonstrate that naive combinations of these ideas are ineffective, underscoring the necessity and complementarity of our design choices.

---

> ### Author Response · Authors · 2025-11-20
>
> > **Response to Weakness 3 (Concern on overfitting, bias, and training stability)**
>
> We thank the reviewer for raising the important concern about potential overfitting, bias, and instability when focusing training on “hard” and high-quality examples, especially under mis-calibrated weights and thresholds.
>
> **Hyperparameter sensitivity and stability.**
>
> We explicitly study the sensitivity of Uni-DPO to the performance-weight hyperparameters (\gamma) and (\tau_{\text{ref}}) in the main paper Section 4.4 (*Parameter Sensitivity*). The results show that Uni-DPO behaves stably within a broad and reasonable range of these hyperparameters. This robustness is largely because Uni-DPO **shifts emphasis toward** hard and high-quality samples, rather than exclusively focusing on them. Only under extremely aggressive settings (e.g., very large (\gamma) or very small (\tau_{\text{ref}})) do we observe performance degradation, which aligns with the reviewer’s intuition: such settings over-concentrate weight on a small subset of samples, potentially harming overall performance. We also provide the final hyperparameter configurations for all experiments: the text-domain settings in Appendix Table C.1, and the math and multimodal settings in Appendix Table C.3.
>
> **Overfitting and generalization across tasks and model families.**
>
> Regarding overfitting, Uni-DPO is evaluated with **almost identical hyperparameter configurations** across multiple model families and parameter scales, and across a wide range of benchmarks: 4 text benchmarks, 8 math benchmarks, and 10 multimodal benchmarks. In all these cases, Uni-DPO consistently yields improvements over DPO and SimPO, without exhibiting signs of overfitting to any single task or domain. The results of downstream tasks evaluations (in Appendix Table D.10) further confirm that Uni-DPO-trained models generalize well to unseen tasks and domains.
>
> For example, Appendix Table C.1 shows that we use the same quality-weight parameter (\eta), performance-weight parameter (\gamma), c-NLL loss coefficient (\lambda), and data threshold (\tau\_{\text{good}}) across all text setups, yet the gains generalize broadly.
>
> **Convergence behavior vs. standard DPO.**
>
> To provide a more intuitive comparison between Uni-DPO and the baselines, we include in Figure C.1 (Appendix C) the training curves of the loss, gradient norms, and the policy log-probabilities of the chosen and rejected responses, all trained on the same data for DPO, SimPO, and Uni-DPO. A detailed comparison and analysis of these dynamics is presented in Appendix C.5 (_Training Dynamics_).
>
> Overall, these results show that Uni-DPO provides more stable training dynamics than both DPO and SimPO:
>
> - Uni-DPO provides more stable training dynamics with smoother and more consistent loss curves and fewer spikes in gradient norms.
> - Uni-DPO better preserves the generation probability of positive samples.
> - Uni-DPO better separates positive and negative samples.
>
> These empirical observations are fully consistent with the design goals of Uni-DPO to leverage quality- and difficulty-aware weighting to _stabilize_ and _refine_ preference learning, rather than to introduce additional sources of instability. If the reviewer has specific suggestions for additional diagnostics or stress tests (e.g., more extreme threshold sweeps), we would be very happy to incorporate them in future revisions.
>
> &nbsp;
>
> > **Response to Weakness 4 (Impacts on diversity or other alignment metrics)**
>
> We appreciate the reviewer’s concern regarding the potential impact of Uni-DPO designs on diversity, broader alignment metrics, and possible fairness or bias trade-offs.
>
> In response, we have conducted additional experiments and report the results in Appendix D.4 (Downstream task evaluation), with detailed comparisons presented in Table D.10. Overall, Uni-DPO not only outperforms DPO and SimPO on standard textual understanding benchmarks, but also delivers consistent improvements across a broad range of downstream tasks. Notably, we observe especially pronounced gains in knowledge-intensive understanding and truthfulness-related evaluations, suggesting that the proposed c-NLL and weighting scheme do not simply overfit to preference training metrics, but also enhance downstream robustness and factual reliability.

---

> ### Author Response · Authors · 2025-11-20
>
> > **Response to Question 1 (Combine Uni-DPO with reward model–based RLHF methods)**
>
> We appreciate the reviewer’s insightful suggestion on combining Uni-DPO with reward–model-based RLHF methods such as PPO or GRPO. We are also very interested in this direction. Conceptually, Uni-DPO’s adaptive weighting mechanism is complementary to explicit reward signals in RLHF and could further enhance policy optimization.
>
> More specifically, the _data quality_ weighting in Uni-DPO could be integrated into RLHF by reweighting or reshaping the training objective, so that high-quality training samples contribute more strongly to the reward signal while the influence of low-quality samples is suppressed. This may help RL methods make better use of valuable supervision and become more robust to noisy data.
>
> On the other hand, the weighting based on _model learning dynamics_ may help RLHF focus on samples that are currently difficult for the model. For example, similar to Uni-DPO, we could compute difficulty-related weights within a batch (e.g., based on disagreement or loss) and multiply the reward signal by these weights before applying the policy gradient update in PPO-style methods. For GRPO, one could compute difficulty across multiple rollouts for the same prompt and use these difficulty scores to modulate the group-wise rewards, which naturally fits the GRPO mechanism.
>
> Overall, we believe there are multiple promising ways to combine Uni-DPO’s dual-perspective weighting with RLHF methods, and systematically exploring these combinations is an exciting direction for future work.
>
> &nbsp;
>
> > **Response to Question 2 (Sensitivity of Uni-DPO to the source and accuracy of the quality scores)**
>
> We appreciate the reviewer’s concern about the sensitivity of quality scores. In terms of the source of quality scores, Uni-DPO is designed to be flexible. In our experiments, we demonstrate that the scores can come from multiple types of expert scorers, including Qwen2.5-72B, GPT-4, GPT-4o, scalar-based ArmoRM, a 7B PRM in the math setting, and a 7B reward model in the multimodal setting, and they all lead to performance improvements. For example, as reported in Appendix D.3 (Further ablation study), in the paragraph _“Ablation of expert model”_, we investigate the impact of different expert models on Uni-DPO’s performance, with results shown in the last row of Table D.9 (we copy it below):
>
> - On the left side of the table, we apply Uni-DPO using quality scores derived from a weaker and more accessible open-source model, Qwen2.5-72B. The results show that even with this weaker expert, Uni-DPO achieves competitive performance.
> - On the right side, we use the 7.5B ArmoRM model, which is weaker and more accessible than GPT-4o, to generate scalar scores. Uni-DPO with these scores still outperforms SimPO, illustrating the robustness of our framework to the choice of scorer.
>
> These results suggest that while higher-quality scores generally lead to better performance (as is also the case for RL algorithms such as GRPO and PPO), Uni-DPO does not require a near-perfect or extremely strong external evaluator to be effective. Instead, it can work with a range of reasonable scoring sources, including open-source models and smaller reward models, which mitigates the practicality concern raised by the reviewer. We have clarified these points and the corresponding experiments more explicitly in the revised manuscript.
>
> |   Model & Method    | Arena-Hard |     IFEval      |     IFEval     |       Model & Method       | Arena-Hard |     IFEval      |     IFEval     |
> | :-----------------: | :--------: | :-------------: | :------------: | :------------------------: | :--------: | :-------------: | :------------: |
> |   Llama3-8B-Base    |   WR (%)   | Strict Acc. (%) | Loose Acc. (%) |     Llama3-8B-Instruct     |   WR (%)   | Strict Acc. (%) | Loose Acc. (%) |
> |         SFT         |    3.3     |      33.5       |      35.3      |            SFT             |    25.7    |        -        |       -        |
> |     DPO (GPT-4)     |    15.9    |    **40.5**     |      45.5      |            DPO             |    32.6    |        -        |       -        |
> |    SimPO (GPT-4)    |    23.4    |    **40.5**     |      45.7      |         SimPO v0.2         |    36.5    |      60.8       |      68.6      |
> |  Uni-DPO (GPT-4)   |    23.9    |      38.1       |    **47.9**    |   Uni-DPO v0.2 (GPT-4o)   |  **40.6**  |    **66.9**     |      73.2      |
> | **Uni-DPO (Qwen)** |  **24.2**  |      39.2       |      46.0      | **Uni-DPO v0.2 (ArmoRM)** |    37.5    |      66.2       |    **73.8**    |

---

> ### Author Response · Authors · 2025-11-27
>
> ## To Reviewer 5qbs
>
> > **Further response for Question 2 (Sensitivity to the source and accuracy of quality scores)**
>
> To further test sensitivity to weaker / noisier scores, we added a **self-rewarding** setting where the _training model itself_ (Llama3-8B-Instruct) is used to generate quality scores, with prompts and other data processing processes strictly following Appendix C.2:
>
> |           Model & Method           | Arena-Hard |     IFEval      |     IFEval     |
> | :--------------------------------: | :--------: | :-------------: | :------------: |
> |         Llama3-8B-Instruct         |   WR (%)   | Strict Acc. (%) | Loose Acc. (%) |
> |               SimPO                |    33.8    |      61.2       |      65.8      |
> |   Uni-DPO w/o $w_{\text{qual}}$    |    36.1    |      61.9       |      66.7      |
> |              Uni-DPO               |    37.3    |      61.2       |      67.8      |
> |                                    |            |                 |                |
> |             SimPO v0.2             |    36.5    |      60.8       |      68.6      |
> | Uni-DPO v0.2 w/o $w_{\text{qual}}$ |    36.6    |      62.8       |      70.2      |
> |   Uni-DPO v0.2 (self-rewarding)    |    36.9    |      66.1       |      72.8      |
> |       Uni-DPO v0.2 (ArmoRM)        |    37.5    |      66.2       |    **73.8**    |
> |       Uni-DPO v0.2 (GPT-4o)        |  **40.6**  |    **66.9**     |      73.2      |
>
> We observe that:
>
> - Even **without** quality weights, Uni-DPO (w/o $w_{\text{qual}}$) already outperforms SimPO, showing that its gains are not solely due to external scores.
> - The **self-rewarding** variant, despite relying on a relatively weak and noisy scorer, still outperforms SimPO and Uni-DPO w/o $w_{\text{qual}}$ across metrics.
> - Stronger scorers (ArmoRM, GPT-4o) further improve performance, as expected, but are _not_ required for Uni-DPO to provide clear benefits.
>
> Overall, these results suggest that Uni-DPO is robust to the source and accuracy of quality scores: better scores help, but even weak/self-generated scores are sufficient to yield improvements over strong baselines.
>
> &nbsp;
>
> We would be happy to further discuss if the reviewer has additional suggestions.

---

### Official Review · Reviewer_uYa8 · 2025-11-01

**Soundness:** 2
**Presentation:** 4
**Contribution:** 3
**Rating:** 4
**Confidence:** 3

**Summary:**

The paper proposes Uni-DPO, a “dual-perspective” extension of Direct Preference Optimization that dynamically reweights each preference pair by (i) a quality-aware weight $w_{\text{qual}}$ estimated from external expert scores and (ii) a performance-based weight $w_{\text{perf}}$ that emphasizes pairs the model currently fails to fit; it also adds a calibrated NLL term for difficult, high-quality positives. The stated goal is to correct DPO’s uniform treatment of pairs and better utilize data during preference learning for text, math, and multimodal LLMs. Empirically, the authors report consistent gains over SFT, DPO, and SimPO on instruction-following (AlpacaEval2, Arena-Hard, IFEval, SedarEval), math (GSM8K, MATH, MinervaMath, etc.), and multimodal suites, and claim Gemma-2-9B-it+Uni-DPO surpasses Claude 3 Opus on Arena-Hard.

**Strengths:**

- Clear motivation and formulation: The paper explicitly identifies DPO's uniform-pair limiattions and proposes a dual weighting + c-NLL objective.
- Empirical gains across domains: Reported improvements over SFT/DPO/SimPO on text, meth and multimodal data depicts the strength of Uni-DPO
- Ablations: The paper shows that removing each of the component degrades performance, empirically supoprting the necessity of each component.

**Weaknesses:**

- Lack of transparency and reproducibility of the scalar quality score $w_{\text{qual}}$.
    - The core of Uni-DPO's contribution, $w_{\text{qual}}$, depends on external scalar scores produced by "expert evaluators" (e.g., GPT-4). However, the exact procedure is not disclosed. Since $w_{\text{qual}}$ is an important component of Uni-DPO, the procedure of obtaining this score should be thoroughly described in the paper.
- Insufficient engagement with existing sample-wise weighting preference optimization (PO) literature.
    - There are no related work section regarding the prior methods that apply per-sample or instance-wise weighting in PO[a,b,c,d]. This weakens the novelty and makes it unclear whether if Uni-DPO has any overlaps between the related works.
- Potential unfairness from applying Length Normalization (LN) only to Uni-DPO.
    - The paper notes that LN is used when computing rewards or scores for Uni-DPO, but not for baseline method such as DPO in Table 1. As LN can boost the performance regardless of the underlying loss, the fairness of the baseline is questionable.
- External scorer dependence and fairness
    - The $w_{\text{qual}}$ depends on strong proprietary LLM scorers, which hinders the fairness against baselines trained with weaker/cheaper labels. How is using these external strong models justified?
- If these weaknesses are taken care of, I am willing to raise my score.

[a] Reward Difference Optimization For Sample Reweighting In Offline RLHF, ACL, 2024

[b] Enhancing RLHF with Weighted Preference Optimization, EMNLP, 2024

[c] MWPO: Enhancing LLMs Performance through
Multi-Weight Preference Strength and Length Optimization, ACL, 2025

[d] Relative Preference Optimization: Enhancing LLM Alignment through Contrasting Responses across Identical and Diverse Prompts, ArXiv, 2025

**Questions:**

- Described in the Weakness section.

---

> ### Author Response · Authors · 2025-11-20
>
> ## To Reviewer uYa8
>
> We sincerely thank Reviewer **uYa8** for careful reading of our manuscript and for the constructive and insightful comments. We have revised the manuscript accordingly, and all changes are highlighted in **blue** for ease of inspection. (The hyperlink may not display correctly in the browser; please download the PDF to view it.) In this rebuttal, we address each comment in detail and explicitly point to the corresponding modifications in the revised version. We greatly appreciate your feedback, and we would be very happy to provide further clarifications if there are any additional questions or concerns.
>
> > **Response to Weakness 1 (The transparency and reproducibility of the scalar quality score)**
>
> We appreciate the reviewer's concern regarding the transparency and reproducibility of the scalar quality score used in $w_{\text{qual}}$. As mentioned in Appendix C.2 _Training Data_, for the v0.1 version of our training data, the quality score of each preference pair is directly provided by the publicly available UltraFeedback dataset. Unlike DPO and SimPO, which only exploit paired preference signals, Uni-DPO leverages these supervision signals in a more fine-grained manner and performs adaptive correction throughout the preference optimization process.
>
> For the v0.2 version of the training data, which utilizes preference labels from a stronger reward model, we obtain the scalar quality scores using GPT-4o. These scores are produced via API calls, and the results in Table 1 (main paper) demonstrate the effectiveness of this setup. Besides, as shown in the v0.2 lines of Table 1 and the ablation study in Table D.9, regardless of the form of the reward (generative or scalar) and regardless of the strength of the reward model, Uni-DPO consistently outperforms the baseline DPO and SimPO. Moreover, the more accurate the reward signal is, the better the performance, which is also the case for reinforcement learning algorithms such as GRPO and PPO.
>
> We thank the reviewer for pointing out the need for more detailed documentation of the scoring procedure, and **have updated the manuscript** to provide a more thorough description of this procedure in Appendix C.2 _Training Data_. Specifically, for the textual training data, we now include: the model checkpoint used to obtain the quality scores, the exact prompts used for scoring, the aspects the model is instructed to consider when assigning scores, the post-processing progress of scores, and the range of the final scores.
>
> For the mathematical training data, we also specify the temperature and other decoding hyperparameters during sampling, as well as details of the data preprocessing pipeline, scoring prompts, and post-processing used to derive the final training data. For the multimodal training data, we also provide the scoring prompts and other construction details.
>
> Furthermore, we will release all training data and code upon acceptance of the paper to ensure full reproducibility of our results. If the reviewer has any additional suggestions regarding transparency or reproducibility, we would be very grateful to receive them and are happy to incorporate further clarifications.

---

> ### Author Response · Authors · 2025-11-20
>
> > **Response to Weakness 2 (Engagement with existing sample-wise weighting preference optimization literature)**
>
> We thank the reviewer for highlighting prior work on sample-wise / instance-wise weighting in preference optimization, and for pointing us to the references [a–d]. We have now **explicitly cited and discussed these four works** in the revised manuscript. In particular, we added a dedicated discussion of adaptive weighting strategies for Direct Preference Optimization in the appendix section F.3.
>
> Concretely, we have revised the manuscript to discuss the following lines of work:
>
> - **RDO [a]** observes that current RLHF mainly captures the ordinal relationship between responses, while overlooking how much one response is preferred over another. Thus, RDO directly predicts a differential score between two responses and uses this margin to guide training.
> - **WPO [b]** reweights preference pairs according to their likelihood under the current policy, so that off-policy data are adjusted to better match the on-policy distribution. This addresses the distribution mismatch and strengthens the optimization process without introducing extra training cost.
> - **MWPO [c]** proposes a DPO-based multi-weighting scheme that jointly considers implicit reward margins and response length margins. These two factors are geometrically combined into synthetic weights, which are then used to rescale preference pairs for optimization.
> - **RPO [d]** introduces a contrastive weighting mechanism that allows the model to train on a broader range of preference data, including both paired and unpaired examples, by assigning appropriate weights across different preference formats.
>
> In contrast, Uni-DPO **jointly integrates off-policy data quality signals and on-policy preference optimization dynamics** into a unified per-example adaptive correction mechanism. Our goal is to directly address DPO’s tendency to underutilize high-quality or difficult samples by adaptively amplifying or down-weighting examples based on their quality and model training dynamics. To the best of our knowledge, this dual perspective on preference optimization with per-sample adaptive corrections has not been explored in existing sample-wise weighting methods such as RDO, WPO, MWPO, or RPO. We have made these distinctions more explicit in the revised related work section and eagerly welcome any further suggestions from the reviewer.
>
> [a] Reward Difference Optimization For Sample Reweighting In Offline RLHF, ACL, 2024
>
> [b] Enhancing RLHF with Weighted Preference Optimization, EMNLP, 2024
>
> [c] MWPO: Enhancing LLMs Performance through Multi-Weight Preference Strength and Length Optimization, ACL, 2025
>
> [d] Relative Preference Optimization: Enhancing LLM Alignment through Contrasting Responses across Identical and Diverse Prompts, ArXiv, 2025
>
> &nbsp;
>
> > **Response to Weakness 3 (Applying Length Normalization to baseline DPO)**
>
> We thank the reviewer for the suggestion to apply length normalization (LN) to the baseline DPO. As reported in the SimPO paper (Appendix H and Table 15), the performance of vanilla DPO with and without LN does not differ significantly, which is consistent with our preliminary experiments and is why we chose the standard DPO formulation as our baseline. Moreover, when both methods use LN, Uni-DPO still significantly outperforms SimPO, suggesting that the improvements of Uni-DPO cannot be attributed solely to the introduction of LN. That said, if the reviewer believes that adding LN to the baseline DPO would make the comparison fairer, we are happy to include the corresponding experimental results in the next version.

---

> ### Author Response · Authors · 2025-11-20
>
> > **Response to Weakness 4 (The need for an external scorer)**
>
> We appreciate the reviewer's concern that Uni-DPO benefits from a high-quality expert (judge) model. Indeed, all RL-based optimization methods (like DPO, PPO, and GRPO) basically rely on accurate training signals to guide the policy optimization. Recent work on improving reward models is driven by the community's recognition that better judges yield stronger downstream performance.
>
> Theoretically, similar to other RL-based methods, under the Uni-DPO framework, the more precise the reward model, the more effective the resulting policy updates. Empirically, we strictly follow related works (like SimPO) in using **the same publicly available dataset**, UltraFeedback, with preference samples and quality scores already given by the dataset as our training signal. Compared to SimPO, Uni-DPO leverages these resources more finely to facilitate adaptive corrections during preference learning. As shown in Tables 1, 2 (main paper) and D.4 (appendix), this yields consistently better results across modality and benchmarks.
>
> To further address the reviewer's concern, we conducted additional experiments, which are shown in Appendix D.3 (further ablation study). In the paragraph "Ablation of expert model", we investigate the impact of different expert models on Uni-DPO's performance, with results shown in the last row of Table D.9 (we copy it below). On the left, we apply the Uni-DPO framework using data quality scores derived from a weaker and more accessible open-source model, Qwen2.5-72B. The results indicate that even with a weaker expert model, Uni-DPO still achieves competitive performance. On the right, we use the 7.5B ArmoRM model, which is weaker and more accessible than GPT-4o, for scoring. The results also outperformed SimPO, demonstrating the stability and scalability of Uni-DPO.
>
> |   Model & Method    | Arena-Hard |     IFEval      |     IFEval     |       Model & Method       | Arena-Hard |     IFEval      |     IFEval     |
> | :-----------------: | :--------: | :-------------: | :------------: | :------------------------: | :--------: | :-------------: | :------------: |
> |   Llama3-8B-Base    |   WR (%)   | Strict Acc. (%) | Loose Acc. (%) |     Llama3-8B-Instruct     |   WR (%)   | Strict Acc. (%) | Loose Acc. (%) |
> |         SFT         |    3.3     |      33.5       |      35.3      |            SFT             |    25.7    |        -        |       -        |
> |     DPO (GPT-4)     |    15.9    |    **40.5**     |      45.5      |            DPO             |    32.6    |        -        |       -        |
> |    SimPO (GPT-4)    |    23.4    |    **40.5**     |      45.7      |         SimPO v0.2         |    36.5    |      60.8       |      68.6      |
> |  Uni-DPO (GPT-4)   |    23.9    |      38.1       |    **47.9**    |   Uni-DPO v0.2 (GPT-4o)   |  **40.6**  |    **66.9**     |      73.2      |
> | **Uni-DPO (Qwen)** |  **24.2**  |      39.2       |      46.0      | **Uni-DPO v0.2 (ArmoRM)** |    37.5    |      66.2       |    **73.8**    |
>
> Therefore, a strong expert model is not a prerequisite for the good performance of Uni-DPO. Instead, the outstanding performance of Uni-DPO can be attributed to its ability to dynamically assign focus to each training example.

---

> ### Comment · Reviewer_uYa8 · 2025-11-25
>
> I thank the authors for the detailed responses. Although my concerns regarding weaknesses 1-3 are resolved, I still have some unresolved concerns regrading weakness 4.
>
> Uni-DPO has three main components; $w_{qual}$, $w_{pref}$, and $\mathcal{L}_{c-\text{NLL}}$.
>
> I agree with the usefulness of $w_{pref}$, but using external supervision score $\mathcal{S}$ apart from the preference label itself seems like an unfair training setup. For example, $w_{qual}$ can be applied to other baselines such as DPO or SimPO as a sample-wise weight to magnify the effect of high-quality pairs and diminish the ones of low-quality pairs.
>
> Can the authors show that measuring such scores with the base model(e.g., Llama3-8b-Instruct) is sufficient for Uni-DPO to exceed the existing baselines without the help of external model's guidance? I believe resolving this is an important part to highlight the contribution and the effectiveness of Uni-DPO.

---

> > ### Author Response · Authors · 2025-11-27
> >
> > > **Response to remaining concern on external supervision scores**
> >
> > We thank the reviewer for the follow-up comments and for raising the question about relying on external supervision scores. Although the proposed _self-rewarding_ setting is somewhat orthogonal to our original focus, we agree it is an interesting and important direction, and we have accordingly conducted additional experiments. The results are summarized in the table below. The row **Uni-DPO v0.2 (self-rewarding)** is newly added and uses the _training model itself_ (Llama3-8B-Instruct) to score the training data. The scoring prompts and all data processing strictly follow Appendix C.2 _Training Data_. The other rows are taken from Table 1 (main result), Table 3 (ablation study), and Appendix Table D.9 of the paper.
> >
> > |           Model & Method           | Arena-Hard |     IFEval      |     IFEval     |
> > | :--------------------------------: | :--------: | :-------------: | :------------: |
> > |         Llama3-8B-Instruct         |   WR (%)   | Strict Acc. (%) | Loose Acc. (%) |
> > |               SimPO                |    33.8    |      61.2       |      65.8      |
> > |   Uni-DPO w/o $w_{\text{qual}}$    |    36.1    |      61.9       |      66.7      |
> > |              Uni-DPO               |    37.3    |      61.2       |      67.8      |
> > |                                    |            |                 |                |
> > |             SimPO v0.2             |    36.5    |      60.8       |      68.6      |
> > | Uni-DPO v0.2 w/o $w_{\text{qual}}$ |    36.6    |      62.8       |      70.2      |
> > |   Uni-DPO v0.2 (self-rewarding)    |    36.9    |      66.1       |      72.8      |
> > |       Uni-DPO v0.2 (ArmoRM)        |    37.5    |      66.2       |    **73.8**    |
> > |       Uni-DPO v0.2 (GPT-4o)        |  **40.6**  |    **66.9**     |      73.2      |
> >
> > From the comparison, we see that even **without** the quality weight (Uni-DPO w/o $w_{\text{qual}}$), Uni-DPO still outperforms SimPO on both Arena-Hard and IFEval. This indicates that the gains of Uni-DPO do not solely rely on external quality signals, and that the weighting over learning dynamics and the calibrated NLL loss already bring clear improvements over SimPO.
> >
> > In the **v0.2** setting, using Llama3-8B-Instruct itself to produce quality scores (self-rewarding) yields lower scores than using stronger teachers such as ArmoRM or GPT-4o, which is expected given the base model’s more limited ability to judge fine-grained preference gaps. Nevertheless, the self-rewarding variant **still outperforms SimPO and the Uni-DPO variant without quality weighting** across benchmarks, showing that even a relatively weak self-scoring signal is sufficient for Uni-DPO to retain its advantages.
> >
> > We view this as evidence of the _stability and generality_ of Uni-DPO, which can benefit from stronger external reward models when available but remains effective when restricted to self-generated scores without external guidance.
> >
> > We hope these additional results help clarify our design choices, and we would be very happy to provide further analyses if there are any remaining questions or suggestions.

---

> > > ### Comment · Reviewer_uYa8 · 2025-11-28
> > >
> > > I thank the authors for the additional experiments and clarifications. The self-rewarding results in particular address my remaining concern: Uni-DPO still outperforms SimPO even when its quality scores come from the base model itself. This shows the method’s benefits are not solely due to external supervision.
> > >
> > > With these updates, I am happy to raise my score to 6 (weak accept).
> > >
> > > I would, however, still encourage the authors to make the following points clearer in the final version:
> > > The strongest Uni-DPO results rely on external scalar scores not available to baseline methods. This should be stated clearly in the main paper so the comparison is not misinterpreted. Also, make it explicit that baselines use only paired preferences, while Uni-DPO may use additional scalar signals from external models.
> > >
> > > With these clarifications, I believe the contribution is solid, and I support acceptance.

---

> > > > ### Author Response · Authors · 2025-11-28
> > > >
> > > > Thanks for the review, and we appreciate the reviewer’s careful reading and constructive suggestions. We will explicitly state in the main paper that our strongest Uni-DPO results use external scalar scores (which are not available to the baseline methods). These clarifications will be added to the revised manuscript to ensure a fair and transparent comparison.
> > > >
> > > > We are grateful for your detailed review and for your meaningful suggestions.

---

### Author Response · Authors · 2025-12-03
**Summary of Rebuttal**

Dear Area Chairs (ACs),

Thank you and all the reviewers for handling our submission and for the careful, constructive reviews. We greatly appreciate the time and effort invested by you and all the reviewers. In response to the reviewers' constructive and valuable feedback, we have provided detailed, point-by-point replies and updated the manuscript accordingly.

Notably, during the discussion period, one reviewer (**uYa8**) engaged in multiple rounds of discussion with us and, after carefully examining our additional experiments and clarifications, expressed that **the contribution of our Uni-DPO is solid** and **raised their score** (now recommending acceptance). We are grateful for this support.

We reviewed the other three reviewers' reports and observed that their core concerns are similar to those raised by uYa8. Because our new analyses directly and thoroughly addressed the same issues that convinced uYa8, we expect that, had similar back-and-forth been possible with the other reviewers, many of their concerns would likely have been alleviated as well. Unfortunately, due to the system reset, we have not received any further feedback from the other reviewers.

Before the system reset, our rebuttal and revisions successfully addressed the main concerns raised by all reviewers, resulting in a score increase. Below, we summarize the current review thread, the reviewers' praises and main concerns, the main replies and modifications we made to the paper for your convenience.

Sincerely,

The authors

---

> ### Author Response · Authors · 2025-12-03
> **Overall status and reviewer positive feedback**
>
> The original reviews highlighted many strengths of our work, including **clear motivation**, **solid empirical gains across domains and model families**, **principled analysis targeted at practical problems**, and a **practically useful weighting mechanism**. These positive comments were both specific and comprehensive, reflecting the multifaceted advantages of Uni-DPO. We are grateful for this encouraging feedback. In brief:
>
> - **uYa8:** Commended the clear motivation and formulation, great empirical improvements across domains, and ablations that support the necessity of each component.
> - **5qbs:** Noted that the Uni-DPO targets an important problem in the RLHF area, and praised the unified weighting scheme for addressing mismatches between external score margins and a model's internal reward margin; reported consistent gains across diverse tasks.
> - **Azer:** Appreciated the closed-form derivation of Uni-DPO's gradient coefficient and the preservation of single-stage preference training simplicity, while observing consistent benchmark improvements from the unified weighting.
> - **HSvf:** Highlighted the main strength as the dynamic per-sample adjustment that combines data quality and model learning performance, effectively incorporating advantage-style ideas from RL to improve training efficiency.
>
> During rebuttal, we responded to all reviewer comments point-by-point and revised the manuscript accordingly. Notably, reviewer **uYa8** engaged in multiple rounds of discussion and, after carefully reviewing our additional experiments and clarifications, expressed that **the contribution of our Uni-DPO is solid** and **raised their score** to Weak Accept. We greatly appreciate this support.
>
> Because our new experiments and clarifications address many of the same substantive issues that other reviewers raised, we believe that, had similar interactions been possible with them, a comparable increase in confidence could reasonably have followed. Below, we summarize the main concerns raised by all reviewers and how we addressed each point in our rebuttal and revision.

---

> ### Author Response · Authors · 2025-12-03
> **Main reviewer concerns and our responses**
>
> Here, we summarize the main concerns raised by the reviewers and how we addressed each point in our rebuttal and revision, intending to provide you with a clear and convenient overview of the discussion.
>
> 1. **Relation to existing methods**
>
>    - Concern: Uni-DPO appears to combine known components, questioning whether gains are simply from heuristics or weighting.
>    - What we did: We clarified the design philosophy and provided theoretical analysis (gradient-level perspective) and ablations showing that naive combinations fail. We emphasize three distinctive design elements: (a) a unified per-example weighting that jointly models external data quality and on-policy learning dynamics, (b) a stabilized focal-style modulation anchored with a fixed reference margin, and (c) a calibrated NLL that selectively emphasizes hard, high-quality positive examples. Ablation tables and plots (Table 3, Table D.7, Fig. 5) demonstrate that removing or naively combining components harms general performance and training stability. Only the **full Uni-DPO design** achieves strong, consistent gains across tasks and model scales.
>
> 2. **Sensitivity to scorer source**
>
>    - Concern: Are gains dependent on using strong external scalar scorers? Are results robust to weaker/noisy scorers?
>    - What we did: We ran additional experiments, where weaker and more accessible open source reward models and even the training model itself provide quality scores (“self-rewarding”). Key observations: (a) Uni-DPO variants _without_ quality weighting already outperform DPO and SimPO (i.e., the learning-dynamics and calibrated NLL are useful by themselves); (b) the self-rewarding variant still outperforms SimPO and Uni-DPO w/o $w_{\text{qual}}$, showing robustness to weaker/noisy scorers; (c) stronger scorers (ArmoRM, GPT-4o) yield additional gains as expected. These findings demonstrate Uni-DPO's effectiveness even with weaker or self-generated quality signals, broadening its applicability.
>
> 3. **Baselines / RL comparisons**
>
>    - Concern: Need more, stronger baselines (including modern RL-based alignment methods).
>    - What we did: We extended experiments to include SFT baselines (Eurus2-SFT, RAFT), preference alignment methods (DPO, SimPO, Iterative DPO), and RL-based methods (Eurus2-PRIME, OpenReasoner, SimpleRL, Dr. GRPO, PPO). Results are in Appendix Table D.5. Uni-DPO remains highly competitive and often **outperforms these baselines while being simpler and more data efficient**. We explicitly note that Iterative DPO uses many more training iterations/data, but is still outperformed by our Uni-DPO.
>
> 4. **Concern on overfitting, bias, and diversity**
>
>    - Concern: Focusing training on hard and high-quality examples, there is a risk of overfitting or bias.
>    - What we did: We highlighted the hyperparameter sensitivity and stability results of the paper; clarified that Uni-DPO is evaluated with almost **identical hyperparameter configurations** across multiple model families and parameter scales, and across a wide range of benchmarks; We also conducted additional experiments and demonstrated that Uni-DPO also gains in diversity, knowledge-intensive understanding, and truthfulness-related evaluations.
>
> 5. **Training dynamics and figure clarity**
>
>    - Concern: Figures (e.g., Fig. 3) and training curves were not sufficiently clear.
>    - What we did: We updated the Figure 3 caption and added annotations; provided training-dynamics visualizations (Appendix Fig. C.1 and D.1) showing loss/probability/gradient behavior and periodic performance evaluation across benchmarks. These plots demonstrate Uni-DPO's more stable optimization and higher peak/stable late-stage performance compared to DPO and SimPO, which align with our theoretical motivations.
>
> 6. **Comparison with DPO variants and weighting literature**
>
>    - Concern: There are existing weighting/data-selection DPO variants.
>    - What we did: We added a clear related-work comparison and empirical comparisons with representative variants (e.g., $\beta$-DPO, D$^2$PO) plus discussion of RDO/WPO/MWPO/RPO-style works. We highlight conceptual differences: other methods typically handle either on-policy dynamics or intra-example token weighting or correct distribution mismatch; Uni-DPO uniquely integrates _both_ off-policy data-quality signals and on-policy learning-dynamics at the per-example level.
>
> 7. **Lack of transparency and reproducibility**
>
>    - Concern: The scalar quality score used in our experiments lacks transparency and reproducibility.
>    - What we did: We have updated the manuscript to provide a more thorough description of this procedure in Appendix C.2 Training Data.
>
> We hope this detailed summary helps you in your assessment of our submission. We believe our thorough responses and revisions address the key concerns raised and significantly strengthen the manuscript.
>
> Sincerely,
>
> The authors

---

### Meta-Review · Area_Chair_gWC8 · 2025-12-31

**Summary:**

The reviewers' initial reservations primarily centered on algorithmic fairness and the source of improvements. A major concern was that Uni-DPO’s gains stemmed from *privileged* external information (scalar quality scores from GPT-4) rather than a superior optimization strategy. Reviewers also questioned the conceptual novelty, suggesting the method might be a heuristic combination of existing techniques like Focal Loss and NLL. Concerns regarding reproducibility and the lack of modern RL baselines (e.g., PPO, GRPO) further contributed to initial marginal ratings.

The decision to recommend **acceptance** was informed by the authors' rigorous rebuttal, specifically a **self-rewarding** experiment which proved that Uni-DPO outperforms baselines even when using its own (weaker) scores. This demonstrated that the dual-perspective weighting mechanism is a robust algorithmic contribution. Furthermore, the authors provided a gradient-level analysis to justify their specific architectural choices and added extensive comparisons showing the method is more data-efficient than iterative DPO and RL-based alternatives.

**Reviewer Concerns:**

The authors provided very detailed rebuttals. Thus, I believe that most of the reviewers' concerns have been addressed. In particular, one reviewer (uYa8) participated in the discussion very actively, and he/she was satisfied with the authors' responses.

**Reviewer Scores:**

As aforementioned, the only one reviewer (uYa8) who participated in the discussion was satisfied by with the authors' responses. Because the concerns of the other reviewers are similar to those of Reviewer uYa8, I expect that at least two reviewers would raise their score if they had been able to participate fully in the discussion.

---

### Decision · Program_Chairs · 2026-01-26

Accept (Poster)